# Selecting fitted models under epistemic uncertainty using a stochastic process on quantile functions

**Alexandre René** [1,2,3] ✉ **& André Longtin** [3,4,5]

Fitting models to data is an important part of the practice of science. Advances in machine learning have made it possible to fit more—and more complex—models, but have also exacerbated a problem: when multiple models fit the data equally well, which one(s) should we pick? The answer depends entirely on the modelling goal. In the scientific context, the essential goal is replicability: if a model works well to describe one experiment, it should continue to do so when that experiment is replicated tomorrow, or in another laboratory. The selection criterion must therefore be robust to the variations inherent to the replication process. In this work we develop a nonparametric method for estimating uncertainty on a model's empirical risk when replications are non-stationary, thus ensuring that a model is only rejected when another is reproducibly better. We illustrate the method with two examples: one a more classical setting, where the models are structurally distinct, and a machine learning-inspired setting, where they differ only in the value of their parameters. We show how, in this context of replicability or "epistemic uncertainty", it compares favourably to existing model selection criteria, and has more satisfactory behaviour with large experimental datasets.

Much of our understanding of the natural world is built upon mathematical models. But how we build those models evolves as new techniques and techonologies are developed. With the arrival of machine learning methods, it has become even more feasible to solve inverse problems and learn complex descriptions by applying data-driven methods to scientifically-motivated models[1-3]. However, even moderately complex models are generally non-identifiable: many different parameter sets may yield very similar outputs[4-6]. With imperfect real-world data, it is often unclear which—if any—of these models constitutes a trustable solution to the inverse problem.

This presents us with a model selection problem which is somewhat at odds with the usual statistical theory. First, we require a criterion which can distinguish between structurally identical models differing only in their specific parameters. Moreover, even with the most tightly controlled experiments, we expect all candidate models to be misspecified due to experimental variations. Those variations also change when the experiment is repeated in the same lab or another lab; in other words, the replication process is non-stationary, even when the data-generation process is stationary within individual trials. While there has been some work to develop model selection criteria which are robust to misspecification[7-9], and some consideration of locally stationary processes[10], the question of non-stationary replications has received little attention. Criteria which assume the data generating process to be stationary therefore underestimate *epistemic uncertainty*, which has important repercussions when they are used for scientific induction, where the goal is to learn a model which generalises. In recent years, a few authors have advanced similar arguments in the context of machine learning models[11,12].

To support data-driven scientific methods, there is therefore a need for a criterion which more accurately estimates this uncertainty

[1]Fakultät 1, RWTH Aachen, Physik, Aachen, Germany. [2]Fakultät für Informatik, RWTH Aachen, Chair of Computational Network Science, Aachen, Germany. [3]University of Ottawa, Department of Physics, Ottawa, Canada. [4]University of Ottawa, Department of Cellular and Molecular Medicine, Ottawa, Canada. [5]Center for Neural Dynamics, Ottawa, Canada. ✉e-mail: a.rene@physik.rwth-aachen.de

due to differences between our model and the true data generating process. Epistemic uncertainty may be due to learning the model from limited samples, the model being misspecified or non-identifiable, or the data generating process being non-stationary. We distinguish this from *aleatoric* uncertainty (also called sampling uncertainty)[13,14], which is due to the intrinsic stochasticity of the process.

Some of the pioneering work on this front came from the geosciences, which methods like GLUE[15–17] and Bayesian calibration[18]. More recently, we have also seen strategies to account for epistemic uncertainty in machine learning[14,19], astrophysics[20] and condensed matter physics[21]. All of these employ some form of ensemble: epistemic uncertainty is represented as a concrete distribution over models, and predictions are averaged over this distribution.

Unfortunately, ensemble models are difficult to interpret since they are generally invalid in a Bayesian sense[22]. In our view, if the goal is to find interpretable models, then an approach like that advocated by Tarantola[5] seems more appropriate: instead of assigning probabilities to an ensemble of plausible models, simply treat that ensemble as the final inference result. With a finite (and hopefully small) number of plausible models, each can be interpreted individually.

The high-level methodology Tarantola proposes comes down to the following: 1. Construct a (potentially large) set of *candidate models*. 2. Apply a *rejection criterion* to each candidate model in the set. 3. Retain from the set of candidate models those that satisfy the criterion.

Step 1 can be accomplished in different ways; for example, Prinz et al.[6] performed a grid search using a structurally fixed model of the lobster's pyloric rhythm, and found thousands of distinct parameter combinations which reproduce chosen features of the recordings. More recently, machine learning methods have also been used to learn rich mechanistic models of molecular forces[1], cellular alignment[3] and neural circuits[2,23]. Since these are intrinsically nonlinear models, their objective landscape contains a multitude of local minima, which translates to a multitude of candidate models. For a general formulation of the learning problem which fits our framing especially well, see Vapnik's *Function estimation model*[24].

The focus of our work is to present a practical criterion for step 2: **we therefore assume that we have already obtained a set of candidate models**. We make only three hard requirements. First, a candidate model $\mathcal{M}$ must be probabilistic, taking the form

$$p(y_i|x_i; \mathcal{M}), \quad (x_i, y_i) \in \mathcal{D}_{\text{test}}, \tag{1}$$

where $\mathcal{D}_{\text{test}}$ is the observed dataset and $(x_i \in \mathbb{R}^n, y_i \in \mathbb{R}^m)$ is the $i$-th input/output pair (or independent/dependent variables) that was observed. Often this takes the form of a mechanistic model with some additional observation noise. Second, it must be possible to generate arbitrarily many synthetic sample pairs $(x, y)$ following each candidate model's distribution. And third, any candidate model $\mathcal{M}$ must assign a non-vanishing probability to each of the observations:

$$p(y_i|x_i; \mathcal{M}) > 0, \quad \forall (x_i, y_i) \in \mathcal{D}_{\text{test}}. \tag{2}$$

For example, a model whose predictions $y$ are restricted to an interval $[a, b]$ is only allowed if all observations are within that interval. We also require the definition of a loss function $Q$ to quantify the accuracy of model predictions $y_i$. Taking the expectation of $Q$ over data samples yields the risk $R$, a standard measure of performance used to fit machine learning models.

Key to our approach is a novel method for assigning to each model a *risk-* or *R-distribution* representing epistemic uncertainties due either to finite samples, model misspecification or non-stationarity. Only when two models have sufficiently non-overlapping $R$-distributions do we reject the one with higher $R$. This approach allows us to compare any number of candidate models, by

reducing the problem to a sequence of pairwise comparisons. We make no assumption on the structure of those models: they may be given by two completely different sets of equations, or they may have the same equations but differ only in their parameters, as long as they can be cast in the form of equation (1).

In many cases, models will contain a "physical" component—the process we want to describe—and an "observation" component—the unavoidable experimental noise. Distinguishing these components is often useful, but it makes no difference from the point of view of our method: only the combined "physical + observation" model matters. In their simplest forms, the physical component may be deterministic and the observation component may be additive noise, so that the model is written as a deterministic function $f$ plus a random variable $\xi$ affecting the observation:

$$y_i = f(x_i; \mathcal{M}) + \xi_i.$$

In such a case, the probability in equation (1) reduces to $p(\xi_i = y_i - f(x_i; \mathcal{M}))$; if $\xi_i$ is Gaussian with variance $\sigma^2$, this further reduces to $p(y_i \mid x_i; \mathcal{M}) \propto \exp(-(y_i - f(x_i; \mathcal{M}))^2 / 2\sigma^2)$. Of course, in many cases the model itself is stochastic, or the noise is neither additive nor Gaussian. Moreover, even when such assumptions seem justified, we should be able to test them against alternatives.

We will illustrate our proposed criterion using two examples from biology and physics. The first compares candidate models of neural circuits (from the aforementioned work of Prinz et al.[6]) with identical structure but different parameters; the second compares two structually different models of black body radiation, inspired by the well-known historical episode of the "ultraviolet catastrophe".

In the next few sections, we present the main steps of our analysis using the neural circuit model for illustration. We start by arguing that modelling discrepancies are indicative of epistemic uncertainty. We then use this idea to assign uncertainty to each model's risk, and thereby define the *EMD rejection rule*. Here EMD stands for *empirical modelling discrepancy*, which describes the manner in which we estimate epistemic uncertainty; this mainly involves three steps, schematically illustrated in Fig. 1. First, we represent the prediction accuracy of each model with a quantile function $q^*$ of the loss. Second, by measuring the self-consistency of $q^*$ with the model's own predictions ($\bar{q}$), we obtain a measure ($\delta^{\text{EMD}}$) of epistemic uncertainty, from which we construct a stochastic process $\mathfrak{Q}$ over quantile functions. Since each realisation of $\mathfrak{Q}$ can be integrated to yield the risk, $\mathfrak{Q}$ thus induces the $R$-distributions we seek. This requires however the introduction of a new type of stochastic process, which we call *hierarchical beta process*, in order to ensure that realisations are valid quantile functions. Those $R$-distributions are used to compute the tail probabilities, denoted $B^{\text{EMD}}$, in terms of which the rejection rule is defined. The third step is a calibration procedure, where we validate the estimated probabilities $B^{\text{EMD}}$ on a set of simulated experiments. To ensure the soundness of our results, we used over 24,000 simulated experiments, across 16 forms of experimental variation.

The following sections then proceed with a systemic study of the method, using the simpler model of black body radiation. We illustrate how modelling errors and observation noise interact to affect the shape of $R$ distributions and ultimately the EMD rejection criterion. Finally, we compare our method with a number of other popular criteria—AIC, BIC, MDL, elpd and Bayes factors. We highlight the major modelling assumptions each method makes, and therefore in which circumstances it is likely most appropriate. Special attention is paid to the behaviour of the model selection statistics as the sample size grows, which is especially relevant to the scientific context.

## Results

### The risk as a selection criterion for already fitted models

As set out in the Introduction, we start from the assumption that models of the natural phenomenon of interest have already been

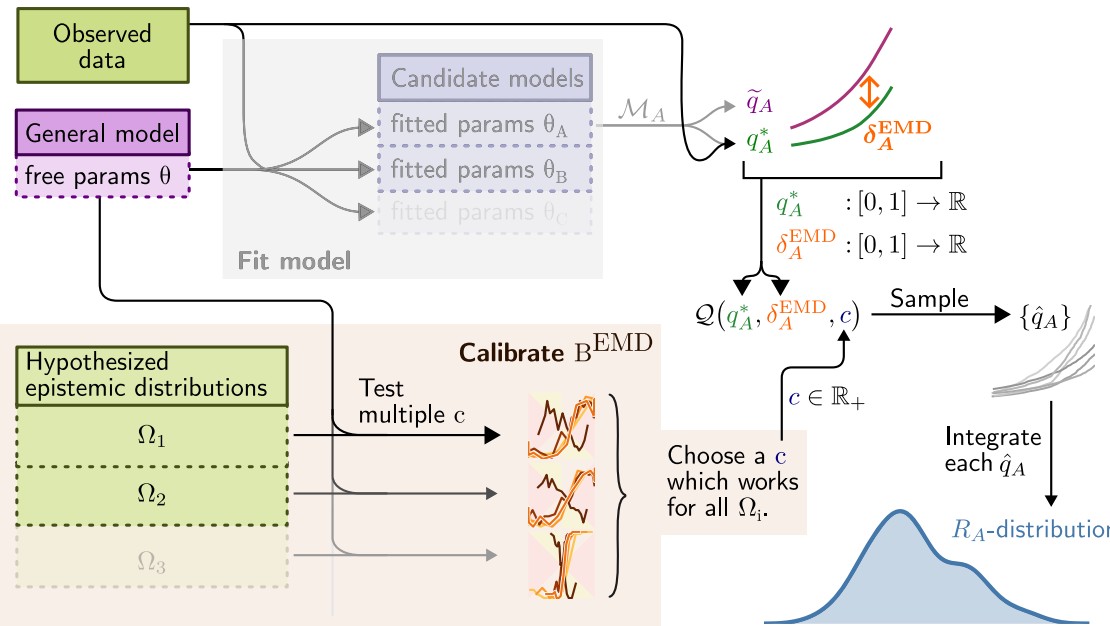

**Fig. 1 | Overview of the approach for computing $R$-distributions.** We assume the model has already been fitted to obtain a set of candidate parameter sets $\theta_A$, $\theta_B$... (not shown: the models may also be structurally different). Each candidate parameter set $\Theta_A$ defines a candidate model $\mathcal{M}_A$, for which we describe the statistics of the loss with two different quantile functions: the purely synthetic $\tilde{q}_A$ (equation (22)) which depends only on the model, and the mixed $q_A^*$ (equation (19)) which depends on both the model and the data. A small discrepancy $\delta_A^{\text{EMD}}$ (equation (23)) between those two curves indicates that model predictions concord with the observed data. Both $q_A^*$ and $\delta_A^{\text{EMD}}$ are then used to parametrise a stochastic process $\mathfrak{Q}$ which generates random quantile functions. This induces a distribution for the risk $R_A$ of model $\mathcal{M}_A$, which we ascribe to epistemic uncertainty. The stochastic process $\mathfrak{Q}$ also depends on a global scaling parameter $c$. This is independent of the specific model, and is obtained by calibrating the procedure with simulated experiments $\Omega_i$ that reflect variations in laboratory conditions. The computation steps on the right (white background) have been packaged as available software[74].

fitted to data; this anterior step is indicated by the grey faded rectangle in Fig. 1. The manner in which this is done is not important; model parameters for example could result from maximising the likelihood, running a genetic algorithm or performing Bayesian inference.

Our starting points are the true data-generating process $\mathcal{M}_{\text{true}}$ and the pointwise loss $Q$. The loss depends on the model $\mathcal{M}_A$, evaluates on individual data samples $(x_i, y_i)$ and returns a real number:

$$Q(x_i, y_i; \mathcal{M}_A) \in \mathbb{R}. \tag{3}$$

Often, but not always, the negative log likelihood is chosen as the loss. What we seek to estimate is the risk, i.e., the expectation of $Q$:

$$R_A = \mathbb{E}_{(x_i, y_i) \sim \mathcal{M}_{\text{true}}}[Q(x_i, y_i; \mathcal{M}_A)]; \tag{4}$$

we use the notation $(x_i, y_i) \sim \mathcal{M}_{\text{true}}$ to indicate that the samples $(x_i, y_i)$ are drawn from $\mathcal{M}_{\text{true}}$. For simplicity, our notation assumes that model parameters are point estimates. For Bayesian models, one can adapt the definition of the loss $Q$ to include an expectation over the posterior; equation (4) would then become the Bayes risk. Our methodology is agnostic to the precise definition of $Q$, as long as it is evaluated pointwise; it can differ from the objective used to fit the models.

The risk describes the expected performance of a model on replicates of the dataset, when each replicate is drawn from the same data-generating process $\mathcal{M}_{\text{true}}$. It is therefore a natural basis for comparing models based on their ability to generalise. It is also the gold standard objective for a machine learning algorithm[24,25], for the same reason, and has been used in classical statistics to analyse model selection under misspecification[26]. In practice we usually estimate $R_A$

from finite samples with the *empirical risk*:

$$\hat{R}_A = \frac{1}{|\mathcal{D}|} \sum_{(x_i, y_i) \in \mathcal{D}} Q(x_i, y_i; \mathcal{M}_A). \tag{5}$$

For the types of problems we are interested in, it is safe to assume that $\hat{R}_A$ converges to $R_A$ in probability.

A common consideration with model selection criteria is whether they are *consistent*, i.e., whether they eventually select the true model when the number of samples grows. When models are misspecified, none of the candidates are actually the true model, so the definition of consistency is often generalised to mean convergence to the "pseudo-true" model: the one with the smallest Kullback-Leibler divergence $D_{\text{KL}}$ from the true model[27,28]. If consistency in this sense is desired, then using the log likelihood for $Q$ is recommended, since for a fixed true model, minimising the $D_{\text{KL}}$ is equivalent to minimising the log likelihood. The log likelihood is also known to be *proper* (see e.g., references [29], [30] or [31] (Chap. 7.1)); a selection rule which minimises the log likelihood is thus also consistent in the original sense of asymptotically selecting the true model when it is among the candidates.

The statistical learning literature instead defines a "learning machine" as *consistent* if it asymptotically minimises the true risk $R$ when trained with the empirical risk $\hat{R}_A$[24,25]. As long as we have a finite number of candidate models, our procedure will satisfy this notion of consistency.

Convergence to $R_A$ also means that the result of a risk-based criterion is insensitive to the size of the dataset, provided we have enough data to estimate the risk of each model accurately. This is especially useful in the scientific context, where we want to select models which can be used on datasets of any size.

In addition to defining a consistent, size-insensitive criterion, risk also has the important practical advantage that it remains informative

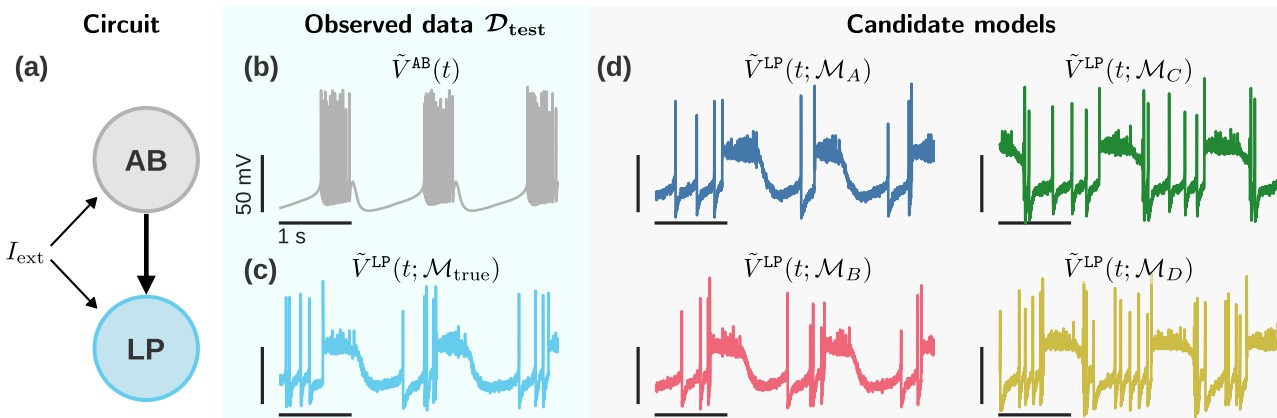

**Fig. 2 | Comparison of LP neuron responses.** The goal is to estimate a good set of candidate models for predicting the membrane potential $\tilde{V}$ of the neuron LP. **a)** We consider a simple circuit with a known pacemaker neuron (AB: anterior bursting) and a post-synaptic neuron of unknown response (LP: lateral pyloric). **b)** Output of the AB neuron. This serves as input to the LP neuron. In practice, these data would

more likely be provided by an experimental recording of neuron AB, but here we assume it is a simulatable known model for convenience. **c)** Response of neuron model LP 1 (i.e., the true model $\mathcal{M}_{\text{true}}$) to the input in (b). Together, (b) and (c) serve as our observed data $\mathcal{D}_{\text{test}}$. **d)** Response of neuron models LP 2 to LP 5 to the input in (b). These are our four candidate models ($\mathcal{M}_A$ to $\mathcal{M}_D$) for neuron LP.

when the models are structurally identical. We contrast this with the marginal likelihood (also known as the model evidence), which for some dataset $\mathcal{D}$, model $\mathcal{M}_A$ parametrised by $\theta$, and prior $\pi_A$, would be written

$$\mathcal{E}_A = \int p(\mathcal{D} \mid \mathcal{M}_A(\theta)) \, \pi_A(\theta) d\theta. \tag{6}$$

Since it is integrated over the entire parameter space, $\mathcal{E}_A$ characterises the family of models $\mathcal{M}_A(\cdot)$, but says nothing of a specific parametrisation $\mathcal{M}_A(\theta)$. We include a comparison of our proposed EMD rejection rule to other common selection criteria, including the model evidence, at the end of our Results.

A criterion based on risk however will not account for a model which overfits the data, which is why it is **important to use separate datasets for fitting and comparing models**. This in any case better reflects the usual scientific paradigm of recording data, forming a hypothesis, and then recording new data (either in the same laboratory or a different one) to test that hypothesis. Splitting data into training and test sets is also the standard practice in machine learning to avoid overfitting and ensure better generalisation. In the following therefore we will always denote the observation data as $\mathcal{D}_{\text{test}}$ to stress that they should be independent from the data to which the models were fitted.

In short, the EMD rejection rule we develop in the following sections ranks models based on their empirical risk equation (5), augmented with an uncertainty represented as a risk-distribution. Comparisons based on the risk are consistent and insensitive to the number of test samples, but require a separate test dataset to avoid overfitting.

**Example application: Selecting among disparate parameter sets of a biophysical model**

To explain the construction of the EMD rejection rule, we use the dynamical model for a neuron membrane potential described by Prinz et al.[6] This choice was motivated by the fact that fitting a neuron model is a highly ill-posed problem, and therefore corresponds to the situation we set out in the Introduction: disparate sets of parameters for a structurally fixed model which nevertheless produce similar outputs. We will focus on the particular LP neuron type; Prinz et al.[6] find five distinct parameter sets which reproduce its experimental characteristics. Throughout this work, we reserve LP 1 as the model that generates through simulation the true data $\mathcal{D}_{\text{test}}$, and use LP 2 to LP 5 to define the

candidate models $\mathcal{M}_A$ to $\mathcal{M}_D$ which we compare against $\mathcal{D}_{\text{test}}$. We intentionally exclude LP 1 from the candidate parameters to emulate the typical situation where none of the candidate models fit the data perfectly. Visual inspection of model outputs suggests that two candidates (models $\mathcal{M}_A$ and $\mathcal{M}_B$) are more similar to the true model output (Fig. 2). We will show that our method not only concords with those observations, but makes the similarity between models quantitative.

It is important to note that we treat the models $\mathcal{M}_A$ to $\mathcal{M}_D$ as simply given. We make no assumption as to how they were obtained, or whether they correspond to the maximum of a likelihood. We chose this neural example, where parameters are obtained with a multi-stage, semi-automated grid search, partly to illustrate that our method is agnostic to the fitting procedure.

The possibility of comparing models visually was another factor in choosing this example. Since numbers can be misleading, this allows us to confirm that the method works as expected. Especially for our target application where all candidate models share the same equation structure, a visual validation is key to establishing the soundness of the $B^{\text{EMD}}$, since none of the established model selection like the Bayes factor, AIC or WAIC[31,32] are applicable. Indeed, these other methods only compare alternative equation structures, not alternative parameter sets. We include a more in-depth comparison to these other methods at the end of the Results.

The datasets in this example take the form of one-dimensional time series, with the time $t \in \mathcal{T}$ as the independent variable and the membrane potential $V^{\text{LP}} \in \mathcal{V}$ as the dependent variable. We denote the space of all possible time-potential tuples $\mathcal{T} \times \mathcal{V}$. Model specifics are given in the Methods; from the point of view of model selection, what matters is that we have ways of generating series of these time-potential tuples: either using the true data-generating process ($\mathcal{M}_{\text{true}}$) or one of the candidate models ($\mathcal{M}_A$ to $\mathcal{M}_D$).

We will assume that the dataset used to evaluate models is composed of $L$ samples, with each sample a $(t, \tilde{V}^{\text{LP}})$ tuple:

$$\mathcal{D}_{\text{test}} = \left\{ (t_k, \tilde{V}^{\text{LP}}(t_k; \mathcal{M}_{\text{true}}, I_{\text{ext}})) \in \mathcal{T} \times \mathcal{V} \right\}_{k=1}^{L}. \tag{7}$$

The original model by Prinz et al.[6] produced deterministic traces $V^{\text{LP}}$. Experimental measurements however are variable, and our approach depends on that variability. For this work we therefore augment the model with two sources of stochasticity. First, the system as a whole receives a coloured noise input $I_{\text{ext}}$, representing an external current

received from other neurons. (External currents may also be produced by the experimenter, to help model the underlying dynamics.) Second, we think of the system as the combination of a biophysical model–described by equations (50) to (52)–and an observation model which adds Gaussian noise. The observation model represents components which don't affect the biophysics–like noise in the recording equipment–and can be modelled to a first approximation as:

$$\tilde{V}^{\text{LP}}(t_k; \mathcal{M}_{\text{true}}) = \underbrace{V^{\text{LP}}(t_k, I_{\text{ext}}(t_k; \tau, \sigma_i); \mathcal{M}_{\text{true}})}_{\text{biophysical model}} + \underbrace{\xi(t_k; \sigma_o)}_{\text{observation model}} . \quad (8)$$

The parameters of the external input $I_{\text{ext}}$ are $\tau$ and $\sigma_i$, which respectively determine its autocorrelation time and strength (i.e., its amplitude). The observation model has only one parameter, $\sigma_o$, which determines the strength of the noise. The symbol $\mathcal{M}_{\text{true}}$ is shorthand for everything defining the data-generating process, which includes the parameters $\tau$, $\sigma_i$ and $\sigma_o$. The indices $i$ and $o$ are used to distinguish *input* and *output* model parameters.

Since the candidate models in this example assume additive observational Gaussian noise, a natural choice for the loss function is the negative log likelihood:

$$\begin{aligned} Q(t_k, \tilde{V}^{\text{LP}}; \mathcal{M}_a) &= -\log p(\tilde{V}^{\text{LP}} \mid t_k, \mathcal{M}_a) \\ &= \frac{1}{2}\log(2\pi\sigma_o) - \frac{(\xi_i)^2}{2\sigma} , \\ &= \frac{1}{2}\log(2\pi\sigma_o) - \frac{(\tilde{V}^{\text{LP}} - V^{\text{LP}}(t_k; \mathcal{M}_a))^2}{2\sigma_o} . \end{aligned} \quad (9)$$

However one should keep in mind that a) it is not necessary to define the loss in terms of the log likelihood, and b) the loss used to compare models need not be the same used to fit the models. For example, if the log likelihood of the assumed observation model is non-convex or non-differentiable, one might use a simpler objective for optimisation. Alternatively, one might fit using the log likelihood, but compare models based on global metrics like the interspike interval. Notably, Prinz et al.[6] do not directly fit to potential traces $\tilde{V}$, but rather use a set of data-specific heuristics and global statistics to select candidate models. Gonşalves et al.[23] similarly find that working with specially crafted summary statistics–rather than the likelihoods–is more practical for inferring this type of model. In this work we nevertheless stick with the generic form of equation (9), which makes no assumptions on the type of data used to fit the models and is thus easier to generalise to different scenarios.

The definition of the risk then follows immediately:

$$R_a := \mathbb{E}_{(t, \tilde{V}^{\text{LP}}) \sim \mathcal{M}_{\text{true}}}\left[Q(t, \tilde{V}^{\text{LP}}; \mathcal{M}_a)\right]. \quad (10)$$

Here $a$ stands for one of the model labels *A*, *B*, *C* or *D*. The notation $(t, \tilde{V}^{\text{LP}}) \sim \mathcal{M}_{\text{true}}$ denotes that the distribution from which the samples $(t, \tilde{V}^{\text{LP}})$ are drawn is $\mathcal{M}_{\text{true}}$. We can think of $\mathcal{M}_{\text{true}}$ as producing an infinite sequence of data points by integrating the model dynamics and appplying equation (8) multiple times, or by going to the laboratory and performing the experiment multiple times.

As alluded to above, the candidate model traces in Fig. 2d suggest two groups of models: the $\mathcal{M}_A$ and $\mathcal{M}_B$ models seem to better reproduce the data than $\mathcal{M}_C$ or $\mathcal{M}_D$. Within each group however, it is hard to say whether one model is better than the other; in terms of the risks, this means that we expect $R_A, R_B < R_C, R_D$. This also means that we should be wary of a selection criterion which unequivocally ranks $\mathcal{M}_A$ better than $\mathcal{M}_B$, or $\mathcal{M}_C$ better than $\mathcal{M}_D$.

In other words, we expect that uncertainties on $R_A$ and $R_B$ should be at least commensurate with the difference $|R_A - R_B|$, and similarly for the uncertainties on $R_C$ and $R_D$.

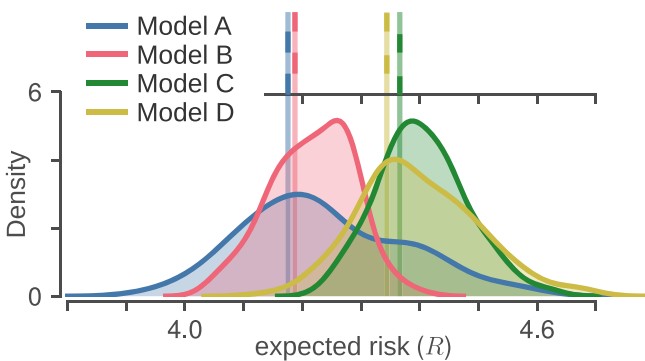

**Fig. 3 | Empirical risk vs. *R*-distributions for the four candidate LP models. (top)** The empirical risk(5) for each of the four candidate LP models. (bottom) Our proposed $B^{\text{EMD}}$ criterion replaces the risk by an *R-distribution*, where the spread of each distribution is due to the *replication uncertainty* for that particular model. *R*-distributions are distributions of the *R* functional in equation (20); we estimate them by sampling the quantile function (i.e., inverse cumulative density function) $q$ according to a stochastic process $\mathfrak{Q}$ on quantile functions. We used an EMD sensitivity factor of $c = 2^{-2}$ (see later section on calibration) for $\mathfrak{Q}$ and drew samples $\hat{q} \sim \mathfrak{Q}$ until the relative standard error on the risk was below 3 %. A kernel density estimate (KDE) is used to display those samples as distributions. The *R*-distribution for the true model is much narrower (approximately Dirac) and far to the left, outside the plotting bounds.

## A conceptual model for replicate variations

Evaluating the risk (10) for each candidate model on a dataset $\mathcal{D}_{\text{test}}$ yields four scalars $\hat{R}_A$ to $\hat{R}_D$; since a lower risk should indicate a better model, a simple naive model selection rule would be

$$\begin{cases} \text{reject } \mathcal{M}_b & \text{if } \hat{R}_a < \hat{R}_b , \\ \text{reject } \mathcal{M}_a & \text{if } \hat{R}_a > \hat{R}_b , \quad \text{for } a, b \in \{A, B, C, D\}. \\ \text{no rejection} & \text{if } \hat{R}_a = \hat{R}_b , \end{cases} \quad (11)$$

As with many selection criteria (e.g., AIC[33,34], BIC[35], DIC[36,37]), this rule is effectively binary: the third option, to reject neither model, has probability zero. It therefore always selects either $\mathcal{M}_a$ or $\mathcal{M}_b$, even when the evidence favouring one of the two is extremely weak. Another way to see this is illustrated at the top of Fig. 3: the lines representing the four values $R_A$ through $R_D$ have no error bars, so even minute differences suffice to rank the models.

The problem is that from a scientific standpoint, if the evidence is too weak, this ranking is likely irrelevant–or worse, misinformative. Hence our desire to assign uncertainty to each estimate of the risk. Ideally we would like to be able to compute tail probabilities like $P(R_A < R_B)$, which would quantify the strength of the evidence in favour of either $\mathcal{M}_A$ or $\mathcal{M}_B$. Selecting a minimum evidence threshold $\epsilon$ would then allow us to convert equation (11) into a true ternary decision rule with non-zero probability of keeping both models:

$$\begin{cases} \text{reject } \mathcal{M}_b & \text{if } P(R_a < R_b) > \epsilon, \\ \text{reject } \mathcal{M}_a & \text{if } P(R_a < R_b) < (1 - \epsilon), \\ \text{no rejection} & \text{if } (1 - \epsilon) \le P(R_a < R_b) \le \epsilon. \end{cases} \quad (12)$$

As we explain in the Introduction, our goal is to select a model which can describe not just the observed data $\mathcal{D}_{\text{test}}$, but also new data generated in a replication experiment. In order for our criterion to be robust, the tail probabilities $P(R_a < R_b)$ should account for uncertainty in the replication process.

Note that for a given candidate model $\mathcal{M}_a$ and fixed $\mathcal{M}_{\text{true}}$, we can always estimate the risk with high accuracy if we have enough samples,

irrespective of the amount of noise intrinsic to $\mathcal{M}_a$ or of misspecification between $\mathcal{M}_a$ and $\mathcal{M}_{\text{true}}$. Crucially however, we *also do not assume the replication process to be stationary*, so a replicate dataset $\mathcal{D}'_{\text{test}}$ may be drawn from a slightly different data-generating process $\mathcal{M}'_{\text{true}}$. This can occur even when $\mathcal{M}_{\text{true}}$ itself is stationary, reflecting the fact that it is often easier to control variability within a single experiments than across multiple ones. Ontologically therefore, the uncertainty on $\hat{R}$ is a form of epistemic uncertainty arising from the variability across (experimental) replications.

To make this idea concrete, consider that the input $I_{\text{ext}}$ and noise $\xi$ might not be stationary over the course of an experiment with multiple trials. We can represent this by making their parameters random variables, for example

$$\Omega := \begin{cases} \log \sigma_o & \sim \mathcal{N}\left(0.0\text{mV}, (0.5\text{mV})^2\right) \\ \log \sigma_i & \sim \mathcal{N}\left(-15.0\text{mV}, (0.5\text{mV})^2\right), \\ \log_{10} \tau & \sim \text{Unif}\left([0.1\text{ms}, 0.2\text{ms}]\right) \end{cases} \quad (13)$$

and drawing new values of $(\sigma_o, \sigma_i, \tau)$ for each trial (i.e., each replicate). Since it is a distribution over data-generating processes, we call $\Omega$ an *epistemic distribution*. For illustration purposes, here we have parametrised $\Omega$ in terms of two parameters of the biophysical model and one parameter of the observation model, thus capturing epistemic uncertainty within a single experiment. In general the parametrisation of $\Omega$ is a modelling choice, and may represent other forms of non-stationarity−for example due to variations between experimental setups in different laboratories.

Conceptually, we could estimate the tail probabilities $P(R_a < R_b)$ by sampling $J$ different data-generating processes $\mathcal{M}^j_{\text{true}}$ from $\Omega$, for each then drawing a dataset $\mathcal{D}^j_{\text{test}} \sim \mathcal{M}^j_{\text{true}}$, and finally computing the empirical risks $\hat{R}^j_a$ and $\hat{R}^j_b$ on $\mathcal{D}^j_{\text{test}}$. The fraction of datasets for which $\hat{R}^j_a < \hat{R}^j_b$ would then estimate the tail probability:

$$P(R_a < R_b) \approx \frac{1}{J} \left| \left\{ j \mid \hat{R}^j_a < \hat{R}^j_b \right\}^J_{j=1} \right|. \quad (14)$$

The issue of course is that we cannot know $\Omega$. First because we only observe data from a single $\mathcal{M}_{\text{true}}$, but also because there may be different contexts to which we want to generalise: one researcher may be interested in modelling an individual LP neuron, while another might seek a more general model which can describe all neurons of this type. These two situations would require different epistemic distributions, with the latter one being in some sense broader.

However we need not commit to a single epistemic distribution: if we have two distributions, and we want to ensure that conclusions hold under both, we can instead use the condition

$$\min_{\Omega \in \{\Omega_1, \Omega_2\}} P_\Omega\left(R_a < R_b\right) > \epsilon \quad (15)$$

to attempt to reject $\mathcal{M}_b$. In general, considering more epistemic distributions will make the model selection more robust, at the cost of discriminatory power. Epistemic distributions are not therefore prior distributions, since a Bayesian calculation is always tied to a specific choice of prior. (The opposite however holds: a prior can be viewed as a particular choice of epistemic distribution.)

We view epistemic distributions mostly as conceptual tools. For calculations, we will instead propose in the next sections a different type of distribution ($\mathfrak{Q}$; technically a stochastic process), which is not on the data-generating process, but on the distribution of pointwise losses. Being lower-dimensional and more stereotyped than $\Omega$, we will be able to construct $\mathfrak{Q}$ entirely non-parametrically, up to a scaling constant $c$ (c.f. Fig. 1 and the section listing desiderata for $\mathfrak{Q}$). A later section will then show, through numerical calibration and validation experiments, that $\mathfrak{Q}$ also has nice universal properties, so that the only

thing that really matters is the overall scale of the epistemic distributions. The constant $c$ is matched to this scale by numerically simulating epistemic distributions as part of the calibration experiments.

## Model discrepancy as a baseline for non-stationary replications

To keep the notation in the following sections more general, we use the generic $x$ and $y$ as independent and dependent variables. To recover expressions for our neuron example, substitute $x \to t, y \to \bar{V}^{\text{LP}}$, $\mathcal{X} \to \mathcal{T}$ and $\mathcal{Y} \to \mathcal{V}$. Where possible we also use $A$ and $B$ as generic placeholders for a model label.

Our goal is to define a selection criterion which is robust against variations between experimental replications, but which can be computed using knowledge only of the candidate models and the observed empirical data. To do this, we make the following assumption:

**EMD assumption (version 1)** *Candidate models represent that part of the experiment which we understand and control across replications.*

More precisely, in the next section we define the empirical model discrepancy function $\delta_A^{\text{EMD}} : (0, 1) \to \mathbb{R}_{\geq 0}$ such that if model $\mathcal{M}_A$ exactly reproduces the observations, then $\delta_A^{\text{EMD}}$ is identically zero. This function therefore measures the discrepancy between model predictions and actual observations. Since we expect misspecified models to have positive discrepancy ($\int_0^1 \delta_A^{\text{EMD}}(\Phi)d\Phi > 0$), in the following we treat the discrepancy $\delta_A^{\text{EMD}}$ as a measure of misspecification.

Under our EMD assumption, misspecification in a model corresponds to experimental conditions we don't fully control, and which could therefore vary across replications of the experiment. Concretely this means that we can reformulate the EMD assumption as

**EMD assumption (version 2)** *The variability of $R_A$ across replications is predicted by the model discrepancy $\delta_A^{\text{EMD}}$.*

We also assume that the data-generating process $\mathcal{M}_{\text{true}}$ within one replication is strictly stationary with finite correlations, i.e., that all observations are identically distributed but may be correlated. For simplicity in fact we treat the samples as i.i.d., since if necessary this could be done by thinning. See references 7 or 38 for discussions on constructing estimators from correlated time series.

Below we further assume a particular linear relationship between $\delta_A^{\text{EMD}}$ and the replication uncertainty: the function $\delta_A^{\text{EMD}}$ is scaled by the aforementioned sensitivity factor $c$ to determine the variance of the stochastic process $\mathfrak{Q}_A$ (which we recall induces the $R_A$-distribution for the risk of model $\mathcal{M}_A$). The parameter $c \in \mathbb{R}_+$ therefore represents a conversion from *model discrepancy* to *epistemic uncertainty*. A practitioner can use this parameter to adjust the sensitivity of the criterion to misspecification: a larger value of $c$ will emulate more important experimental variations. Since the tail probabilities (equation (14)) we want to compute will depend on $c$, we will write them

$$B_{AB;c}^{\text{EMD}} := P\left(R_A < R_B \mid c\right). \quad (16)$$

**The EMD rejection rule** *For a chosen rejection threshold $\epsilon \in (0.5, 1]$, reject model $\mathcal{M}_A$ if there exists a model $\mathcal{M}_B$ such that $B_{AB;c}^{\text{EMD}} < \epsilon$ and $\hat{R}_A > \hat{R}_B$.*

The second condition ($\hat{R}_A > \hat{R}_B$) ensures the rejection rule remains consistent, even if the $R$-distributions become skewed. (See comment below equation (5).)

As an illustration, Table 1 gives the value of $B^{\text{EMD}}$ for each candidate model pair in our example from Figs. 2 and 3. As expected, models that were visually assessed to be similar also have $B^{\text{EMD}}$ values close to $\frac{1}{2}$. In practice one would not necessarily need to compute the entire table,

**Table 1 | Comparison of $B^{\text{EMD}}$ probabilities for candidate LP models**

|   | A | B | C | D |
|---|---|---|---|---|
| A | 0.500 | 0.483 | 0.846 | 0.821 |
| B | 0.517 | 0.500 | 0.972 | 0.940 |
| C | 0.154 | 0.028 | 0.500 | 0.463 |
| D | 0.179 | 0.060 | 0.537 | 0.500 |

Values are the probabilities given by equation (16) with the candidate labels $a$ and $b$ corresponding to rows and columns respectively. Candidate models being compared are those of Fig. 2. Probabilities are computed for the $R$-distributions shown in Fig. 3, which used an EMD constant of $c = 2^{-2}$. Since $P(R_a < R_b) = 1 - P(R_b < R_a)$, the sum of symmetric entries equals 1.

since the $B^{\text{EMD}}$ satisfy *dice transitivity*[39,40]. In particular this implies that for any threshold $\epsilon > \varphi^{-2}$ (where $\varphi$ is the golden ratio), we have

$$\left. \begin{array}{ccc} B^{\text{EMD}}_{AB;c} & > & \sqrt{\epsilon} \\ B^{\text{EMD}}_{BC;c} & > & \sqrt{\epsilon} \end{array} \right\} \Rightarrow B^{\text{EMD}}_{AC;c} > \epsilon. \tag{17}$$

Since any reasonable choice of threshold will have $\epsilon > 0.5 > \varphi^{-2}$, we can say that whenever the comparisons $B^{\text{EMD}}_{AB;c}$ and $B^{\text{EMD}}_{BC;c}$ are both greater than $\sqrt{\epsilon}$, we can treat them as transitive. We give a more general form of this result in the Supplementary Methods.

**$\delta^{\text{EMD}}$: Expressing misspecification as a discrepancy between CDFs**
We can treat the loss function for a given model $\mathcal{M}_A$ as a random variable $Q(x, y; \mathcal{M}_A)$ where the $(x, y)$ are sampled from $\mathcal{M}_{\text{true}}$. A key realisation for our approach is that the CDF (cumulative distribution function) of the loss suffices to compute the risk. Indeed, we have for the CDF of the loss

$$\begin{aligned} \Phi^*_A(q) &:= p\big(Q(x, y; \mathcal{M}_A) \le q \mid x, y \sim \mathcal{M}_{\text{true}}\big) \\ &= \int_{\mathcal{X} \times \mathcal{Y}} H\big(q - Q(x, y; \mathcal{M}_A)\big)\, p\big(x, y \mid \mathcal{M}_{\text{true}}\big)\, dx\, dy \\ &\approx \frac{1}{L} \sum_{x_i, y_i \in \mathcal{D}_{\text{test}}} H\big(q - Q(x_i, y_i; \mathcal{M}_A)\big), \end{aligned} \tag{18}$$

where $H$ is the Heaviside function with $H(q) = 1$ if $q \ge 0$ and 0 otherwise.

Since $\mathcal{M}_{\text{true}}$ is unknown, a crucial feature of (18) is that $\Phi^*_A(q)$ can be estimated without needing to evaluate $p(x, y \mid \mathcal{M}_{\text{true}})$; indeed, all that is required is to count the number of observed data points in $\mathcal{D}_{\text{test}}$ which according to $\mathcal{M}_A$ have a loss less than $q$. Moreover, since $Q$ returns a scalar, the data points $(x_i, y_i)$ can have any number of dimensions: the number of data points required to get a good estimate of $\Phi^*_A(q)$ does not depend on the dimensionality of the data, for the same reason that estimating marginals of a distribution requires much fewer samples than estimating the full distribution. Finally, because the loss is evaluated using $\mathcal{M}_A$, but the expectation is taken with respect to a distribution determined by $\mathcal{M}_{\text{true}}$, we call $\Phi^*_A$ the *mixed CDF*.

We can invert $\Phi^*_A$ to obtain the *mixed PPF* (percent point function, also known as quantile function, or percentile function):

$$q^*_A(\Phi) := \Phi^{*-1}_A(\Phi) \tag{19, mixed PPF}$$

which is also a 1-d function, irrespective of the dimensionality of $\mathcal{X}$ or $\mathcal{Y}$. We can then rewrite the risk as a one dimensional integral in $q^*_A$:

$$\begin{aligned} R_A = R[q^*_A] &= \int_0^1 q^*_A(\Phi)\, d\Phi \\ &\approx \frac{1}{L} \sum_{x_i, y_i \in \mathcal{D}_{\text{test}}} Q(x_i, y_i; \mathcal{M}_A). \end{aligned} \tag{20}$$

To obtain equation (20), we simply used Fubini's theorem to reorder the integral of (18) and marginalised over all slices of a given loss $q^*_A(\Phi)$. The integral form (first line of equation (20)) is equivalent to averaging an infinite number of samples, and therefore to the (true) risk, whereas the average over observed samples (second line of equation (20)) is exactly the empirical risk defined in equation (5).

In practice, to evaluate equation (20), we use the observed samples to compute the sequence of per-sample losses $\{Q(x_i, y_i; \mathcal{M}_A)\}_{x_i, y_i \in \mathcal{D}_{\text{test}}}$. This provides us with a sequence of losses, which we use as ordinate values. We then sort this sequence so that we have $\{Q_i\}_{i=1}^L$ with $Q_i \le Q_i + 1$, and assign to each the abscissa $\Phi_i = i/L + 1$, such that losses are motonically increasing and uniformly distributed on the $[0, 1]$ interval. This yields the empirical PPF of the loss—the "empirical" qualifier referring to this construction via samples, as opposed to an analytic calculation. Interpolating the points then yields a continuous function which can be used in further calculations. All examples in this paper linearly interpolate the PPF from $2^{10} = 1024$ points.

In Fig. 4 we show four examples of empirical PPFs, along with their associated empirical CDFs. We see that the statistics of the additive observational noise affects the shape of the PPF: for noise with exponential tails, as we get from Gaussian or Poisson distributions, we have strong concentration around the minimum value of the loss followed by a sharp increase at $\Phi = 1$. For heavier-tailed distributions like Cauchy, loss values are less concentrated and the PPF assigns non-negligible probability mass to a wider range of values. The dimensionality of the data also matters. High-dimensional Gaussians are known to place most of their probability mass in a thin shell centred on the mode, and we see this in the fourth column of Fig. 4: the sharp increase at $\Phi = 0$ indicates that very low probability is assigned to the minimum loss.

Since by construction, the abscissae $\Phi$ of an empirical PPF are spaced at intervals of $1/L$, the Riemann sum for the integral in equation (20) reduces to the sample average. More importantly, we can interpret the risk as a functional in $q^*_A(\Phi)$, which will allow us below to define a generic stochastic process that accounts for epistemic uncertainty.

Up to this point with equation (20) we have simply rewritten the usual definition of the risk. Recall now that in the previous section, we proposed to equate replication uncertainty with misspecification; specifically we are interested in how differences between the candidate model $\mathcal{M}_A$ and the true data-generating process $\mathcal{M}_{\text{true}}$ affect the loss PPF, since this determines the risk. Therefore we also compute the PPF of $Q(x, y; \mathcal{M}_A)$ under its own model (recall from equation (1) that $\mathcal{M}_A$ must be a probabilistic model):

$$\begin{aligned} \widetilde{\Phi}_A(q) &:= p\big(Q(x, y; \mathcal{M}_A) \le q \mid x, y \sim \mathcal{M}_A\big) \\ &= \int_{\mathcal{X} \times \mathcal{Y}} dx\, dy\, H\big(q - Q(x, y; \mathcal{M}_A)\big)\, p\big(x, y \mid \mathcal{M}_A\big) \\ &\approx \frac{1}{L_{\text{synth}, A}} \sum_{x_i, y_i \in \mathcal{D}_{\text{synth}, A}} H\big(q - Q(x_i, y_i; \mathcal{M}_A)\big), \end{aligned} \tag{21}$$

from which we obtain the *synthetic PPF*:

$$\tilde{q}_A(\Phi) := \widetilde{\Phi}_A^{-1}(\Phi). \tag{22, synth PPF}$$

The only difference between $\tilde{q}_A$ and $q^*_A$ is the use of $p(x, y \mid \mathcal{M}_A)$ instead of $p(x, y \mid \mathcal{M}_{\text{true}})$ in the integral. In practice this integral would also be evaluated by sampling, using $\mathcal{M}_A$ to generate a dataset $\mathcal{D}_{\text{synth}, A}$ with $L_{\text{synth}, A}$ samples. Because in this case the candidate model is used for both generating samples and defining the loss, we call $\tilde{q}_A(\widetilde{\Phi}_A)$ the *synthetic PPF (CDF)*.

The idea is that the closer $\mathcal{M}_A$ is to $\mathcal{M}_{\text{true}}$, the closer also the synthetic PPF should be to the mixed PPF—indeed, equality of the PPFs ($\tilde{q}_a = q^*_A$) is a necessary condition for equality of the models

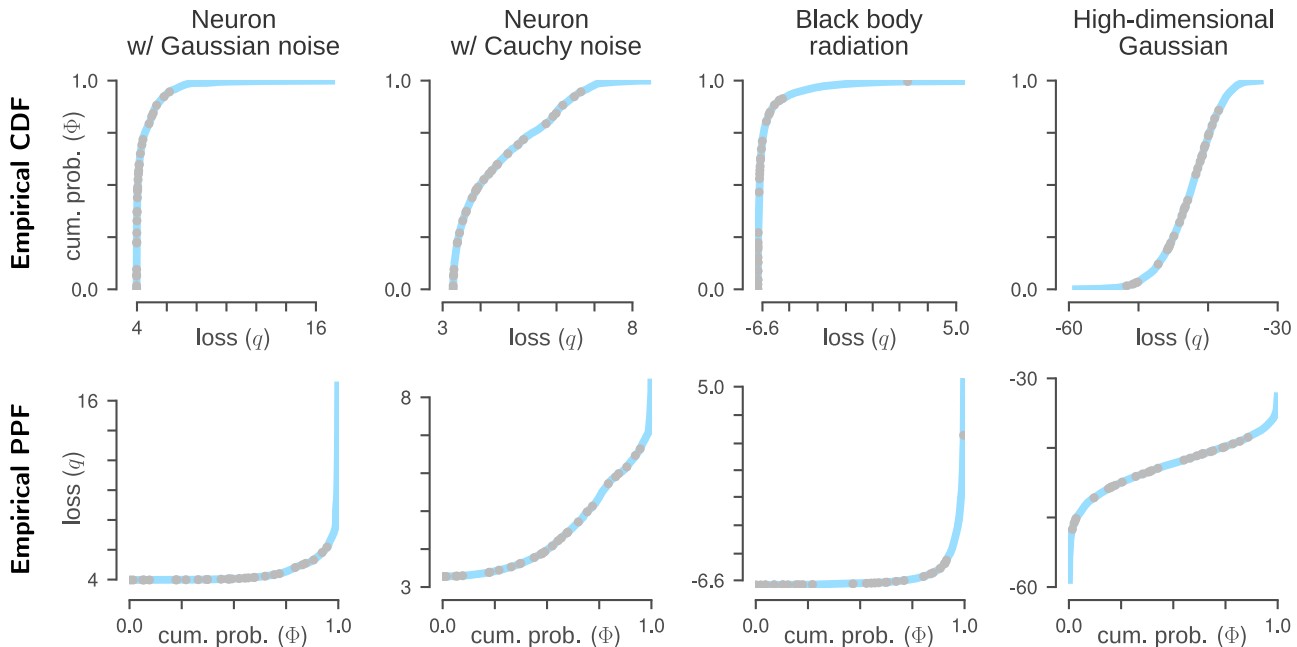

**Fig. 4 | Loss PPF for different models.** Each column corresponds to a different model. The PPF (percent point function; bottom row) is the inverse of the CDF (cumulative density function; top row). For calculations we interpolate $2^{10} = 1024$ points (cyan line) to obtain a smooth function; for illustration purposes here only 30 points are shown. The data for the first two columns were generated with the neuron model described at the top of our Results, where the additive noise follows either a Gaussian or Cauchy distribution. The black body radiation data for the third column were generated from a Poisson distribution using equation (38) with $s = 2^{14}$ and $\lambda$ in the range $6\mu m$ to $20\mu m$. Here the true noise is binomial, but the loss assumes a Gaussian. The fourth column shows an example where the data are high-dimensional; the same 30 dimensional, unit variance, isotropic Gaussian is used for both generating the data and evaluating the loss. In all panels the loss function used is the log likelihood under the model.

($\mathcal{M}_A = \mathcal{M}_{\text{true}}$). Therefore we can quantify the uncertainty due to mis-specification, at least insofar as it affects the risk, as the absolute difference between $\tilde{q}_A$ and $q_A^*$:

$$
\begin{aligned}
\delta_A^{\text{EMD}} &: [0, 1] \to \mathbb{R} \\
\Phi &\mapsto \left| \tilde{q}_A(\Phi) - q_A^*(\Phi) \right|.
\end{aligned} \tag{23}
$$

We refer to $\delta_A^{\text{EMD}}$ as the *empirical model discrepancy* (EMD) function because it measures the discrepancy between two empirical PPFs.

It is worth noting that even a highly stochastic data-generating process $\mathcal{M}_{\text{true}}$, with a lot of aleatoric uncertainty, still has a well defined PPF—which would be matched by an equally stochastic candidate model $\mathcal{M}_A$. Therefore the discrepancies measured by $\delta_A^{\text{EMD}}$ are a representation strictly of the *epistemic* uncertainty. These discrepancies can arise either from a mismatch between $\mathcal{M}_A$ and $\mathcal{M}_{\text{true}}$, or simply having too few samples to estimate the mixed PPF $q_A^*$ exactly; either mechanism contributes to the uncertainty on the expected risk of replicate datasets. We illustrate some differences between aleatoric and epistemic uncertainty in Supplementary Fig. 3, and include further comments on how different types of uncertainties relate to risk in the Supplementary Discussion.

### $\mathfrak{Q}$: a stochastic process on quantile functions

When replication is non-stationary, each replicate dataset will produce a different loss PPF. A distribution over replications (like the previously defined epistemic distribution $\Omega$) therefore corresponds to a distribution over PPFs. This leads us to the final formulation of our EMD assumption, which we now operationalise by assuming a linear relation between empirical model discrepancy and epistemic uncertainty:

**EMD principle** *For a given model $\mathcal{M}_A$, differences between its PPFs on two replicate datasets should be proportional to $\delta_A^{\text{EMD}}$.*

Formalising this idea is made considerably simpler by the fact that PPFs are always scalar functions, irrespective of the model or dataset's dimensionality. We do this as follows, going through the steps schematically represented by downward facing arrows on the right of Fig. 1:

For a given model $\mathcal{M}_A$, we treat the PPFs of different replicates as realisations of a stochastic process $\mathfrak{Q}_A$ on the interval $[0, 1]$; a realisation of $\mathfrak{Q}_A$ (i.e., a PPF) is a function $\hat{q}_A(\Phi)$ with $\Phi \in [0, 1]$. The stochastic process $\mathfrak{Q}_A$ is defined to satisfy the desiderata listed below, which ensure that it is centred on the observed PPF $q_A^*$ and that its variance at each $\Phi$ is governed by $\delta_A^{\text{EMD}}$.

Obtaining the *R*-distributions shown in Fig. 3 is then relatively straightforward: we simply sample an ensemble of $\hat{q}_A$ from $\mathfrak{Q}_A$ and integrate each to obtain an ensemble of risks.

Working with PPFs has the important advantage that we can express our fundamental assumption as a simple linear proportionality relation between empirical model discrepancy and epistemic uncertainty. While this is certainly an approximation, it simplifies the interpretability of the resulting *R*-distribution. As we show in the Supplementary Discussion, the resulting criterion is also more robust than other alternatives. The simplicity of the assumption however belies the complexity of defining a stochastic process $\mathfrak{Q}$ directly on PPFs.

We will now do this in three steps: First we establish a set of *desiderata* for $\mathfrak{Q}$. We then sketch the construction of the *hierarchical beta (HB) process* we propose to satisfy these desiderata. Finally we show how such a process can be used to compare models.

In order to interpret them as such, realisations of $\hat{q}_A \sim \mathfrak{Q}_A$ must valid PPFs. This places quite strong constraints on those realisations, for example:

- All realisations $\hat{q}_A(\Phi) \sim \mathfrak{Q}_A$ must be *monotone*.
- All realisations $\hat{q}_A(\Phi) \sim \mathfrak{Q}_A$ must be *integrable*.

Monotonicity follows immediately from definitions: a CDF is always monotone because it is the integral of a positive function equation (18), and therefore its inverse must also be monotone.

Integrability simply means that the integral in equation (20) exists and is finite. Concretely this is enforced by ensuring that the process $\mathfrak{Q}_A$ is *self-consistent*[41], a property which we explain in the Methods.

Interpreting the realisations $\hat{q}$ as PPFs also imposes a third constraint, more subtle but equally important:

- The process $\mathfrak{Q}_A$ must be *non-accumulating*.

A process which is accumulating would start at one end of the domain, say $\Phi = 0$, and sequentially accumulate increments until it reaches the other end. Brownian motion over the interval $[0, T]$ is an example of such a process. In contrast, consider the process of constructing a PPF for the data in Fig. 2: initially we have few data points and the PPF of their loss is very coarse. As the number of points increases, the PPF gets refined, but since loss values occur in no particular order, this happens simultaneously across the entire interval.

The accumulation of increments strongly influences the statistics of a process; most notably, the variance is usually larger further along the domain. This would not make sense for a PPF: if $\mathbb{V}[\mathfrak{Q}_A(\Phi)]$ is smaller than $\mathbb{V}[\mathfrak{Q}_A(\Phi')]$, that should be a consequence of $\delta^{\text{EMD}}(\Phi)$ being smaller than $\delta^{\text{EMD}}(\Phi')$—not of $\Phi$ occurring "before" $\Phi'$.

This idea that a realisation $\hat{q}_A(\Phi)$ is generated simultaneously across the interval led us to define $\mathfrak{Q}_A$ as a sequence of refinements: starting from an initial increment $\Delta\hat{q}_1(0) = \hat{q}_A(1) - \hat{q}_A(0)$ for the entire $\Phi$ interval $[0, 1]$, we partition $[0, 1]$ into $n$ subintervals, and sample a set of $n$ subincrements in such a way that they sum to $\Delta\hat{q}_1(0)$. This type of distribution, where $n$ random variables are drawn under the constraint of a fixed sum, is called a *compositional* distribution[42]. Note that the constraint reduces the number of dimensions by one, so a pair of increments would be drawn from a 1-d compositional distribution. A typical 1-d example is the beta distribution for $x_1 \in [0, 1]$, with $x_2 = (1 - x_1)$ and $\alpha, \beta > 0$:

$$\text{if } x_1 \sim \text{Beta}(\alpha, \beta), \quad \text{then} \quad p(x_1) \propto x_1^{\alpha-1}(1 - x_1)^{\beta-1}. \quad (24)$$

Interestingly, the most natural statistics for compositional distributions are not the mean and variance, but analogue notions of *centre* and *metric variance*[42,43]; for the beta distribution defined above, these are

$$\mathbb{E}_a\left[(x_1, x_2)\right] = \frac{1}{e^{\psi(\alpha)} + e^{\psi(\beta)}}\left(e^{\psi(\alpha)}, e^{\psi(\beta)}\right) \quad (25a)$$

$$\text{Mvar}\left[(x_1, x_2)\right] = \frac{1}{2}\left(\psi_1(\alpha) + \psi_1(\beta)\right), \quad (25b)$$

where $\psi$ and $\psi_1$ are the digamma and trigamma functions respectively, and $\mathbb{E}_a$ denotes expectation with respect to the Aitchison measure[42,44]. In essence, equations 25 are obtained by mapping $x_1$ and $x_2$ to the unbounded domain $\mathbb{R}$ via a logistic transformation, then evaluating moments of the unbounded variables. Of particular relevance is that—in contrast to the variance—the metric variance Mvar of a compositional distribution is therefore unbounded, which simplifies the selection of $\alpha$ and $\beta$ (see Choosing beta distribution parameters in the Methods).

Of course, we not only want the $\hat{q}_A$ to be valid PPFs, but also descriptive of the model and data. We express this as two additional constraints, which together define a notion of closeness to $q_A^*$:

- At each intermediate point $\Phi \in (0, 1)$, the *centre* of the process is given by $q_A^*(\Phi)$:

$$\mathbb{E}_a[\hat{q}(\Phi)] = q_A^*(\Phi). \quad (26)$$

- At each intermediate point $\Phi \in (0, 1)$, the *metric variance* is proportional to the square of $\delta_A^{\text{EMD}}(\Phi)$:

$$\text{Mvar}[\hat{q}(\Phi)] = c\,\delta_A^{\text{EMD}}(\Phi)^2. \quad (27)$$

Note that (27) formalises the EMD principle stated at the top of this section, with the sensitivity factor $c > 0$ determining the conversion from *empirical model discrepancy* (which we interpret as misspecification) to *epistemic uncertainty*. Note also that equations (26) and (27) are akin to a variational approximation of the stochastic process around $q_A^*$. Correspondingly, we should expect these equations (and therefore $\mathfrak{Q}$) to work best when $c$ is not too large. We describe how one might determine an appropriate value for $c$ in the later section on calibration.

In addition to the above desiderata, for reasons of convenience, we also ask that

- The end points are sampled from Gaussian distributions:

$$\hat{q}(0) \sim \mathcal{N}(q_A^*(0), c\,\delta_A^{\text{EMD}}(0)^2),$$
$$\hat{q}(1) \sim \mathcal{N}(q_A^*(1), c\,\delta_A^{\text{EMD}}(1)^2). \quad (28)$$

Thus the process $\mathfrak{Q}_{A;c}$ is parametrised by two functions and a scalar: $q_A^*$, $\delta_A^{\text{EMD}}$ and $c$. It is moulded to produce realisations $\hat{q}$ which as a whole track $q_A^*$, with more variability between realisations at points $\Phi$ where $\delta_A^{\text{EMD}}(\Phi)$ is larger (middle panel of Fig. 5a).

To the best of our knowledge the current literature does not provide a process satisfying all of these constraints. To remedy this situation, we propose a new *hierarchical beta (HB) process*, which we illustrate in Fig. 5. A few example realisations of $\hat{q} \sim \mathfrak{Q}_A$ are drawn as grey lines in Fig. 5a. The mixed PPF $q_A^*$ (equation 26) is drawn as a green line, while the region corresponding to $q_A^*(\Phi) \pm \sqrt{c}\delta_A^{\text{EMD}}(\Phi)$ is shaded in yellow. The process is well-defined for $c \in [0, \infty]$: the $c \to 0$ limit is simply $q_A^*$, while the $c \to \infty$ limit is a process which samples uniformly at every step, irrespective of $q_A^*$ and $\delta_A^{\text{EMD}}$. In other words, as $c$ increases, the ability of each realisation $\hat{q}_A$ to track $q_A^*$ is limited by the constraints of monotonicity and non-accumulating increments. This translates to step-like PPFs, as seen in the lower panel of Fig. 5a. Figure 5b shows distributions of $\hat{q}(\Phi)$ at three different values of $\Phi$. The value of $q_A^*(\Phi)$ is indicated by the green vertical bar and agrees well with $\mathbb{E}_a[\hat{q}(\Phi)]$; the desideratum of (26) is therefore satisfied. The scaling of these distributions with $\delta_A^{\text{EMD}}$ equation (27) is however only approximate, which we can see as the yellow shading not having the same width in each panel. This is a result of the tension with the other constraints: the HB process ensures that the monotonicity and integrability constraints are satisfied exactly, but allows deviations in the statistical constraints.

A realisation of an HB process is obtained by a sequence of refinements; we illustrate three such refinements steps in Fig. 5d. The basic idea is to refine an increment $\Delta\hat{q}_{\Delta\Phi}(\Phi) := \hat{q}(\Phi + \Delta\Phi) - \hat{q}(\Phi)$ into two subincrements $\Delta\hat{q}_{\frac{\Delta\Phi}{2}}(\Phi)$ and $\Delta\hat{q}_{\frac{\Delta\Phi}{2}}(\Phi + \frac{\Delta\Phi}{2})$, with $\Delta\hat{q}_{\frac{\Delta\Phi}{2}}(\Phi) + \Delta\hat{q}_{\frac{\Delta\Phi}{2}}(\Phi + \frac{\Delta\Phi}{2}) = \Delta\hat{q}_{\Delta\Phi}(\Phi)$. To do this, we first determine appropriate parameters $\alpha$ and $\beta$, draw $x_1$ from the corresponding beta distribution and assign

$$\Delta\hat{q}_{\frac{\Delta\Phi}{2}}(\Phi) \leftarrow x_1\Delta\hat{q}_{\Delta\Phi}(\Phi),$$
$$\Delta\hat{q}_{\frac{\Delta\Phi}{2}}\left(\Phi + \frac{\Delta\Phi}{2}\right) \leftarrow x_2\Delta\hat{q}_{\Delta\Phi}(\Phi), \quad (29)$$

where again $x_2 = (1 - x_1)$. Figure 5c shows distributions for the first (orange) and second (blue) subincrements at the fourth refinement step, where we divide an increment over an interval of length $\Delta\Phi = 2^{-3}$ to two subincrements over intervals of length $2^{-4}$. Each pair of subincrements is drawn for a different distribution, which depends on the particular realisation $\hat{q}$, but we can nevertheless see the PPF reflected in the aggregate distribution: the PPF has positive curvature, so the

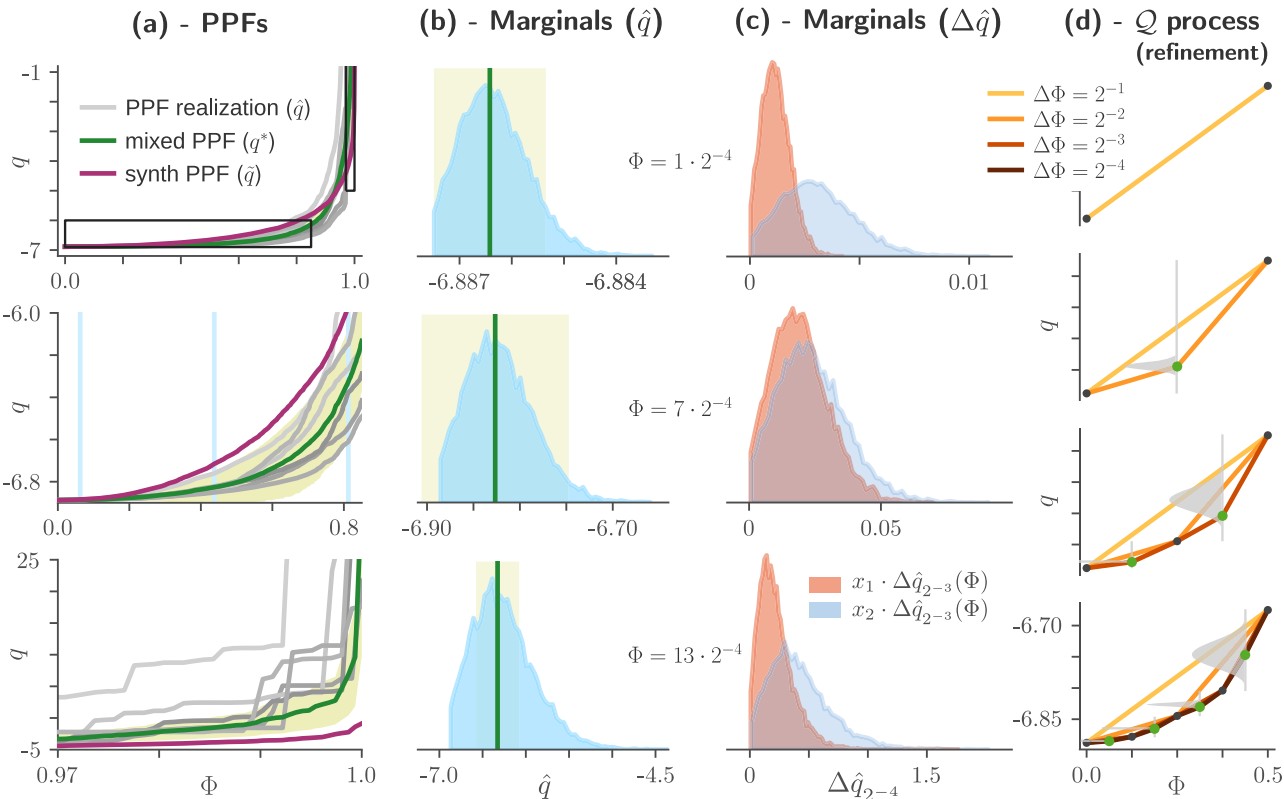

**Fig. 5 | Sampling a hierarchical beta (HB) process. a** PPF samples (grey lines) drawn from a hierarchical beta process $\mathfrak{Q}$, relating the pointwise loss $q$ to the cumulative probability $\Phi$. Mixed (green) and synthetic (red) PPFs are those for the Planck model of Fig. 7f (respectively $q^*$ from equation (19) and $\tilde{q}$ from equation (22)). Lower panels are enlarged portions of the top panel, corresponding to the black rectangles in the latter. At each $\Phi$, the variance between realisations is controlled by $\sqrt{c}\,\delta^{\mathrm{EMD}}$ (yellow shading), per equation (27); here we use $c = 0.5$. **b** Marginals of $\mathfrak{Q}$ at three values of $\Phi$, obtained as histograms of 10,000 realisations $\hat{q}$. As in (a), the green line indicates the value of $q^*(\Phi)$ and the yellow shading describes the range $q^*(\Phi) \pm \sqrt{c}\,\delta^{\mathrm{EMD}}(\Phi)$. Values of $\Phi$ are given alongside the panels and drawn as cyan vertical lines in (a). **c** Distributions of the subincrements drawn at the same

three $\Phi$ positions as in (b), obtained as histograms from the same 10,000 realisations. Subincrements are for the fourth refinement step, corresponding to the fourth panel of (d). Notation in the legend corresponds to equation (29).
**d** Illustration of the refinement process. This $\mathfrak{Q}$ is parametrised by the same PPFs as the one in (**a**–**c**), but we used a larger $c$ (16) to facilitate visualisation. Each refinement step halves the width $\Delta\Phi$ of an increment; here we show four refinements steps, while in most calculations we use eight. New points at each step are coloured green; the beta distribution from which each new point is drawn is shown in grey. The domain of each of these distributions spans the vertical distance between the two neighbouring points and is shown as a grey vertical line.

second subincrement tends to be larger than the first. Also both increments are bounded from below by 0, to ensure monotonicity.

A complete description of the HB process, including a procedure for choosing the beta parameters $\alpha$ and $\beta$ such that our desiderata are satisfied, is given in the Methods.

Having constructed processes $\mathfrak{Q}_{A;c}$ for each candidate model $\mathcal{M}_A$, we can use it to induce the $R$-distributions which serve to compare them. We do this by generating a sequence of PPFs $\hat{q}_{A,1}, \hat{q}_{A,2}, \ldots, \hat{q}_{A,M_A}$, where $M_A \in \mathbb{N}$ and each $\hat{q}_{A,i}$ is drawn from $\mathfrak{Q}_{A;c}$ (see Fig. 5 for examples of sampled $\hat{q}_{A,i}$, and the Methods for more details on how we evaluate $B^{\mathrm{EMD}}$). As we explain in the next section, the sensitivity parameter $c$ is a property of the experiment; it is the same for all candidate models.

For each generated PPF, we evaluate the risk functional (using the integral form of equation (20)), thus obtaining a sequence of scalars $R[\hat{q}_{A,1}], R[\hat{q}_{A,2}], \ldots, R[\hat{q}_{A,M_A}]$ which follows $p(R_A|\mathfrak{Q}_A)$. With $M_A$ sufficiently large, these samples accurately characterise the distribution $p(R_A \mid \mathfrak{Q}_A)$ (we use $\leftrightarrow$ to relate equivalent descriptions):

$$
\begin{aligned}
R_A &\sim p(R_A \mid \mathcal{D}_{\mathrm{test}}, \mathcal{M}_A; c) \\
&\leftrightarrow \left\{ R[\hat{q}_{A,1}], R[\hat{q}_{A,2}], \ldots, R[\hat{q}_{A,M_A}] \mid \hat{q}_A \sim \mathfrak{Q}_A \right\} \\
&\leftrightarrow \left\{ R_{A,1}, R_{A,2}, \ldots, R_{A,M_A} \right\}.
\end{aligned} \tag{30}
$$

Repeating this procedure for a different model $\mathcal{M}_B$ yields a different distribution for the risk:

$$
R_B \sim p(R_B \mid \mathcal{D}_{\mathrm{test}}, \mathcal{M}_B; c) \leftrightarrow \left\{ R_{B,1}, R_{B,2}, \ldots, R_{B,M_B} \right\}. \tag{31}
$$

The $B_{AB;c}^{\mathrm{EMD}}$ criterion (16) then reduces to a double sum:

$$
\begin{aligned}
B_{AB;c}^{\mathrm{EMD}} &:= P(R_A < R_B \mid c) \\
&\approx \frac{1}{M_A M_B} \sum_{i=1}^{M_A} \sum_{j=1}^{M_B} \mathbf{1}_{R_{A,i} < R_{B,j}}.
\end{aligned} \tag{32}
$$

In equation (32), the term within the sum is one when $R_{A,i} < R_{B,j}$ and zero otherwise. A value of $B_{AB;c}^{\mathrm{EMD}}$ greater (lesser) than 0.5 indicates evidence for (against) model $\mathcal{M}_A$.

The undetermined parameter $c$ (which converts discrepancy into epistemic uncertainty; c.f. (27)) can be viewed as a way to adjust the sensitivity of the criterion: larger values of $c$ will typically lead to broader distributions of $R_A$, and therefore lead to a more conservative criterion (i.e., one which is more likely to result in an equivocal outcome). We give some guidelines on choosing $c$ in the next section.

## Calibrating and validating the $B^{\mathrm{EMD}}$

The process $\mathfrak{Q}$ constructed in the previous section succeeds in producing distributions of the risk. And at least with some values of the scaling constant $c$, the distributions look as we expect (Fig. 3) and define a $B^{\mathrm{EMD}}$ criterion which assigns reasonable tail probabilities (Table 1).

To now put the $B^{\mathrm{EMD}}$ criterion on firmer footing, recall that it is meant to model variations between data replications. We can therefore validate the criterion by simulating such variations in silico (in effect simulating many replications of the experiment): since the true data-generating process $\mathcal{M}_{\mathrm{true}}$ is then known, this gives us a ground truth of known comparison outcomes, against which we can test the probabilistic prediction made by $B^{\mathrm{EMD}}$. Because this procedure will also serve to select a suitable value for the scaling constant $c$, we call it a *calibration experiment.*

To generate the required variations we use epistemic distributions of the type introduced at the top of our Results, of which we already gave an example in equation (13). With this we define an alternative tail probability (see the Methods for details),

$$B^{\mathrm{epis}}_{AB;\Omega} := P(R_A < R_B \mid \Omega),\qquad(33)$$

which uses the epistemic distribution $\Omega$ instead of the process $\mathfrak{Q}_A$ to represent epistemic uncertainty. We then look for $c > 0$ such that the criterion $B^{\mathrm{EMD}}_{AB;c}$ a) is correlated with $B^{\mathrm{epis}}_{AB;\Omega}$; and b) satisfies

$$\left| B^{\mathrm{EMD}}_{AB;c} - 0.5 \right| \lesssim \left| B^{\mathrm{epis}}_{AB;\Omega} - 0.5 \right|.\qquad(34)$$

Equation (34) encodes a desire for a conservative criterion: we are most worried about incorrectly rejecting a model, i.e., of making a type I error. The consequences for a type II error (failing to reject either model) are in general less dire: it is an indication that we need better data to differentiate between models. This desire for a conservative criterion reflects a broad belief in science that it is better to overestimate errors than underestimate them. Given the ontological differences between $B^{\mathrm{EMD}}$ and $B^{\mathrm{epis}}$ however, in practice one will need to allow at least small violations, which is why we give the relation in equation (34) as $\lesssim$.

The advantage of a probability like $B^{\mathrm{epis}}_{AB;\Omega}$ is that it makes explicit which kinds of replication variations are involved in computing the tail probability; the interpretation of equation (33) is thus clear. However it can only be computed after generating a large number of synthetic datasets, which requires a known and parametrisable data-generating process $\mathcal{M}_{\mathrm{true}}$; on its own therefore, $B^{\mathrm{epis}}_{AB;\Omega}$ cannot be used directly as a criterion for comparing models. *By choosing $c$ such that the criteria are correlated, we transfer the interpretability of $B^{\mathrm{epis}}_{AB;\Omega}$ onto $B^{\mathrm{EMD}}_{AB;c}$.* Moreover, if we can find such a $c$, then we have de facto validated $B^{\mathrm{EMD}}_{AB;c}$ for the set of simulated experiments described by $\Omega$.

Since defining an epistemic distribution involves making many arbitrary choices, one may want to define multiple distributions $\Omega_1, \Omega_2, \ldots$ to ensure that results are not sensitive to a particular choice of $\Omega$. Equation (34) can easily be generalised to account for this, following a similar logic as equation (15), in which case it becomes

$$\left| B^{\mathrm{EMD}}_{AB;c} - 0.5 \right| \lesssim \min_{\Omega \in \{\Omega_1, \Omega_2, \ldots\}} \left| B^{\mathrm{epis}}_{AB;\Omega} - 0.5 \right|.\qquad(35)$$

We found that an effective way to verify equation (34) is by plotting $B^{\mathrm{epis}}_{AB;\Omega}$ against $B^{\mathrm{EMD}}_{AB;c}$, where values of $B^{\mathrm{epis}}_{AB;\Omega}$ are obtained by averaging comparison outcomes, conditioned on the value of $B^{\mathrm{EMD}}_{AB;c}$ being within an interval (this is explained more precisely in the Methods). We thus obtain a histogram of $B^{\mathrm{epis}}_{AB;\Omega}$ against $B^{\mathrm{EMD}}_{AB;c}$, which works best when the size of bins is adjusted so that they have similar statistical power. We illustrate this in Fig. 6, where histograms are shown as curves to facilitate interpretation. These curves are drawn

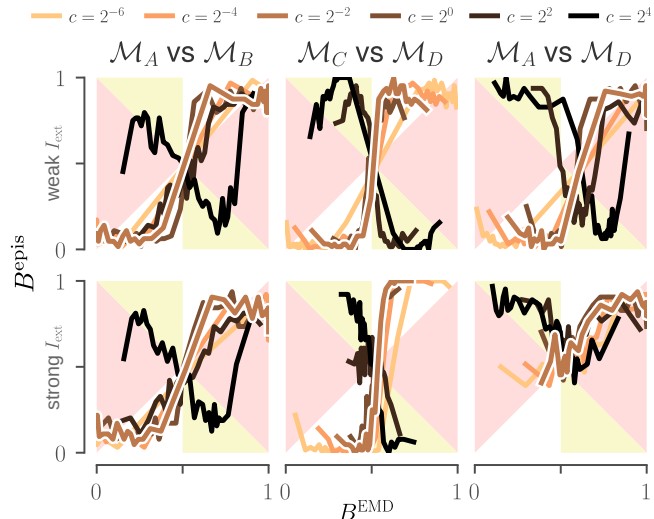

**Fig. 6 | Calibration curves for the LP models of Fig. 2.** Calibration curves for six epistemic distributions, computed following our proposed calibration procedure. Each curve summarises 2048 simulated experiments with datasets of size 4000, and the six panels explore how curves depend on three parameters: the pair of models being compared, the constant $c$ and the input strength $I_{\mathrm{ext}}$. The latter is either weak **(top)** or strong **(bottom)**. Other parameters are kept fixed, namely a short correlation time $\tau$ and a Gaussian observation noise with low standard deviation. The regions depicted in red and yellow are those where equation (34) is violated. (For an extended version where all conditions are tested, see Supplementary Fig. 1. For example $R$-distributions for each of these $c$ values, see Supplementary Fig. 2).

against the "overconfident regions" (where equation (34) is violated), depicted in red or yellow: we look therefore for values of $c$ which as much as possible stay within the white regions. The visual representation makes it easier to judge the extent to which small violations of equation (34) can be tolerated. An extended version of this figure, where all 48 conditions are tested, is provided as Supplementary Fig. 1.

Figure 6 shows calibration curves for six different epistemic distributions: three model pairs ($\mathcal{M}_A$ vs $\mathcal{M}_B$, $\mathcal{M}_C$ vs $\mathcal{M}_D$, and $\mathcal{M}_A$ vs $\mathcal{M}_D$), each with weak and strong external input $I_{\mathrm{ext}}$. (For full details on the choice of epistemic distributions for these experiments, see the Methods.) The model pairs were chosen to test three different situations: one where the candidate models are similar both to each other and the observations ($\mathcal{M}_A$ vs $\mathcal{M}_B$), one where the candidate models are similar to each other but different from the observations ($\mathcal{M}_C$ vs $\mathcal{M}_D$), and one where only one of the candidates is similar to the observations ($\mathcal{M}_A$ vs $\mathcal{M}_D$).

As one might expect, we see some violations of equation (35) in Fig. 6, which can partly be explained by our choice of a pointwise loss: although pedagogical, it tends to prefer models which produce fewer spikes, as we explain in the Methods. This is partly why, for these types of models, practitioners often prefer loss functions based on domain-specific features like the number of spikes or the presence of bursts[23]. We are also less concerned with the calibration of $\mathcal{M}_A$ vs $\mathcal{M}_D$, since those models are very different (c.f. Fig. 2). Not only is the $B^{\mathrm{EMD}}$ not needed to reject $\mathcal{M}_D$ in favour of $\mathcal{M}_A$, but the variational approximation expressed by (27) works best when the discrepancy between the model and the observed data (i.e., $\delta^{\mathrm{EMD}}$) is not too large.

Nevertheless, for values of $c$ between $2^{-4}$ and $2^0$, we see that calibration curves largely avoid the overconfidence (red and yellow) regions under most conditions and that $B^{\mathrm{epis}}$ is strongly correlated with $B^{\mathrm{EMD}}$. This confirms that $B^{\mathrm{EMD}}_{ab;c}$ can be an estimator for the probability $P(R_a < R_b)$ (recall equations (16) and (33)) and validates our choice of $c = 2^{-2}$ for the $R$-distributions in Fig. 2. More importantly, it shows that $c$ does not need to be tuned to a specific value for $B^{\mathrm{EMD}}$ to be

interpretable: it suffices for $c$ to be within a finite range. We observe a similar region of validity for $c$ with the model introduced in the next section (see Fig. 9 in the Methods). The fact that $B^{\mathrm{EMD}}$ remains valid for a range of $c$ values is a major reason why we think it can be useful for analysing real data, where we don't know the epistemic distribution, as well as for comparing multiple models simultaneously.

Within its range of validity, the effect of $c$ is somewhat analogous to a confidence level: just like different confidence levels will lead to different confidence intervals, different values of $c$ can lead to different (but equally valid) values of $B_{ab;c}^{\mathrm{EMD}}$. See the Supplementary Discussion for more details. Note also that due to the constraints of the $\mathfrak{Q}$ process, increasing $c$ does not simply increase the spread of $R$-distributions, but can also affect their shape; this is illustrated in Supplementary Fig. 2.

There are limits however to the range of validity. Too small values of $c$ will remove any overlap between $R$-distributions and produce an overconfident $B^{\mathrm{EMD}}$. Too large values of $c$ will exaggerate overlaps, underestimating statistical power. Large values can also lead to distortions in the $\mathfrak{Q}$ process, due to the monotonicity constraint placing an upper bound on the achievable metric variance; we see this as the curves reversing in Fig. 6 for $c \geq 2^2$. In the Supplementary Discussion we list some possible approaches to enlarge the range of validity, or otherwise improve the calibration of the $B^{\mathrm{EMD}}$.

## Characterising the behaviour of $R$-distributions

To better anchor the interpretability of $R$-distributions, in this section we perform a more systematic study of the relationship between epistemic uncertainty, aleatoric uncertainty, and the shape of the $R$-distributions. To do this we use a different example, chosen for its illustrative simplicity, which allows us to independently adjust the ambiguity (how much two models are qualitatively similar) and the level of observation noise.

Concretely, we imagine a fictitious historical scenario where the Rayleigh-Jeans

$$\mathcal{B}_{\mathrm{RJ}}(\lambda; T) = \frac{2ck_B T}{\lambda^4} \qquad (36)$$

and Planck

$$\mathcal{B}_{\mathrm{P}}(\lambda; T) = \frac{2hc^2}{\lambda^5} \frac{1}{\exp\left(\frac{hc}{\lambda k_B T}\right) - 1} \qquad (37)$$

models for the radiance of a black body are two candidate models given equal weight in the scientific community. They stem from different theories of statistical physics, but both agree with observations at infrared or longer wavelengths, and so both are plausible if observations are limited to that window. (When it is extended to shorter wavelengths, the predictions diverge and it becomes clear that the Planck model is the correct one.)

For our purposes, these are just two models describing the relationship between an independent variable $\lambda$ (the wavelength) and a dependent variable $\mathcal{B}$ (the spectral radiance), given a parameter $T$ (the temperature) which is inferred from data; our discussion is agnostic to the underlying physics. The parameters $h$, $c$ and $k_B$ are known physical constants (the Planck constant, the speed of light and the Boltzmann constant) and can be omitted from the discussion.

We use a simple Poisson counting process to model the data-generating model $\mathcal{M}_{\mathrm{true}}$ including the observation noise:

$$\mathcal{B} \mid \lambda, T, s \sim \frac{1}{s}\mathrm{Poisson}\left(s\,\mathcal{B}_{\mathrm{P}}(\lambda; T)\right) + \mathcal{B}_0, \qquad (38)$$

where $s$ is a parameter related to the gain of the detector (see the Methods for details). Most relevant to the subsequent discussion is

that the mean and variance of $\mathcal{B}$ are

$$\mathbb{E}[\mathcal{B}] = \mathcal{B}_{\mathrm{P}}(\lambda; T) + \mathcal{B}_0,$$
$$\mathbb{V}[\mathcal{B}] = \frac{\mathcal{B}_{\mathrm{P}}(\lambda; T)}{s},$$

and can therefore be independently controlled with the parameters $\mathcal{B}_0$ and $s$.

For the purposes of this example, both candidate models $\mathcal{M}_{\mathrm{RJ}}$ and $\mathcal{M}_{\mathrm{P}}$ make the incorrect (but common) assumption of additive Gaussian noise, such that instead of equation (38) they assume

$$\mathcal{B} \mid \lambda, T, \sigma \sim \mathcal{N}\left(\mathcal{B}_a(\lambda; T), \sigma^2\right), \qquad (39)$$

with $\mathcal{B}_a \in \{\mathcal{B}_{\mathrm{RJ}}, \mathcal{B}_{\mathrm{P}}\}$ and $\sigma > 0$. This ensures that there is always some amount of mismatch between $\mathcal{M}_{\mathrm{true}}$ and the two candidates. That mismatch is increased when $\mathcal{B}_0 > 0$, which we interpret as a sensor bias which the candidate models neglect.

With this setup, we have four parameters which move the problem along three different "axes": The parameters $\lambda_{\min}$ and $\lambda_{\max}$ determine the spectrometer's detection window, and thereby the **ambiguity**: the shorter the wavelength, the easier it is to distinguish the two models. The parameter $s$ determines the **level of noise**. The parameter $\mathcal{B}_0$ determines an additional amount of **misspecification** between the candidate model and the data.

We explore these three axes in Fig. 7, and illustrate how the overlap of the $R_{\mathrm{P}}$ and $R_{\mathrm{RJ}}$ distributions changes through mainly two mechanisms: Better data can shift one $R$-distribution more than the other, and/or it can tighten one or both of the $R$-distributions. Either of these effects can increase the separability of the two distributions (and therefore the strength of the evidence for rejecting one of them).

## Different criteria embody different notions of robustness

As a final step, we now wish to locate our proposed method within the wider constellation of model selection methods.

The motivation behind any model selection criterion is to make the selection in some way robust—i.e., to encourage a choice which is good not just for the given particular data and models, but also for variations thereof. Otherwise we would just select the model with the lowest empirical risk and be done with it. Criteria vary widely however in terms of what kinds of variations they account for. For the purposes of comparing with the EMD rejection rule, we consider three types:

*In-distribution variations of the data* where new samples are drawn from the same data-generating process $\mathcal{M}_{\mathrm{true}}$ as the data.

*Out-of-distribution variations of the data* where new samples are drawn from a different data-generating process $\mathcal{M}'_{\mathrm{true}}$ (than the data-generating process $\mathcal{M}_{\mathrm{true}}$). How much $\mathcal{M}'_{\mathrm{true}}$ is allowed to differ from $\mathcal{M}_{\mathrm{true}}$ will depend on the target application.

*Variations of model parameters*, for example by sampling from a Bayesian posterior, or refitting the model to new data.

Robustness to data variations, whether in- or out-of-distribution, is generally considered a positive. In contrast, robustness to model variations (what might more commonly be described as insensitivity to model parameters) is often attributed to excess model complexity and thus considered a negative.

Exactly which type of variation a criterion accounts for—and just as importantly, how it does so—defines what we might call the *paradigm* for that criterion. In Fig. 8 we compare the EMD approach to five other commonly used criteria with a variety of model selection paradigms; we expand on those differences below.

For our purposes the most important of those differences is how a criterion accounts for epistemic uncertainty. Four of the considered criteria do this by comparing parametrised families of models—either explicitly by averaging their performance over a prior (BIC, $\log \mathcal{E}$), or implicitly by refitting the model to each new dataset (MDL, AIC). In

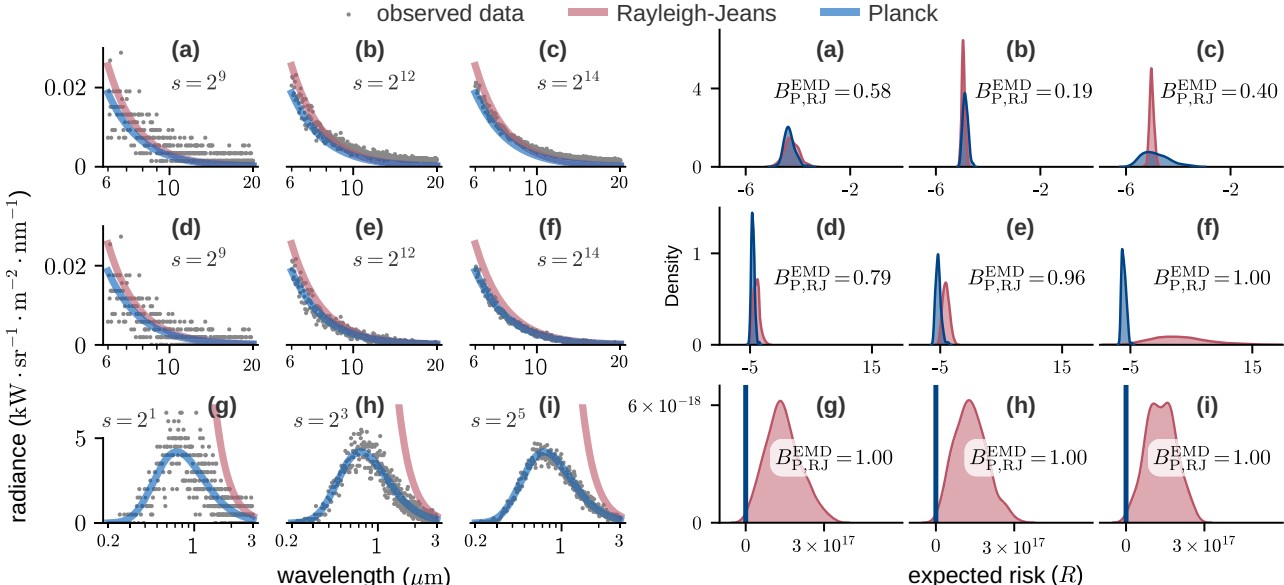

**Fig. 7 | R-distributions (right) for different simulated datasets of spectral radiance.** (left). Datasets were generated using equation (38). For each model $a$, the $R_a$-distribution was obtained by sampling an HB process $\mathfrak{Q}$ parametrised by $\delta_a^{\mathrm{EMD}}$ and a sensitivity $c = 2^{-1}$; see the respective Results sections for definitions of $\mathfrak{Q}$ and $\delta^{\mathrm{EMD}}$. A kernel density estimate is used to visualise the resulting samples (equation 30) as densities. In all rows, noise ($s$) decreases left to right. **Top row:** Over a

range of long wavelengths with positive bias $\mathcal{B}_0 = 0.0015$, both models fit the data equally well. **Middle row:** Same noise levels as the first row, but now the bias is zero. Planck model is now very close to $\mathcal{M}_{\mathrm{true}}$, and consequently has nearly-Dirac $R$-distributions and lower expected risk. **Bottom row:** At visible wavelengths, the better fit of the Planck model is incontrovertible.

other words, such criteria compare the mathematical structure of different models, rather than specific parameters; they would not be suited to the neural circuit example of earlier sections, where the candidate models were solely distinguished by their parameters. The comparison we do here is therefore *only possible* when the models being compared are *structurally distinct*.

Our comparison is also tailored to highlight features of particular importance for experimental data and which differentiate our method. It is by no means exhaustive, and many aspects of model comparison are omitted−such as their consistency or how they account for informative priors. The example data were also chosen to be in a regime favourable to all models; this is partly why many panels show similar behaviour.

We can broadly classify criteria according to whether they select models based solely on their predictive accuracy, or whether they also penalise their complexity (so as to prefer simpler models).

Let us begin with the latter. The most common of these is likely the *Bayes factor*, formally defined as the ratio of model probabilities and in practise the ratio of marginal likelihoods, or *model evidence*[31,45,46]:

$$B_{AB}^{\mathrm{Bayes}} = \frac{p(\mathcal{M}_A \mid \mathcal{D})}{p(\mathcal{M}_B \mid \mathcal{D})} = \frac{\int p(\mathcal{D} \mid \theta, \mathcal{M}_A)\,\pi_A(\theta)d\theta}{\int p(\mathcal{D} \mid \theta, \mathcal{M}_B)\,\pi_B(\theta)d\theta} =: \frac{\mathcal{E}_A}{\mathcal{E}_B}. \qquad (40)$$

(We previously defined the evidence $\mathcal{E}_A$ the same way in equation (6).) Here $p(\mathcal{D} \mid \theta, \mathcal{M}_A)$ and $\pi_A$ are likelihood and prior distributions respectively. Going forward, instead of Bayes factors we will consider each model's log evidence: the former are easily recovered from the difference of the latter,

$$\log B_{AB}^{\mathrm{Bayes}} = \log \mathcal{E}_A - \log \mathcal{E}_B, \qquad (41)$$

and this makes it easier to compare multiple models at once. It also matches the conventional format of information criteria. Numerical evaluation of the model evidence is non-trivial, but generic software solutions are available. For this comparison we used an

implementation of nested sampling[47] provided by the Python package `dynesty`[48].

The model evidence corresponds to studying *variations of the model* (represented by the prior) for a *fixed dataset* $\mathcal{D}$. A model is penalised if its prior is unnecessarily broad or high-dimensional; this is the so-called "Occam's razor" principle which favours simpler models[45,46], which sometimes leads to undesirable results when priors are non-informative[22].

The *Bayesian information criterion* (BIC) is an asymptotic approximation of the log evidence, either around the maximum likelihood or maximum a posteriori estimate[31,35]. It will approach the log evidence ($\log \mathcal{E}$) when the likelihood is non-singular and approximately Gaussian around its maximum.

Another approach which penalises complexity is the *minimum description length (MDL)*. This is more accurately described as a methodological framework built around the choice of a "universal probability distribution"[49]; usually however the *normalised maximum likelihood* (NML) distribution is implied, and this is what we do here. Assuming equal prior preference for each model, MDL selects the one with the highest NML probability; equivalently, it computes for each model the quantity

$$\mathrm{MDL}_A := -\max_\theta p(\mathcal{D} \mid \mathcal{M}_A(\theta)) + \mathrm{COMP}(\mathcal{M}_A). \qquad (42)$$

and selects the model for which $\mathrm{MDL}_A$ is minimal. (It is assumed here that $\mathcal{M}_A$ is parametrised, and $\mathcal{M}_A(\theta)$ denotes the model $\mathcal{M}_A$ with its parameters set to the values $\theta$.) The first term is the maximum of the likelihood evaluated on the training data. The second term is the *MDL complexity*, defined by integrating over the entire event space:

$$\mathrm{comp}(\mathcal{M}_A) := \int \max_\theta p(\mathcal{D}' \mid \mathcal{M}_A(\theta))\,d\mathcal{D}'. \qquad (43)$$

In other words, MDL complexity quantifies the ability of a model to fit arbitrary data. Note that it does not matter here what data-generating process $\mathcal{M}_{\mathrm{true}}$ generated the data since each possible dataset is given

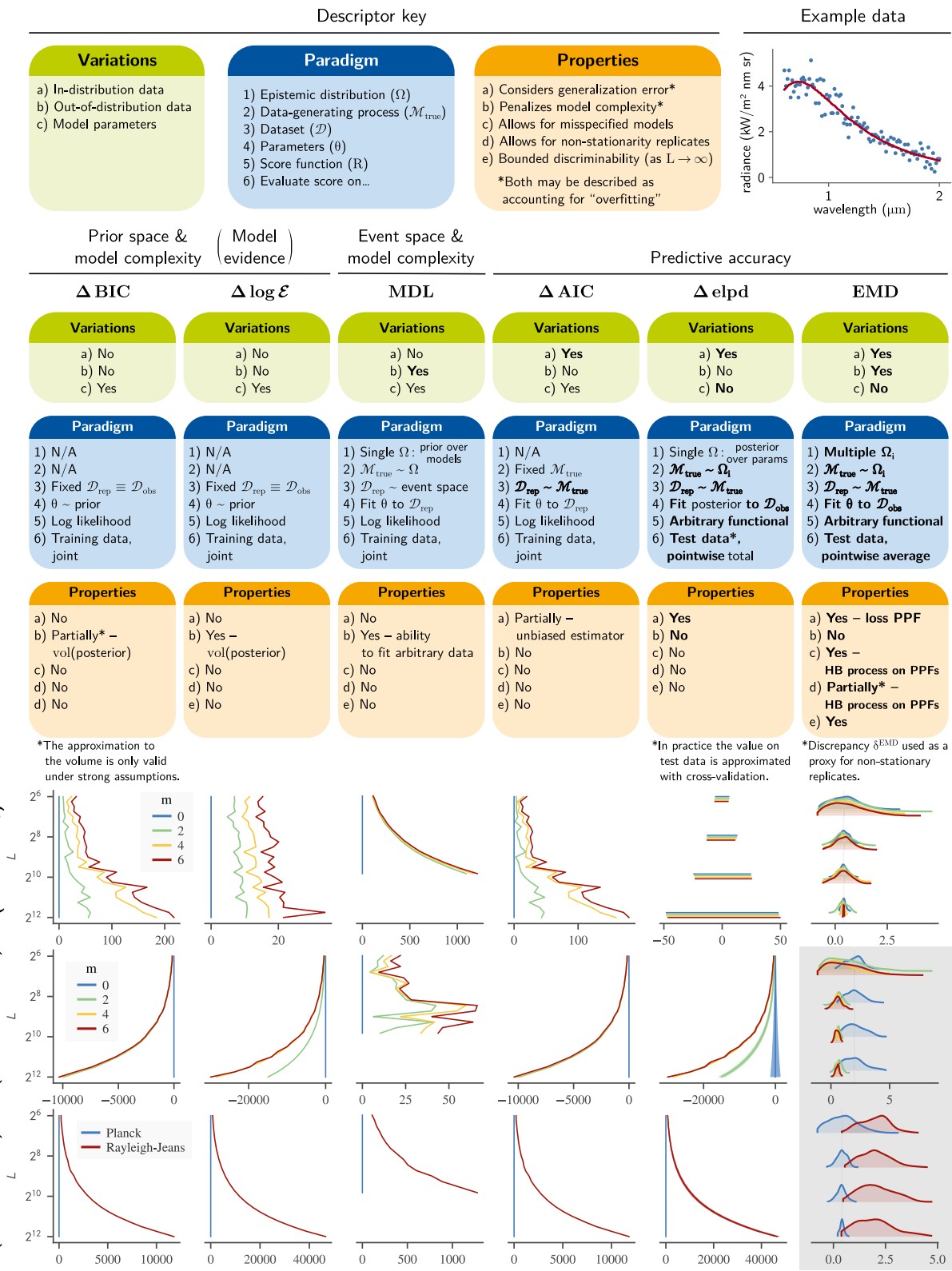

**Fig. 8 | Behaviour of model selection criteria as a function of dataset size.** Datasets were generated using equation (38); they differ only in their number of samples and random seed; an example is shown in the top right. The comparison outcome between two models is given by the difference between their criteria; these are plotted as differences w.r.t. an arbitrary reference model (blue curve). We compare the Bayesian information criterion (BIC), log evidence (log $\mathcal{E}$), minimum description length (MDL), Akaike information criterion (AIC) and expected log pointwise predictive density (elpd) against our empirical model discrepancy (EMD) approach. The log evidence (log $\mathcal{E}$) and elpd provide uncertainties, shown either as shading or as error bars. MDL curves are shorter because computations exceeding 100 hours were aborted.

equal weight. Approximations are almost always required to compute equation (43) since the integral is usually intractable; for this comparison, we chose a data regime where it is just about possible to enumerate the possible datasets and estimate $comp(\mathcal{M}_A)$ directly with Monte Carlo.

With MDL therefore we consider variations of the models, since each model is refitted to each dataset $\mathcal{D}'$. We have no in-distribution dataset variations (the true process $\mathcal{M}_{true}$ is ignored), but we do have out-of-distribution variations in the form of the integral over the event space.

Moving on now to criteria which compare predictive accuracy, we start with the *expected log pointwise predictive density (elpd)*[32]. This is defined as the expectation of the loss $Q$ for $L$ samples drawn from the data-generating process:

$$elpd_A := \sum_{i=1}^{L} \mathbb{E}_{(x_i, y_i) \sim \mathcal{M}_{true}} \left[ Q(x_i, y_i; \mathcal{M}_A) \right]$$
$$= L\, R_A . \qquad (44)$$

The most common form of the elpd uses the (negative) log likelihood as the loss function $Q$. In this form where each sample is drawn from the same distribution, the elpd is proportional to the risk $R_A$ defined in equation (4), and therefore conceptually analogous: it describes robustness to in-distribution variations of the dataset. The models are kept fixed in these comparisons; they are not refitted to the new data. The uncertainty on elpd comes from the finite number of samples used to compute the expectation.

Since in current practice the elpd is mostly used with Bayesian models, this is also what we do in Fig. 8. Efficient estimators exist for computing the elpd from training data, using either leave-one-out cross-validation[32] or the widely applicable information criterion (WAIC)[50]. This avoids the need to reserve a separate test dataset. We used ArviZ's[51] implementation of WAIC for this comparison.

In contrast to the elpd, the *Akaike information criterion (AIC)* does not assume a fixed model. Although it also ranks models according to their expected loss, this time the expectation is (conceptually) over both the test set $\mathcal{D}_{test}$ and training data $\mathcal{D}_{train}$, both drawn independently from the data-generating process[33,34]:

$$elpd_A^{AIC} = \mathbb{E}_{\mathcal{D}_{test} \sim \mathcal{M}_{true}} \mathbb{E}_{\mathcal{D}_{train} \sim \mathcal{M}_{true}} \left[ Q(\mathcal{D}_{test}; \mathcal{M}_A(\hat{\theta}_{\mathcal{D}_{train}})) \right], \qquad (45)$$

where $\hat{\theta}_{\mathcal{D}_{train}}$ is the maximum likelihood estimate given the training data. Concretely however the AIC is a simple statistic, similar to the BIC and other information criteria:

$$AIC_A := 2\, Q\left(\mathcal{D}_{train}; \mathcal{M}_A(\hat{\theta}_{\mathcal{D}_{train}})\right) + 2k_A , \qquad (46)$$

where $k_A$ is the number of parameters of model $\mathcal{M}_A$. The derivation of the AIC relies on $Q$ being the log likelihood, and assumes that the likelihood is approximately Gaussian and non-singular around $\hat{\theta}_{\mathcal{D}_{train}}$. When these conditions are verified, then $\Delta AIC_{AB} := AIC_A - AIC_B$ is an unbiased estimator of the difference $elpd_A^{AIC} - elpd_B^{AIC}$. In this case, the model with the lowest AIC will—in expectation—also be the one with the lowest expected risk.

So how does the $B^{EMD}$ compare to the criteria listed above? First, it computes $R$-distributions for *fixed models*. This is a key feature, since otherwise it is impossible to compare different parameter vectors for otherwise structurally identical models. (Note that EMD also allows models to have different structure—it simply does not require it.) This also means that the $B^{EMD}$ does not penalise model complexity, in contrast to the evidence, MDL and AIC.

Second, like the elpd, the $B^{EMD}$ accounts for *in-distribution variability* by estimating the risk using held-out test data $\mathcal{D}_{test}$.

Third, EMD $R$-distributions also account for *out-of-distribution variability*. Conceptually this is done via the epistemic distributions $\Omega$, concretely via the HB process on PPFs—the variance of the latter stemming ultimately from the discrepancy function $\delta^{EMD}$. Better models therefore result in less variability, reflecting our better grasp of the process being modelled. In contrast, only the MDL allows for some form of out-of-distribution variability (in the form of a complexity penalty), and it is essentially rigid: always an integral over the entire event space.

The biggest difference between the EMD $R$-distributions and other criteria however is their behaviour as the number of samples $L$ increases. This we illustrate in the lower part of Fig. 8, which plots the value of a criterion for different models as a function of $L$. We do this for three collections of models, where the data-generating process is always the Planck model $\mathcal{B}_P$ described in the previous section. In all panels, scores are defined so that models with lower scores are preferred.

In the first row we compare four candidate models of the form

$$\mathcal{B} \mid \lambda, T, \sigma, \{b_j\} \sim \mathcal{N}\left( \mathcal{B}_P(\lambda; T) + \sum_{j=0}^{m} b_j \lambda^j, \ \sigma^2 \mathcal{B}_P(\lambda; T) \right). \qquad (47)$$

The candidates are therefore nested: any model within the class $\mathcal{M}_{m=2}$ can be recovered from the class $\mathcal{M}_{m=4}$ by fixing $b_3 = b_4 = 0$. Moreover, all models effectively contain $\mathcal{M}_{true}$, since it corresponds to $m = 0$. (The data are in a regime where the Poisson distribution of $\mathcal{M}_{true}$ is close to Gaussian.)

The second row is similar, but expands around the Rayleigh-Jeans model instead:

$$\mathcal{B} \mid \lambda, T, \sigma, \{b_j\} \sim \mathcal{N}\left( \mathcal{B}_{RJ}(\lambda; T) + \sum_{j=0}^{m} b_j \lambda^j, \ \sigma^2 \mathcal{B}_{RJ}(\lambda; T) \right). \qquad (48)$$

In this case all models are misspecified, but those with higher degree polynomials are better able to compensate for this and thus achieve lower prediction errors.

The third row compares the two $m = 0$ models from the first two rows.

If we interpret the score difference between two models (measured as the horizontal distance between two curves) as the strength of the evidence supporting a particular model choice, then it is clear from Fig. 8 that for all criteria except EMD, strength grows unbounded as we increase the number of samples $L$. A bounded strength however is desirable when analyzing experimental data, since no experiment has infinite accuracy.

In contrast, we see that the $R$-distributions in the grey shaded area stabilise as a function of $L$. Therefore also the tail probabilities $B_{ab;c}^{EMD} = P(R_A < R_B : c)$ (equation (16)), which define the EMD rejection rule, converge in general to finite, non-zero values. In some cases the $R$-distributions may approach a Dirac delta as $L$ increases, as we see in the first row of Fig. 8. This occurs when the candidate model is able to match $\mathcal{M}_{true}$ exactly.

We provide a different set of comparisons in the Supplementary Results, listing numerical values of criteria differences in Supplementary Table 1 to complement the graphical presentation of Fig. 8. These alternative comparisons place greater emphasis on how misspecification can affect a criterion's value, and how that value can (mis)represent the strength of the evidence.

## Discussion

Model selection in the scientific context poses unique challenges. Ranking models according to their empirical risk captures the scientific principle that models should be able to predict new data, but in order for results to be reproducible across experiments, that risk should also be robust to variations in the replication process. We propose that the empirical model discrepancy between quantile functions ($\delta^{EMD}$)—measured from a single dataset—can serve as a baseline for the uncertainty across replications. We represent this uncertainty by the stochastic process $\Omega$; due to intrinsic constraints of quantile functions, this process is relatively stereotyped, allowing us to automate the generation of *R(isk)-distributions*—visually summarised in Fig. 1—from empirical data (see Code availability). Moreover, we can calibrate the process against simulated replications—by adjusting the proportionality factor $c$ which converts model discrepancy into epistemic uncertainty—so that it approximates the epistemic variability expected in experiments. In the end, the decision to reject a model in favour of another reduces to a simple tail probability $B_{AB;c}^{EMD} = P(R_A < R_B \mid c)$ between two $R$-distributions (equation (12)). Crucially, we do not force a model choice: a model is only rejected if this probability exceeds a chosen threshold $\epsilon$. Guidelines for choosing rejection thresholds should be similar to those for choosing significance levels.

A key feature of our approach is to compare models based on their risk. This is an uncommon choice for statistical model selection, because it tends to select overfitted models when the same data are used both to fit and test them. In more data-rich machine learning paradigms however, risk is the preferred metric, and overfitting is instead avoided by testing models on held-out data. Our method assumes this latter paradigm, in line with our motivation to compare complex, data-driven models. A selection rule based on risk is also easily made consistent (in the sense of asymptotically choosing the correct model) by choosing a proper loss function, such as the negative log likelihood. Moreover, with the specific choice of a log likelihood loss, differences in risk can be interpreted as differences in differential entropy. The risk formulation however is more general, and allows the choice of loss to be tailored to the application.

We illustrated the approach on two example problems, one describing the radiation spectrum of a black body, and the other the dynamical response of a neuron of the lobster pyloric circuit. In the case of the former, we compared the Planck and Rayleigh-Jeans models; these being two structurally different models, we could also compare the $B^{EMD}$ with other common selection criteria (Fig. 8). We thereby highlight not only that different criteria account for different types of variations, but also that $B^{EMD}$ probabilities are unique in having well-defined, finite limits (i.e., neither zero nor infinite) when the number of samples becomes large. This is especially relevant in the scientific context, where we want to select models which work for datasets of all sizes. The simplicity of the black body radiation models also allowed us to systematically characterise how model ambiguity, misspecification, and observation noise affect $R$-distributions (Fig. 7).

The neural dynamics example was inspired by the increasingly common practice of data-driven modelling. Indeed, Prinz et al.[6,52]—whose work served as the basis for this example—can be viewed as early advocates for data-driven approaches. Although their exhaustive search strategy faces important limitations[53], more recent approaches are much more scalable. For example, methods using gradient based optimisation can simultaneously learn dozens—or in the case of neural networks, millions—of parameters[2,54]. These methods however are generally used to solve non-convex problems, and therefore run against the same follow-up question: having found (possibly many) candidate models, which ones are truly good solutions, and which ones should be discarded as merely local optima? The $B^{EMD}$ probabilities, which account for epistemic uncertainty on a model's risk, can address this latter question. They can do so even when candidate models are structurally identical (distinguished therefore only by the values of their parameters), which sets this method apart from most other model selection criteria.

A few other model selection methods have been proposed for structurally identical candidate models, either within the framework of approximate Bayesian computing[55] or by training a machine learning model to perform model selection[56]. Largely these approaches extend Bayesian model selection to cases where the model likelihood is intractable, and as such should behave similarly to the Bayesian methods studied in Fig. 8. Epistemic uncertainty is treated simply as a prior over models, meaning in particular that these approaches do not account for variations in the replication process.

Epistemic uncertainty itself is not a novel idea: Kiureghian & Ditlevsen[13] discuss how it should translate into scientific practice, and Hüllermeier & Waegeman[14] do the same for machine learning practice. There again however, the fact that real-world replication involves variations of the data-generating process is left largely unaddressed. Conversely, the property of a model of being robust to replication variations is in fact well ingrained in the practice of machine learning, where it is usually referred to as "generalisability", or more specifically "out-of-distribution performance". This performance however is usually only measured post hoc on specific alternative datasets. Alternatively, there have been efforts to quantify the epistemic uncertainty by way of ensemble models, including the GLUE methodology[16,57], Bayesian calibration[18,58], Bayesian neural networks[21], and drop-out regularisation at test time[19]. Since these methods focus on quantifying the effect of uncertainty on model predictions, they improve the estimate of risk, but still do not assign uncertainty to that estimate. In contrast, with the hierarchical beta process $\Omega$, we express epistemic uncertainty on the statistics of the sample loss. This has two immediate advantages: the method trivially generalises to high-dimensional systems, and we obtain distributions for the risk (Fig. 3). In this way, uncertainty in the model translates to uncertainty in the risk.

A few authors have argued for making robustness against replication uncertainty a logical keystone for translating fitted models into scientific conclusions[11,12], although without modelling replications explicitly. There, as here, the precise definition of what we called an epistemic distribution is left to the practitioner, since appropriate choices will depend on the application. One would of course expect a selection rule not to be too sensitive to the choice of epistemic distribution, exactly because it should be robust.

One approach which does explicitly model replications, and uses similar language as our own to frame its question, is that of Gelman et al.[59] Those authors also model the distribution of discrepancies (albeit here again of model predictions rather than losses) and compute tail probabilities. Their method however is designed to assess the plausibility of a single model; it would not be suited to choosing the better of two approximations. It is also intrinsically Bayesian, using the learned model itself (i.e., the posterior) to simulate replications; we do not think it could be generalised to non-Bayesian models like our neural circuit example.

Building on similar Bayesian foundations but addressing model comparison, Moran et al.[60] propose the posterior predictive null check (PPN), which tests whether the predictions of two models are statistically indistinguishable. In contrast to the $B^{EMD}$, the PPN is purely a significance test: it can detect if two models are distinguishable, but not which one is better, or by how much. One can see parallels between this approach and the model selection tests (MST) proposed by Golden[7,61]. While MST is framed within the same paradigm as AIC—and so makes the same approximation that inferred parameters follow a Gaussian distribution—instead of simply proposing an unbiased estimator for the difference $R_A - R_B$, that difference is mapped to a $\chi^2$ distribution. This allows the author to propose a selection procedure similar in spirit to equation (11), but where the "no rejection" case is selected on the basis of a significance test. Although replication

uncertainty is not treated—and therefore the test can be made arbitrarily significant by increasing the number of samples—the author anticipates our emphasis on misspecification and our choice to select models based on risk.

A guiding principle in designing the EMD rejection rule was to emulate how scientists interpret noisy data. This approach of using conceptually motivated principles to guide the development of a method seems to be fruitful, as it has lead to other recent developments in the field of model inference. For instance, Swigon et al.[62] showed that by requiring a model to be invariant under parameter transformation, one can design a prior (i.e., a regulariser) which better predicts aleatoric uncertainty. Another example is simulation-based calibration (SBC)[63,64], for which the principle is self-consistency of the Bayesian model. This is conceptually similar to the calibration procedure we proposed: in both cases a consistency equation is constructed by replacing one real dataset with many simulated ones. The main difference is that SBC checks for consistency with the Bayesian prior (and therefore the aleatoric uncertainty), while we check for consistency with one or more epistemic distributions. In both cases the consistency equation is represented as a histogram: in the case of SBC it should be flat, while in our case it should follow the identity $B^{\mathrm{epis}} = B^{\mathrm{EMD}}$.

The model discrepancy $\delta^{\mathrm{EMD}}$ (equation (23)) also has some similarities with the Kolmogorov-Smirnoff (K-S) statistic; the latter is defined as the difference between two CDFs and is also used to compare models. Methodologically, the K-S statistic is obtained by taking the supremum of the difference, whereas we keep $\delta^{\mathrm{EMD}}$ as a function over $[0, 1]$; moreover, a Kolmogorov-Smirnoff test is usually computed on the distribution of data samples ($\mathcal{D}_{\mathrm{test}}$) rather than that of their losses ($\{Q(x, y) : (x, y) \in \mathcal{D}_{\mathrm{test}}\}$).

One would naturally expect a comparison criterion to be symmetric: the evidence required to reject model $\mathcal{M}_A$ must be the same whether we compare $\mathcal{M}_A$ to $\mathcal{M}_B$ or $\mathcal{M}_B$ to $\mathcal{M}_A$. Moreover, it must be possible that neither model is rejected if the evidence is insufficient. Bayes factors and information criteria do allow for this, but with an important caveat: the relative uncertainty on those criteria is strongly tied to dataset size, so much so that with enough samples, there is always enough data to reject a model, as we illustrate in Fig. 8. This is in clear contradiction with scientific experience, where more data cannot compensate for bad data.

Another important consequence of a symmetric criterion is that it trivially generalises to comparing any number of models, such as we did in Table 1. One should however take care in such cases that models are not simply rejected by chance when a large number of models are simultaneously compared. By how much that chance would increase, and how best to account for this, are questions we leave for future work.

From a practical standpoint, the most important feature of the $B^{\mathrm{EMD}}$ is likely its ability to compare both specific model parametrisations of the same structural model, as well as structurally different models. Most other criteria listed in Fig. 8 only compare model structures, because they either assume globally optimal parameters (AIC, BIC), integrate over the entire parameter space (Bayes factor) or refit their parameters to each dataset (AIC, MDL). The calculation of the $B_{AB;c}^{\mathrm{EMD}}$ between two models is also reasonably fast, taking less than a minute in most of our examples. Although the calibration experiments are more computationally onerous, for both of our models we found that the $B_{AB;c}^{\mathrm{EMD}}$ is valid over a range of $c$ values, with the high end near $c = 1$. This suggests a certain universality to the $B_{AB;c}^{\mathrm{EMD}}$, which would allow practitioners to perform initial analyses with a conservative value like $c = 1$, only then proceeding to calibration experiments if there is indication that they are worth pursuing. Alternatively, since calibration is a function of experimental details, a laboratory might perform it once to determine a good value of $c$ for its experimental setup, and then share that value between its projects.

This property that the same value of $c$ can be used to compare different models in different contexts is the key reason why we expect the $B^{\mathrm{EMD}}$ to generalise from simulated epistemic distributions to real-world data. It is clear however that this cannot be true for arbitrary epistemic distributions: we give some generic approaches for improving the outcome of calibrations in the Supplementary Discussion, but the $B^{\mathrm{EMD}}$ criterion will always work best in cases where one has a solid experimental control (to minimise variability between replications) and deep domain knowledge (to accurately model both the system and the replications).

Another important feature is that computing the $B^{\mathrm{EMD}}$ only requires knowledge which is usually accessible in practice: a method to generate synthetic samples from both $\mathcal{M}_A$ and $\mathcal{M}_B$, a method to generate true samples, and a loss function. Some reasoned choices are used to define the stochastic process $\mathfrak{Q}$ in the PPF space, but they are limited by the hard constraints of that space: PPFs must be one-dimensional on $[0, 1]$, monotone, integrable, and non-accumulating. Although they make defining an appropriate stochastic process considerably more complicated, these constraints seem to be key for obtaining distributions which are representative of epistemic uncertainty (see Supplementary Fig. 4). We proposed *hierarchical beta (HB) processes* (further detailed in the Methods) to satisfy these constraints, but it would be an interesting avenue of research to look for alternatives which also satisfy the desiderata for $\mathfrak{Q}$. In particular, if one could define processes which better preserve the proportionality of (27) at large values of $c$, or allow for a faster computational implementation, the $B^{\mathrm{EMD}}$ could be applied to an even wider range of problems.

There are other open questions which would be worth clarifying in future work. One is the effect of finite samples: having fewer samples increases the uncertainty on the shape of the estimated mixed PPF $q^*$, and should therefore increase the discrepancy with the synthetic PPF $\tilde{q}$ (and thereby the estimated epistemic uncertainty). In this sense our method accounts for this implicitly, but whether it should do so explicitly—and how—is yet unexplored. It may also be desirable in some cases to use the same samples both to fit and test models, as it is commonly done in statistical model selection. One might explore for such cases the possibility of combining our approach with existing methods, such as those developed to estimate the elpd[32,65], to avoid selecting overfitted models.

Looking more broadly, it is clear that the wide variety of possible epistemic distributions cannot be fully captured by the linear relationship of (27), which formalises as a desideratum for the stochastic process $\mathfrak{Q}$ what we referred to as the *EMD principle*. Going forward, it will be important therefore to consider the $B^{\mathrm{EMD}}$ criterion an imperfect solution, and to continue looking for ways to better account for epistemic uncertainty. These could include relating misspecification directly to the distribution of losses (going further than the $B^Q$ considered our Supplementary Discussion), or more sophisticated self-consistency metrics than our proposed $\delta^{\mathrm{EMD}}$. A key strategy here may be to look for domain-specific solutions.

Already however, the $B^{\mathrm{EMD}}$ as proposed offers many opportunities to incorporate domain expertise: in the choice of the candidate models, loss function, sensitivity parameter $c$, rejection threshold $\epsilon$, and epistemic distributions used to validate $c$. These are clear interpretable choices which help express scientific intent. These choices, along with the assumptions underpinning the $B^{\mathrm{EMD}}$, should also become testable: if they lead to selecting models which continue to predict well on new data, that in itself will provide for them a form of empirical validation.

## Methods

### Poisson noise model for black body radiation observations

In equation (38), we used a Poisson counting process to simulate the observation noise for recordings of a black body's radiance. This is a

more realistic model than Gaussian noise for this system, while still being simple enough to serve our illustration.

The physical motivation is as follows. We assume that data are recorded with a spectrometer which physically separates photons of different wavelengths and measures their intensity with a CCD array. We further assume for simplicity that wavelengths are integrated in bins of equal width, such that the values of $\lambda$ are sampled uniformly (the case with non-uniform bins is less concise but otherwise equivalent). We also assume that the device uses a fixed time window to integrate fluxes, such that what it detects are effective photon counts. The average number of counts is proportional to the radiance, but also to physical parameters of the sensor (including size, integration window and sensitivity) which we collect into the factor $s$; the units of $s$ are $m^2 \cdot nm \cdot photons \cdot sr \cdot kW^{-1}$, such that $s\mathcal{B}_a(\lambda; T)$ is a number of photons. Since the photons are independent, the recorded number of photons will be random and follow a Poisson distribution. This leads to the following data-generating process $\mathcal{M}_{\text{true}}$ (first line is repeated from equation (39)):

$$\mathcal{B}|\lambda, T \sim \frac{1}{s}\text{Poisson}\left(s\,\mathcal{B}_P(\lambda; T)\right) + \mathcal{B}_0\,, \tag{49}$$
$$\lambda \in \{\lambda_{\min}, \lambda_{\min} + \Delta\lambda, \lambda_{\min} + 2\Delta\lambda, \ldots, \lambda_{\max}\}.$$

Here $\mathcal{B}_0$ captures the effect of dark currents and random photon losses on the radiance measurement. Recall that a random variable $k$ following a distribution $\text{Poisson}(\mu)$ has probability mass function $P(k) = \frac{\mu^k e^{-\mu}}{k!}$.

Note that we divide by $s$ so that $\mathcal{B}$ also has dimensions of radiance and is comparable with the models $\mathcal{B}_P$ and $\mathcal{B}_{RJ}$ defined in equations (36) and (37).

### Neuron model

The neuron model used in our Results is the Hodgkin-Huxley-type model of the lobster pyloric rhythm studied by Prinz et al.[6]. The specific implementation we used can be obtained from reference 66, along with a complete description of all equations and parameters. We summarise the model below and refer the reader to that reference for more details.

For each cell $k$, the potential across a patch of area $A$ and capacitance $C$ evolves according to the net ionic current through the cell membrane:

$$\frac{C}{A}\frac{dV^k}{dt} = -\underbrace{\sum_{i \in \text{ion channels}} I_i^k}_{\substack{\text{ion diffusion} \\ \text{through membrane}}} - \underbrace{\sum_{l \in \text{neurons}} I_s^l}_{\substack{\text{chemical} \\ \text{synapses}}} - \underbrace{\sum_{l \in \text{neurons}} I_e^{kl}}_{\substack{\text{electrical} \\ \text{synapses}}} - I_{\text{ext}}^k\,, \tag{50}$$

where $I_{\text{ext}}$ is an arbitrary external current, the $I_i$ describe ion exchanges between a cell and its environment, and $I_s$ and $I_e$ describe ion exchanges between different cells. The $I_{\text{ext}}$ current can be used to represent a current applied by the experimenter, or the inputs from other cells in the network. The voltage $\tilde{V}$ which an experimenter records would then be some corrupted version of $V$, subject to noise sources which depend on their experiment.

$$\tilde{V}(t) \sim \text{Experimental noise}\,(V(t)). \tag{51}$$

We generate the external input $I_{\text{ext}}$ as a Gaussian coloured noise with autocorrelation:

$$\langle I_{\text{ext}}(t)\,I_{\text{ext}}(t')\rangle = \sigma_i^2 e^{-(t-t')^2/2\tau^2}\,. \tag{52}$$

We do this using an implementation[67] of the sparse convolution algorithm[68]. We found that in addition to being more realistic, coloured noise also smears the model response in time and thus reduces degeneracies when comparing models.

Each cell in the model has eight currents through ion channels, indexed by $i$: one $Na^+$ current, two $Ca^{2+}$ currents, four $K^+$ currents and one leak current. Each current is modelled as (square brackets indicate functional dependence)

$$I_e^{kl} = g_e(V^k - V^l)\,, \qquad \tau_{m,i}[V^k]\frac{dm}{dt} = m_{\infty,i}[V^k] - m_i^k\,,$$

$$I_i^k = g_i^k\,(m_i^k)^{p_i}\,h_i\,(V^k - E_i)\,, \qquad \tau_{h,i}[V^k]\frac{dh}{dt} = h_{\infty,i}[V^k] - h_i^k\,,$$

$$I_s^{kl} = g_s^{kl}\,s^l\,(V^k - E_s^k)\,, \qquad \tau_s^l[V^l]\frac{ds}{dt} = s_\infty^l[V^l] - s^l\,.$$

These equations are understood as describing currents through permeable channels with maximum conductivity $g_i^k$ (for membrane currents within the same cells) or $g_s^{kl}$ (for synaptic currents between different cells). Conductivities are dynamic: they are governed by the equations for the gating variables $m$, $h$ and $s$ given above. (Some channels do not have inactivating gates; for these, $h_i$ is set to 1.) The fixed points $m_\infty$, $h_\infty$ and $s_\infty$, as well as the time constants $\tau_m$, $\tau_h$ and $\tau_s$, are functions of the voltage; the precise shape of these functions is specific to each channel type and can be found in either references 6 or 66.

Following Prinz et al., we treat the functions $m_\infty$, $h_\infty$, $s_\infty$, $\tau_m$, $\tau_h$ and $\tau_s$, as well as the electrical conductance $g_e$ and the Nernst ($E_i$) and synaptic ($E_{s,k}$) reversal potentials, as known fixed quantities. Thus the only free parameters in this model are the maximum conductances $g_i^k$ and $g_s^{kl}$: the former determine the type of each neuron, while the latter determine the circuit connectivity.

The pyloric circuit model studied by Prinz et al.[52] consists of three populations of neurons with eight different ion channels. Biophysically plausible values for the channel and connectivity parameters were determined through separate exhaustive parameter searches by Prinz et al.[6,52], the results of which were reduced to sixteen qualitatively different parameter solutions: 5 *AB/PD* cells, 5 *LP* cells and 6 *PY* cells. Importantly, these parameter solutions are distinct: interpolating between them does not yield models which reproduce experimental recordings. For purposes of illustration we study the simple two-cell circuit shown in Fig. 2a, where an AB cell drives an LP cell. Moreover we assume the parameters of the AB cell to be known, such that we only need to compare model candidates for the LP cell. (These assumptions are not essential to applying our method, but they avoid us contending with model-specific considerations orthogonal to our exposition.)

The AB neuron is an autonomous pacemaker and serves to drive the circuit with realistic inputs; all of our examples use the same AB model (labelled 'AB/PD 3' in Table 2 of Prinz et al.[6]), whose output is shown in Fig. 2b. Panels c and d show the corresponding response for each of the five LP models given in Table 2 of Prinz et al.[6]

To generate our simulated observations, we use the output of LP 1, add Gaussian noise and then round the result (in millivolts) to the nearest 8-bit integer; in this way $\mathcal{M}_{\text{true}}$ includes both electrical and digitisation noise. This leaves LP 2 through LP 5 to serve as candidate models; for these we assume only Gaussian noise, and we label them $\mathcal{M}_A$ to $\mathcal{M}_D$. We use $\Theta_a$ to denote the concatenation of all parameters for a given model $a$, which here consist of the vectors of conductance

**Table 2 | Neuron circuit model labels**

| Model symbol | Model components |
|---|---|
| $\mathcal{M}_{\text{true}}$ | $I_{\text{ext}} \rightarrow$ AB 3 $\rightarrow$ LP 1 $\rightarrow$ Gaussian noise $\rightarrow$ digitise |
| $\mathcal{M}_A$ | $I_{\text{ext}} \rightarrow$ AB 3 $\rightarrow$ LP 2 $\rightarrow$ Gaussian noise |
| $\mathcal{M}_B$ | $I_{\text{ext}} \rightarrow$ AB 3 $\rightarrow$ LP 3 $\rightarrow$ Gaussian noise |
| $\mathcal{M}_C$ | $I_{\text{ext}} \rightarrow$ AB 3 $\rightarrow$ LP 4 $\rightarrow$ Gaussian noise |
| $\mathcal{M}_D$ | $I_{\text{ext}} \rightarrow$ AB 3 $\rightarrow$ LP 5 $\rightarrow$ Gaussian noise |

values $g_i$ and $g_s$. Model definitions are summarised in Table 2 and in Algorithm 1.

**Algorithm 1.** Neuron model

 **procedure** Generate Data($\mathcal{T}$)
 **for** $t \in \mathcal{T}$ **do**
 **integrate** equation (50) to obtain $V^{AB}(t)$, $V^{LP}(t; \Theta_{\text{true}})$
 **draw** $\xi(t) \sim \mathcal{N}(0, \sigma_o)$
 **evaluate** $\tilde{V}^{LP}(t) = \texttt{digitize}_8\left(V^{LP}(t) + \xi(t)\right)$
 **end for**
 **return** $\left\{ \tilde{V}^{LP}(t) : t \in \mathcal{T} \right\}$
 **end procedure**

 **procedure** Simulate Candidate($\mathcal{T}, a \in \{A, B, C, D\}$)
 **for** $t \in \mathcal{T}$ **do**
 **integrate** equation (50) to obtain $V^{AB}(t)$, $V^{LP}(t; \Theta_a)$
 **draw** $\xi(t) \sim \mathcal{N}(0, \sigma_o)$
 **evaluate** $\tilde{V}^{LP}(t) = V^{LP}(t; \Theta_a) + \xi(t)$
 **end for**
 **return** $\left\{ \tilde{V}^{LP}(t) : t \in \mathcal{T} \right\}$
 **end procedure**

 **procedure** digitise$_8$($x$)
 Simulates the data encoding of a digital sensor by converting $x$ to an 8-bit integer
 **clip**: $x \leftarrow \max(-128, \min(x, 127))$
 **return** int($x$)
 **end procedure**

### Loss function for the neuron model

To evaluate the risk of each candidate model, we use the log likelihood of the observations (equation (9)). This standard choice is convenient for exposition purposes: it is simple to explain and illustrates the generality of the method, since a likelihood function is available for any model in the form of equation (1).

However it is not a requirement to use the negative log likelihood as the loss, and in fact for time series models it can be disadvantageous. For example, the neuron models used in this work have sharp temporal responses (spikes), which makes the log likelihood sensitive to the timing of these spikes. In practice a less sensitive loss function may be preferable, although the best choice will depend on the application.

### Construction of an *R*-distribution from an HB process $\mathfrak{Q}$

For each candidate model we use the hierarchical beta process $\mathfrak{Q}_a$ described below to generate on the order of $M_a \approx 100$ PPFs $\hat{q}_{a,1}, \ldots, \hat{q}_{a,M_a}$; the exact number of PPFs is determined automatically, by increasing the number $M_a$ until the relative standard error on $\frac{1}{M_a}\sum_{i=1}^{M_a} R[\hat{q}_{a,i}]$ is below $2^{-5}$ (six such PPFs are shown as grey traces in Fig. 5). Each curve is integrated to obtain a value for the risk (equation (20)), such that the $R_a$ distribution can be represented by the set $\{R[\hat{q}_{a,i}]\}_{i=1}^{M_a}$. We then use a kernel density estimate to visualise these distributions in Fig. 3; specifically we use the `univariate_kde` function provided by Holoviews[69] with default parameters. The function automatically determines the bandwidth.

### Calibration experiments

As described in our Results, the goal of calibration is twofold. First we want to correlate the value of $B_{AB;c}^{\text{EMD}}$ with the probability $B_{AB;\Omega}^{\text{epis}}$ that, across variations of the data-generating process $\mathcal{M}_{\text{true}}$, model $A$ has lower risk than model $B$. We use an epistemic distribution $\Omega$ to describe those variations. Second, we want to ensure that a decision to reject either model based on $B_{AB}^{\text{EMD}}$ is robust: it should hold for any reasonable epistemic distribution $\Omega$ of experimental conditions (and therefore hopefully also for unanticipated experimental variations).

The calibration procedure involves fixing two candidate models and an epistemic distribution $\Omega$. We then sample multiple data-generating models $\mathcal{M}_{\text{true}}$ from $\Omega$, generate a replicate dataset $\mathcal{D}_{\text{test}}^{\text{rep}}$ from each, and compute both $B^{\text{EMD}}$ and $B^{\text{epis}}$ on that replicate dataset.

This process can be repeated for as many epistemic distributions and as many different pairs of candidate models as desired, until we are sufficiently confident in the robustness of $B^{\text{EMD}}$. This means in particular that we cannot expect perfect correlation between $B^{\text{EMD}}$ and $B^{\text{epis}}$ across all models and conditions.

However we don't actually require perfect correlation because this is an asymmetrical test: a decision to reject neither model is largely preferred over a decision to reject the wrong one. Therefore what we especially want to ensure is that for any pair of models $A, B$, and any reasonable epistemic distribution $\Omega$, we have

$$\left| B_{AB;c}^{\text{EMD}} - 0.5 \right| \lesssim \left| B_{AB;\Omega}^{\text{epis}} - 0.5 \right|. \tag{53}$$

This is given as either equation (34) or (35) in the Results.

Note that $B_{AB;\Omega}^{\text{epis}}$ is defined as the fraction of experiments (i.e., data-generating models $\mathcal{M}_{\text{true}}$) for which $R_A$ is smaller than $R_B$. Therefore it cannot actually be calculated for a single experiment; only over sets of experiments. Since our goal is to identify a correlation between $B_{AB;\Omega}^{\text{epis}}$ and $B_{AB;c}^{\text{EMD}}$, we compute the former as a function of the latter:

$$
\begin{aligned}
B_{AB;\Omega}^{\text{epis}}\left(B_{AB;c}^{\text{EMD}}\right) &:= P\left(R_A < R_B \mid \Omega, B_{AB;c}^{\text{EMD}}\right) \\
&= P_{\mathcal{M}_{\text{true}} \sim \Omega | B_{AB;c}^{\text{EMD}}}(R_A < R_B) \\
&= \mathbb{E}_{\mathcal{M}_{\text{true}} \sim \Omega | B_{AB;c}^{\text{EMD}}}\left[R_A < R_B\right].
\end{aligned}
\tag{54}
$$

(On the last line, [ ] is the Iverson bracket, which is 1 when the clause it contains is true, and 0 otherwise.) In other words, we partition the experiments in the domain of $\Omega$ into subsets having the same value $B_{AB;c}^{\text{EMD}}$, then compute $B_{AB;\Omega}^{\text{epis}}$ over each subset. In practice we do this by binning the experiments according to $B_{AB;c}^{\text{EMD}}$, as we explain below.

### Calibration and computation of *B*^epis^ for the black body radiation models.

For the black body models, we consider one epistemic distribution $\Omega$ with two sources of experimental variability: the true physical model ($\mathcal{B}_P$ or $\mathcal{B}_{RJ}$) and the bias $\mathcal{B}_0$. All other parameters are fixed to the standard values used throughout the text ($s = 10^5$ m$^2 \cdot$ nm $\cdot$ photons $\cdot$ sr $\cdot$ kW$^{-1}$, $T = 4000$K).

It makes sense to use both candidate models to generate synthetic replicate datasets, since ostensibly both are plausible candidates for the true data generating process. This is also an effective way to ensure that calibration curves cover both high and low values of $B^{\text{EMD}}$.

To generate each replicate dataset $\mathcal{D}_{\text{test}}^{\text{rep}}$, we select

$$\mathcal{B}_{\text{phys}} \sim \text{Unif}\left(\{\mathcal{B}_P, \mathcal{B}_{RJ}\}\right) \tag{55}$$

$$\mathcal{B}_0 \sim \text{Unif}\left([-10^{-4}, 10^{-4}]\right) \cdot \text{m}^2 \cdot \text{nm} \cdot \text{photons} \cdot \text{sr} \cdot \text{kW}^{-1}. \tag{56}$$

We then generate 4096 data points for each dataset using the Poisson observation model of equation (38), substituting $\mathcal{B}_{\text{phys}}$ for $\mathcal{B}_P$.

As in the main text, both candidate models assume Gaussian noise, so for each replicate dataset and each candidate model we fit the standard deviation of that noise to the observation data by maximum likelihood.

We then compute $B_{P,RJ;c}^{\text{EMD}} \in [0, 1]$ for each value of $c$ being tested. We also compute a binary variable—represented again with the Iverson bracket—which is 1 if the true risk of $\mathcal{M}_P$ is lower risk than that of $\mathcal{M}_{RJ}$, and 0 otherwise:

$$[R_P < R_{RJ}] := \begin{cases} 1 & \text{if } R_P < R_{RJ}, \\ 0 & \text{otherwise.} \end{cases} \tag{57}$$

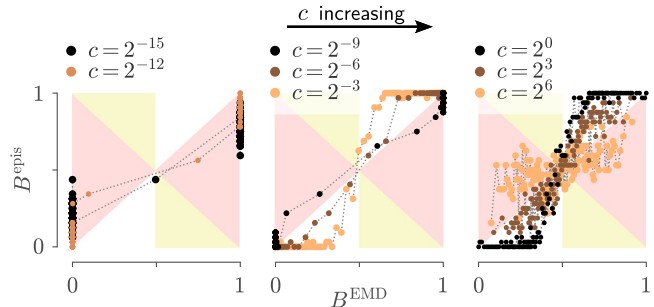

**Fig. 9 | Calibration plots for the $B_{P,RJ}^{EMD}$ criterion comparing Planck and Rayleigh-Jeans models.** The value of the sensitivity factor $c$ (see equations 16, 27) increases from left to right and dark to bright. Each curve consists of 4096 experiments aggregated into 128 bins. Wide gaps between points indicate that few experiments produced similar values of $B^{EMD}$; this is especially salient in the left panel, where almost all experiments have either $B^{EMD} = 0$ or $B^{EMD} = 1$. The two black body radiation models being compared are given by equations (36) to (38); the computation of $B^{epis}$ for these models is described in the Methods.

(In practice the true risks $R_P$ and $R_{RJ}$ are computed as the empirical risk with a very large number of data points.)

This produces a long list of $\left( B_{P,RJ;c}^{EMD}, [R_P < R_{RJ}] \right)$ pairs, one for each dataset $\mathcal{D}_{test}^{rep}$. Formally, we can then compute $B_{P,RJ;\Omega}^{epis}(B_{P,RJ;c}^{EMD})$ as per equation (54): by keeping only those pairs with matching $B_{P,RJ;c}^{EMD}$ value and averaging $B_{P,RJ}^{\infty}$. In practice, all $B_{P,RJ;c}^{EMD}$ values will be slightly different, so we bin neighbouring values together into a histogram. To make best use of our statistical power, we assign the same number of pairs to each bin, which means that bins have different widths; the procedure is as follows:

- Sort all $\left( B_{P,RJ;c}^{EMD}, [R_P < R_{RJ}] \right)$ pairs according to $B_{P,RJ;c}^{EMD}$.
- Bin the experiments by combining those with similar values of $B_{P,RJ;c}^{EMD}$. Each bin should contain a similar number of experiments, so that they have similar statistical power.
- Within each bin, average the values of $B_{P,RJ;c}^{EMD}$ and $[R_P < R_{RJ}]$, to obtain a single pair $(\bar{B}_{P,RJ;c}^{EMD}, \bar{B}_{P,RJ}^{epis})$. These are the values we plot as calibration curves.

The unequal bin widths are clearly visible in Fig. 9, where points indicate the $(\bar{B}_{P,RJ;c}^{EMD}, \bar{B}_{P,RJ}^{epis})$ pair corresponding to each bin. Larger gaps between points are indicative of larger bins, which result from fewer experiments having $B^{EMD}$ values close to $\bar{B}^{EMD}$.

Figure 9 is also a particularly clear illustration of the effect of the sensitivity factor $c$. When $c$ is too small ($c \lesssim 2^{-9}$), $R$-distributions almost never overlap, and the $B_{P,RJ;c}^{EMD}$ is mostly either 0 or 1: the criterion is *overconfident*, with many points located in the red regions. As $c$ increases ($2^{-6} \lesssim c \lesssim 2^3$), $B_{P,RJ;c}^{EMD}$ values become more evenly distributed over the entire [0, 1] interval and either don't encroach at all into the overconfident regions, or do so sparingly. These curves show a nearly monotone relation between $B_{P,RJ;c}^{EMD}$ and $B_{P,RJ;\Omega}^{epis}$, indicating that the former is strongly predictive of the latter; the corresponding $c$ values would likely be suitable for model comparison. In contrast, when $c$ is too large ($c \gtrsim 2^6$), $B^{EMD}$ becomes decorrelated from $B^{epis}$: all points are near $B^{epis} \approx 0.5$, indicating that the $B_{P,RJ;c}^{EMD}$ for such large values of $c$ would not be a useful comparison criterion.

**Calibration for the neural response models.** For the neuron model, we consider four sources of experimental variability: variations in the distribution used to model observation noise $\xi$, variations in the strength ($\sigma_o$) of observation noise, as well as variations in the strength ($\sigma_i$) and correlation time ($\tau$) of the external input $I_{ext}$. Each epistemic

distribution $\Omega$ is therefore described by four distributions over hyperparameters:

*Observation noise model* One of *Gaussian* or *Cauchy*. The distributions are centred and $\sigma_o$ is drawn from the distribution defined below. Note that this is the model used to generate a dataset $\mathcal{D}_{test}^{rep}$. The candidate models always evaluate their loss assuming a Gaussian observation model.

$$\text{Gaussian} \qquad\qquad \text{Cauchy}$$
$$p(\xi) = \frac{1}{\sqrt{2\pi}\sigma} \exp\left(-\frac{\xi^2}{2\sigma_o^2}\right) \qquad p(\xi) = \frac{2}{\pi\sigma\left[1 + \left(\frac{\xi^2}{\sigma_o/2}\right)\right]}$$

*Observation noise strength $\sigma_o$*

Low noise:      $\log \sigma_o \sim \mathcal{N}\left(0.0\text{mV}, (0.5\text{mV})^2\right)$

High noise:     $\log \sigma_o \sim \mathcal{N}\left(1.0\text{mV}, (0.5\text{mV})^2\right)$

*External input strength $\sigma_i$* The parameter $\sigma_i$ sets the strength of the input noise such that $\langle I_{ext}^2 \rangle = \sigma_i^2$.

Weak input:     $\log \sigma_i \sim \mathcal{N}\left(-15.0\text{mV}, (0.5\text{mV})^2\right)$

Strong input:   $\log \sigma_i \sim \mathcal{N}\left(-10.0\text{mV}, (0.5\text{mV})^2\right)$

*External input correlation time $\tau$* The parameter $\tau$ sets the correlation time of the input noise such that $\langle I_{ext}(t)I_{ext}(t+s) \rangle = \sigma_i^2 e^{-s^2/2\tau^2}$.

Short correlation:    $\log_{10} \tau \sim \text{Unif}([0.1\text{ms}, 0.2\text{ms}])$

Long correlation:     $\log_{10} \tau \sim \text{Unif}([1.0\text{ms}, 2.0\text{ms}])$

We thus defined 2 statistical distributions for $\xi$, 2 distributions for $\sigma_o$, 2 distributions for $\tau$, and 2 distributions for $\sigma_i$. Combined with three model pairs (see Fig. 6), this makes a total of $48 = 3 \times 2 \times 2 \times 2 \times 2$ possible epistemic distributions $\Omega$, each of which can be identified by a tuple such as $(\mathcal{M}_A \text{ vs } \mathcal{M}_B,$ Cauchy, Low noise, Strong input, Long correlation$)$. Calibration results for each of these conditions are given in Supplementary Fig. 1.

During calibration against an epistemic distribution $\Omega$, drawing a dataset $\mathcal{D}_{test}^{rep}$ happens in two steps. First, we randomly draw a vector $\omega$ of epistemic parameters (i.e., hyperparameters) from $\Omega$; for example $\omega = (\underbrace{-0.27\text{mV}}_{\sigma_o}, \underbrace{47\text{ms}}_{\tau}, \underbrace{0.003\text{mV}}_{\sigma_i})$. Second, we use those parameters to generate the dataset $\mathcal{D}_{test}^{rep}$, composed in this case of data points $(t, \tilde{V}^{LP}(t))$. We can then evaluate the loss $Q$ (given by equation (9)) on those data points. Note that the loss does not depend on the epistemic parameters $\omega$ directly, but in general will involve parameters which are fitted to the simulated data. In short, the vector $\omega$ describes the parameters of a simulated experiment, while $\mathcal{D}_{test}^{rep}$ describes a particular outcome of that experiment. In theory we could generate multiple datasets with the same vector $\omega$, but in practice it is more statistically efficient to draw a new experiment for each new dataset.

When choosing epistemic distributions, it is worth remembering that the goal of calibration is to empirically approximate a probability over experimental conditions. Thus choosing a distribution which can generate a large number of conditions—ideally an infinite number—will lead to better estimates. Here the use of continuous distributions for $\sigma_o$, $\sigma_i$ and $\tau$, and the fact that $I_{ext}$ is a continuous process, helps us achieve this goal. Also important is to generate data where the model selection problem is ambiguous, so as to resolve calibration curves around $B^{EMD} = 0.5$.

This calibration is an imperfect procedure, and how well it works depends on the quality of the candidate models and the choice of loss function. Here for example, the choice of a pointwise loss makes it sensitive to the timing of spikes; this tends to favour models which produce fewer spikes, since the penalty on a mistimed spike is high. This is why in Fig. 6, in the $\mathcal{M}_A$ vs $\mathcal{M}_D$ comparison, we see a floor on the

values of $B_{AD}^{epis}$. In short, some models have consistently lower loss even on random data, and so their risk—which is the expectation of their loss —is a priori lower. The bias we see in the $\mathcal{M}_A$ vs $\mathcal{M}_B$ comparison is likely due to a similar effect. (In this case because $\mathcal{M}_B$ is slightly better than $\mathcal{M}_A$ on average.)

## The hierarchical beta process

In this work we identify the epistemic uncertainty of a model $\mathcal{M}_A$ with the variability of a stochastic process $\mathfrak{Q}_A$: realisations of $\mathfrak{Q}_A$ approximate the PPF of the model loss. In our Results we listed desiderata which $\mathfrak{Q}_A$ should satisfy and proposed that it be described as a hierarchical beta (HB) process. However we deferred providing a precise definition for $\mathfrak{Q}_A$; we do this now in the form of Algorithm 2. The rest of this section explains the theoretical justifications for each step of this generative algorithm for $\mathfrak{Q}_A$.

**Algorithm 2.** Hierarchical beta process

Given
- $q_A^*$, $c$ and $\delta_A^{EMD}$, ▷ *computed from data*
- $N \in \mathbb{N}^+$, ▷ *number of refinements*
- $p(\hat{q}(0), \hat{q}(1))$, ▷ *2-d distribution over end points*

generate a discretized realisation $\hat{q}_A$.

**Procedure:**
▷ *Initialise the procedure by drawing end point*
1: **repeat**
2: **draw** $(\hat{q}(0), \hat{q}(1)) \sim p(\hat{q}(0), \hat{q}(1))$
3: **Until** $\hat{q}(0) < \hat{q}(1)$ ▷ *PPFs must be increasing*
▷ *Successively refine the interval*
4: **for** $n \in 1, 2, ..., N$ **do** ▷ *refinement levels*
5: **for** $\Phi \in \{k \cdot 2^{-n+1}\}_{k=0}^{2^{n-1}-1}$ **do** ▷ *intermediate incrs.*
6: **compute** $r, v$ according to equations (73)
7: **solve** equations (75) to obtain $\alpha$ and $\beta$
8: **draw** $x_1 \sim \text{Beta}(\alpha, \beta)$
9: $\hat{q}(\Phi + 2^{-n}) \leftarrow \hat{q}(\Phi) + \Delta\hat{q}_{\Delta\Phi}(\Phi) \cdot x_1$
10: **end for**
11: **end for**
12: **return** $(\hat{q}(\Phi) : \Phi \in \{\mathcal{I}_\Phi\}^{(N)})$

The quantities $r$ and $v$ computed in Algorithm 2 conceptually represent the **r**atio between two sucessive increments and the **v**ariance of those increments.

### Relevant concepts of Wiener processes.

Before introducing the HB process, let us first review a few key properties which stochastic processes must satisfy and which are covered in most standard introductions[70–72]. We use for this the well-known Wiener process, and also introduce notation which will become useful when we define the HB process. Since our goal is to define a process for PPFs, we use $\Phi$ to denote the independent "domain" variable and restrict ourselves to 1-d processes for which $\Phi \in [0, 1]$.

For the Wiener process $\mathcal{W}$, each realisation is a continuous function $W : [0, 1] \to \mathbb{R}$. One way to approximate a realisation of $\mathcal{W}$ is to first partition the interval into subintervals $[0, \Phi_1), [\Phi_1, \Phi_2), ..., [\Phi_n, 1)$ with $0 < \Phi_1 < \Phi_2 < \cdots < \Phi_n < 1$; for simplicity we will only consider equal-sized subintervals, so that $\Phi_k = k\,\Delta\Phi$ for some $\Delta\Phi \in \mathbb{R}$. We then generate a sequence of independent random increments (one for each subinterval) $\{\Delta W_{\Delta\Phi}(0), \Delta W_{\Delta\Phi}(\Delta\Phi), \Delta W_{\Delta\Phi}(2\Delta\Phi), ..., \Delta W_{\Delta\Phi}(1 - \Delta\Phi)\}$ and define the corresponding realisation as

$$W(k\,\Delta\Phi) = W(0) + \sum_{l=0}^{k-1} \Delta W_{\Delta\Phi}(l\,\Delta\Phi) \quad (58)$$

(Within each interval the function may be linearly interpolated, so that $W$ is continuous.)

A *refinement* of a partition is obtained by taking each subinterval and further dividing it into smaller subintervals. For instance we can refine the unit interval $[0, 1)$ into a set of two subintervals, $[0, 2^{-1})$ and $[2^{-1}, 2^{-0})$. Let us denote these partitions $\{\mathcal{I}_\Phi\}^{(0)}$ and $\{\mathcal{I}_\Phi\}^{(1)}$ respectively. Repeating the process on each subinterval yields a sequence of ever finer refinements:

$$
\begin{aligned}
\{\mathcal{I}_\Phi\}^{(0)} &:= \left\{ \left[0, 2^{-0}\right) \right\} \\
\{\mathcal{I}_\Phi\}^{(1)} &:= \left\{ \left[0, 2^{-1}\right), \left[2^{-1}, 2^{-0}\right) \right\} \\
\{\mathcal{I}_\Phi\}^{(2)} &:= \left\{ \left[0, 2^{-2}\right), \left[2^{-2}, 2^{-1}\right), \left[2^{-1}, 3 \cdot 2^{-2}\right), \left[3 \cdot 2^{-2}, 2^{-0}\right) \right\} \\
&\;\;\vdots \\
\{\mathcal{I}_\Phi\}^{(n)} &:= \left\{ \left[k \cdot 2^{-n}, (k+1) \cdot 2^{-n}\right) \right\}_{k=0}^{2^n-1}.
\end{aligned}
\quad (59)
$$

With these definitions, for any $m \geq n$, $\{\mathcal{I}_\Phi\}^{(m)}$ is a refinement of $\{\mathcal{I}_\Phi\}^{(n)}$.

Later we will need to refer to the vector of new end points introduced at the $n$-th refinement step. These are exactly the odd multiples of $2^{-n}$ between 0 and 1, which we denote $\{\Phi\}^{(n)}$:

$$\{\Phi\}^{(n)} := \left(2^{-n}, 3 \cdot 2^{-n}, ..., (2^n - 3) \cdot 2^{-n}, (2^n - 1) \cdot 2^{-n}\right). \quad (60)$$

Any random process must be *self-consistent*[41]: for small enough $\Delta\Phi$, the probability distribution at a point $\Phi$ must not depend on the level of refinement. For example, the Wiener process is defined such that the increments $\Delta W_{\Delta\Phi} \sim \mathcal{N}(0, \Delta\Phi)$ are independent; therefore

$$
\begin{aligned}
&W(\Phi + 2\Delta\Phi) \\
&= W(\Phi) + \Delta W_{2\Delta\Phi}(\Phi) = W(\Phi) + \Delta W_{\Delta\Phi}(\Phi) + \Delta W_{\Delta\Phi}(\Phi + \Delta\Phi) \\
&= W(\Phi) + \mathcal{N}(0, 2\Delta\Phi) = W(\Phi) + \mathcal{N}(0, \Delta\Phi) + \mathcal{N}(0, \Delta\Phi).
\end{aligned}
\quad (61)
$$

It turns out that the combination of the Markovian and self-consistent properties set quite strong requirements on the stochastic increments, since they impose the square root scaling of the Wiener increment: $\mathcal{O}(\Delta W_{\Delta\Phi}) = \mathcal{O}(\sqrt{\Delta\Phi})$. (§II.C of reference [41].)

While the Wiener process underlies much of stochastic theory, it is not suitable for defining a process $\mathfrak{Q}$ over PPFs. Indeed, it is not monotone by design, which violates one of our desiderata. Moreover, it has a built-in directionality in the form of accumulated increments. A clear symptom of this is that as $\Phi$ increases, the variance of $W(\Phi)$ also increases. (This follows immediately from equation (58) and the independence of increments.) Directionality makes sense if we think of $W$ as modelling the diffusion of particles in space or time, but empirical PPFs are obtained by first sorting data samples according to their loss (see the definition of $\delta^{EMD}$ in the Results). Since samples of the loss arrive in no particular order, a process which samples PPFs should likewise have no intrinsic directionality in $\Phi$.

### A hierarchical beta distribution is monotone, non-accumulating and self-consistent.

Constructing a stochastic process $\mathfrak{Q}$ which is *monotone* is relatively simple: one only needs to ensure that the random increments $\Delta\hat{q}_{\Delta\Phi}(\Phi)$ are non-negative.

Ensuring that those increments are *non-accumulating* requires more care, because that requirement invalidates most common definitions of stochastic processes. As described in our Results, we achieve this by defining $\mathfrak{Q}$ as a sequence of refinements, starting from a single increment for the entire interval, then doubling the number of increments (and halving their width) at each refinement step. In the rest of this subsection we give an explicit construction of this process and show that it is also *self-consistent*. (Although in this work we consider only pairs of increments sampled from a beta distribution, in general one could consider other compositional distributions. Higher-

dimensional distributions may allow to sample all increments simultaneously, if one can determine the conditions which ensure self-consistency.)

For an interval $\mathcal{I} = [\Phi, \Phi + \Delta\Phi)$, we suppose that the points $\hat{q}_A(\Phi)$ and $\hat{q}_A(\Phi + \Delta\Phi)$ are given. We define

$$\Delta\hat{q}(\mathcal{I}) = \Delta\hat{q}_{\Delta\Phi}(\Phi) := \hat{q}_A(\Phi + \Delta\Phi) - \hat{q}_A(\Phi), \tag{62}$$

then we draw a subincrement $\Delta\hat{q}_{\frac{\Delta\Phi}{2}}(\Phi)$, associated to the subinterval $[\Phi, \Phi + \Delta\Phi)$, from a scaled beta distribution:

$$\frac{1}{\Delta\hat{q}_{\Delta\Phi}(\Phi)}\Delta\hat{q}_{\frac{\Delta\Phi}{2}}(\Phi) := x_1 \sim \text{Beta}(\alpha, \beta). \tag{63}$$

(Refer to the subsection below for the definition of $\text{Beta}(\alpha, \beta)$.) The scaling is chosen so that

$$0 \le \underbrace{\Delta\hat{q}_{\frac{\Delta\Phi}{2}}(\Phi)}_{= x_1\Delta\hat{q}_{\Delta\Phi}(\Phi)} \le \Delta\hat{q}_{\Delta\Phi}(\Phi). \tag{64}$$

The value of $\Delta\hat{q}_{\frac{\Delta\Phi}{2}}(\Phi)$ then determines the intermediate point:

$$\hat{q}_A\left(\Phi + \tfrac{\Delta\Phi}{2}\right) \leftarrow \hat{q}_A(\Phi) + \Delta\hat{q}_{\frac{\Delta\Phi}{2}}(\Phi). \tag{65}$$

If desired, the complementary increment $\Delta\hat{q}_{\frac{\Delta\Phi}{2}}(\Phi + \frac{\Delta\Phi}{2})$ can be obtained as

$$\Delta\hat{q}_{\frac{\Delta\Phi}{2}}\left(\Phi + \tfrac{\Delta\Phi}{2}\right) = (1 - x_1)\Delta\hat{q}_{\Delta\Phi}(\Phi). \tag{66}$$

Generalising the notation to the entire $[0, 1)$ interval, we start from a sequence of increments associated to subintervals at refinement step $n$ (recall (59)):

$$\begin{aligned}4\{\Delta\hat{q}_{2^{-n}}\} &:= \left\{\Delta\hat{q}(\mathcal{I}) : \mathcal{I} \in \{\mathcal{I}_\Phi\}^{(n)}\right\} \\ &\stackrel{\text{def}}{=} \left\{\Delta\hat{q}_{2^{-n}}(\Phi) : [\Phi, \Phi + 2^{-n}) \in \{\mathcal{I}_\Phi\}^{(n)}\right\}.\end{aligned}$$

Applying the procedure just described, for each subinterval we draw $x_1$ and split the corresponding increment $\Delta\hat{q}_{2^{-n}}(\Phi) \in \{\Delta\hat{q}_{2^{-n}}\}$ into a pair $\left(\Delta\hat{q}_{2^{-n-1}}(\Phi), \Delta\hat{q}_{2^{-n-1}}(\Phi + 2^{-n-1})\right)$ of subincrements such that

$$\Delta\hat{q}_{2^{-n}}(\Phi) = \Delta\hat{q}_{2^{-n-1}}(\Phi) + \Delta\hat{q}_{2^{-n-1}}\left(\Phi + 2^{-n-1}\right). \tag{67}$$

The union of subincrements is then the next refinement step:

$$\left\{\Delta\hat{q}(\mathcal{I}) : \mathcal{I} \in \{\mathcal{I}_\Phi\}^{(n+1)}\right\} = \{\Delta\hat{q}_{2^{-n-1}}\}. \tag{68}$$

After $n$ refinement steps, we thus obtain a function $\hat{q}(\Phi)$ defined at discrete points:

$$\hat{q}^{(n)}(k \cdot 2^{-n}) := \hat{q}(0) + \sum_{l < k} \Delta\hat{q}_{2^{-n}}(l \cdot 2^{-n}), \tag{69}$$

which we extend to the entire interval $[0, 1)$ by linear interpolation; see Fig. 5d for an illustration. In practice we found that computations (specifically the risk computed by integrating $\hat{q}^{(n)}(\Phi)$) converge after about eight refinement steps.

This procedure has the important property that once a point is sampled, it does not change on further refinements:

$$\hat{q}^{(n)}(k \cdot 2^{-n}) = \hat{q}^{(m)}(k \cdot 2^{-n}), \quad \forall m \ge n, \tag{70}$$

which follows from equation (67). Recall now that, as stated above, a process is self-consistent if "for small enough $\Delta\Phi$, the probability distribution at a point $\Phi$ [does] not depend on the level of refinement".

Since equation (70) clearly satisfies that requirement, we see that the process obtained after infinitely many refinement steps is indeed self-consistent. We thus define the *hierarchical beta (HB) process* $\mathfrak{Q}_A$ as

$$p(\hat{q} \mid \mathfrak{Q}_A) := p\left(\hat{q} = \lim_{n\to\infty} \hat{q}^{(n)} \mid c, q_A^*, \delta_A^{\text{EMD}}\right). \tag{71}$$

To complete the definition of $\mathfrak{Q}_A$, we need to specify how we choose the initial end points $\hat{q}_A(0)$ and $\hat{q}_A(1)$. In our implementation, they are drawn from normal distributions $\mathcal{N}(q^*(\Phi), \sqrt{c}\,\delta_A^{\text{EMD}}(\Phi)^2)$ with $\Phi \in \{0, 1\}$, where again $c$ is determined via our proposed calibration procedure; this is simple and convenient, but otherwise arbitrary. We also need to explain how we choose the beta parameters $\alpha$ and $\beta$, which is the topic of the next subsection.

**Choosing beta distribution parameters.** All HB processes are monotone, continuous and self-consistent, but within this class there is still a lot of flexibility: since $\alpha$ and $\beta$ are chosen independently for each subinterval, we can mould $\mathfrak{Q}_A$ into a wide variety of statistical shapes. We use this flexibility to satisfy the two remaining desiderata: a) that realisations $\hat{q}_A(\Phi)$ track $q_A^*(\Phi)$ over $\Phi \in [0, 1]$; and b) that the variability of $\hat{q}_A(\Phi)$ be proportional to $\delta_A^{\text{EMD}}(\Phi)$. It is the goal of this subsection to give a precise mathematical meaning to those requirements.

Let $x_1 \sim \text{Beta}(\alpha, \beta)$ and $x_2 = 1 - x_1$. (The density function of a beta distribution is given in (24).) The mean and variance of $x_1$ are

$$\begin{aligned}\mathbb{E}[x_1] &= \frac{\alpha}{\alpha + \beta}, \\ \mathbb{V}[x_1] &= \frac{\alpha\beta}{(\alpha + \beta)^2(\alpha + \beta + 1)}.\end{aligned} \tag{72}$$

For a given $\Phi$, it may seem natural to select $\alpha$ and $\beta$ by matching $\mathbb{E}[x_1]$ to $q_A^*(\Phi)$ and $\mathbb{V}[x_1]$ to $c \cdot \left(\delta_A^{\text{EMD}}(\Phi)\right)^2$. However both equations are tightly coupled, and we found that numerical solutions were unstable and unsatisfactory; in particular, it is not possible to make the variance large when $\mathbb{E}[x_1]$ approaches either 0 or 1 (otherwise the distribution of $x_1$ would exceed $[0, 1]$).

Much more practical is to consider moments with respect to the Aitchison measure; as mentioned in the Results, the Aitchison measure first maps the bounded interval $[0, 1]$ to the unbounded space $\mathbb{R}$ with a logistic transformation, then computes moments in the unbounded space. The first two such moments are called the *centre* and the *metric variance*[42,43]; for the beta distribution, they are given by (reproduced from 25)

$$\mathbb{E}_a[(x_1, x_2)] = \frac{1}{e^{\psi(\alpha)} + e^{\psi(\beta)}}\left(e^{\psi(\alpha)}, e^{\psi(\beta)}\right) \tag{25a}$$

$$\text{Mvar}[(x_1, x_2)] = \frac{1}{2}\left(\psi_1(\alpha) + \psi_1(\beta)\right), \tag{25b}$$

where $\psi$ and $\psi_1$ are the digamma and trigamma functions respectively. The centre and metric variance are known to be more natural statistics for compositional distributions, and this is what we found in practice. Therefore we will relate $q_A^*$ to the centre, and $\sqrt{c}\,\delta_A^{\text{EMD}}$ to the square root of the metric variance.

To be precise, suppose that we have already selected a set of increments $\{\Delta\hat{q}_{2^{-n}}\}$ over the domain $\{k \cdot 2^{-n}\}_{k=0}^{n-1}$, and wish to produce the refinement $\{\Delta\hat{q}_{2^{-n-1}}\}$. For each $\Phi$ in $\{k \cdot 2^{-n}\}_{k=0}^{n-1}$ we define

$$\begin{aligned}r &:= \frac{q_A^*(\Phi + 2^{-n-1}) - q_A^*(\Phi)}{q_A^*(\Phi + 2^{-n}) - q_A^*(\Phi + 2^{-n-1})}, \\ v &:= 2c\left(\delta_A^{\text{EMD}}(\Phi + 2^{-n-1})\right)^2.\end{aligned} \tag{73}$$

The value $r$ is the ratio of subincrements of $\Delta q^*_{2^{-n}}(\Phi)$. Since we want $\hat{q}_A$ to track $q^*_A$, it makes sense to expect $r$ to also approximate the ratio of subincrements of $\Delta \hat{q}_{2^{-n}}(\Phi)$. Identifying

$$r \leftrightarrow \frac{\mathbb{E}_a[x_1]}{\mathbb{E}_a[x_2]} \quad \text{and} \quad v \leftrightarrow 2 \cdot \text{Mvar}\left[(x_1, x_2)\right], \tag{74}$$

and substituting into equation 25 then leads to the following system of equations:

$$\begin{aligned} \psi(\alpha) - \psi(\beta) &= \ln r \\ \ln\left[\psi_1(\alpha) + \psi_1(\beta)\right] &= \ln v, \end{aligned} \tag{75}$$

which we can solve to yield the desired parameters $\alpha$ and $\beta$ for each subinterval in $\{\mathcal{I}_\Phi\}^{(n)}$.

In summary, although the justification is somewhat technical, the actual procedure for obtaining $\alpha$ and $\beta$ is quite simple: first compute $r$ and $v$ following equations (73), then solve equations (75).

## Data availability

All figure data generated in this study can be regenerated from code (see Code availability). Results of especially long computations are included as precomputed data files alongside the accompanying computational notebooks[73].

## Code availability

All source code used to produce the figures in this paper is available as a collection of Jupyter notebooks[73]. Additional Python code for simulating the neuron circuit model[66] and generating coloured noise[67] is also available. We also provide the Python package `emdcmp`[74], available on the Python Packaging Index (PyPI), which automates many of the computation steps in our approach: given a set of observed data $\mathcal{D}_{\text{test}}$, a generative model $\mathcal{M}_A$, a loss function $Q$ and sensitivity parameter $c \in \mathbb{R}$, `emdcmp` will automatically compute $\tilde{q}_A$, $q^*_A$ and $\delta^{\text{EMD}}_A$, then draw samples from the corresponding $R_A$-distribution. In other words, through `emdcmp` we provide a standard implementation for all the steps represented by downward facing arrows on the right of Fig. 1. The package also provides utilities to help execute calibration experiments.

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

## Acknowledgements

We thank Anno Kirth, Aitor Morales-Gregorio, Günther Palm, Moritz Helias, Jan Bölts, Abel Jansma, Thomas Nowotny and Manfred Opper for helpful comments and discussions. This work was partly supported by the German Federal Ministry for Education and Research (BMBF grant no. 01IS19077B; AR), the Natural Sciences and Engineering Research Council of Canada (NSERC; AL), the government of Ontario (OGS; AR), and the European Union (ERC grant "HIGH-HOPeS", no. 101039827; AR).

## Author contributions

A.R. developed the theory, wrote the software implementations, and the first draft of the manuscript. A.R. and A.L. discussed the results and revised the manuscript.

## Funding

## Competing interests

The authors declare no competing interests.
