## [Transparent Peer Review file · Nature Communications]

Selecting fitted models under epistemic uncertainty using a stochastic process on quantile functions

Corresponding Author: Mr Alexandre René

Version 0:

Reviewer comments:

Reviewer #1

(Remarks to the Author)

I sympathize with the author for their ambitious attempt to nearly invent a new framework for model comparison, and it is nearly orthogonal to existing statistical tools. The proposed method is likely useful when comparing many physical models, especially when none is correct. However, after reading the paper, the question I find myself asking is whether the effort of constructing such a new framework and the overhead of introducing new concepts are fully justified by their additional practical or pedagogical merits over classical methods.

I find the writing difficult to follow. I am often overwhelmed by the introduction of numerous new notions and technical terminology. The key method is not introduced until roughly page 10. Even after reading the paper, some key information seems missing: for example, is each candidate model fitted by MLE or by Bayesian inference? The description of the replication process on pages 10-11 is especially confusing. I could make guesses to fill these gaps, but I would prefer that the author clarify the writing and make it more concise.

I think there are at least two unstated assumptions of the method: (1) The data points come in IID pairs, and (2) The sample size is assumed to be "big enough" such that we can ignore any finite sample/statistical uncertainty. Both assumptions are questionable, and arguably the finite sample behavior of model selection or model comparison is crucial to this field. For example, the proposed method may fail to account for "overfitting." When the candidate models are nested, it appears that the more complex model will always have the smaller in-sample risk, and thereby be picked by the proposed method. Perhaps I am misunderstanding the motivation of this work, but if the sample size is assumed to be infinite, the sample mean of empirical risk would already be good enough for model comparison.

The concept of aleatoric/epistemic uncertainty seems overused in the paper and does not fully clarify the proposed method. I must note that what the author refers to as "Aleatoric uncertainty model companions" feels somewhat like a strawman argument to me. Take the example of Fig 4 panel A: the more accepted model comparison approach is to compare the pointwise risk among four models. If we take model A as a benchmark and visualize the empirical distribution of point-wise excess risk of models B, C, and D, this would resemble the difference between a paired t-test and a t-test. This approach would eliminate a large portion of the sample variation. Ref [2] discusses more on uncertainty in model comparison.

In my opinion, the idea of "empirical model discrepancy" is not completely new. It seems closely related to posterior predictive checks (Ref [1]), where the test function is limited to the risk quantiles. This paper does make an extension, in that the classical posterior predictive check is only used for model checks, while here it is used for model comparison.

Overall, I can see the potential of this tool. But I am not yet convinced that existing tools cannot address the issue raised in this paper. A minimal illustration should include a comparison (in testing power or model selection accuracy) between this new tool and existing tools (mean of empirical risk, bootstrap, cross-validation, Bayesian posterior predictive check). Criticizing all existing tools as "aleatoric" does not make the newly proposed tool immune from comparison.

[1] POSTERIOR PREDICTIVE ASSESSMENT OF MODEL FITNESS VIA REALIZED DISCREPANCIES

[2] Uncertainty in Bayesian Leave-One-Out Cross-Validation Based Model Comparison

(Remarks on code availability)

Reviewer #2

(Remarks to the Author)

This is a review of the manuscript entitled: "Epistemically robust selection of the fitted models".

As a note to authors, this review is written in good faith with the goal to improve the scientific work presented in the manuscript, and trying to indicate its merits and shortcomings. As a reference, my formal training is in statistics and I work in theoretical statistics, philosophy of science, and metascience. I mainly develop theories on reproducibility, replicability, and statistical methodologies to test how sciences progress.

The manuscript presents a novel method of model selection with the claim that it is epistemically robust. The key methodological improvement to me is the work that was done on separating and quantifying the model uncertainty and sampling uncertainty in statistical terms. The method has its commended merits due to this development. In separating and quantifying these two sources of uncertainty the method uses a clever modeling approach and fundamental quantities such as quantiles, which are all positives. Using risk as a desirable criterion will alienate somewhat information-theory grounded people, but it is a valid approach and one of the best available approaches that we know, so I am fine with this.

As a personal note, I really liked the methodological development, especially the modeling of stochastic process on quantile functions. This is all interesting solid work and with simple enough concepts to boot.

On the other hand, I have many issues with the motivation and presentation of the work. I am afraid my review might sound mostly critical in a negative way, but I want to clarify the reasons. I have nothing against the method developed in the manuscript. In fact, I enjoyed reading it, it piqued my curiosity, and it motivated me to thinking about challenging aspects of model selection. My criticisms mainly revolve around to the following.

1. The way the idea is motivated with a misleading/misinformed perspective taken with respect to our contemporary understanding of science/statistics,
2. Unfair comparisons with other existing statistical methods,
3. Overreaching and overgeneralization of the merits of the method.

These three factors weigh down what otherwise would be a good methodological contribution. The presentation dives into considerably loaded issues unnecessarily. My suggestion is to present the method as what it is, a model selection procedure with potentially some good properties, compare/contrast it with state of the art model selection procedures and leave it at that. In particular, I suggest to stay away from the idea of "falsification" and overemphasizing "convergence" (what is known as consistency in statistics).

Regarding point 1 above.

The manuscript motivates the method presented starting with a reference to Tarantola's 95 work. This is mostly a philosophy of science work. There is a long way from the issues raised by Tarantola to the solution of these issues (if indeed, they exist). These issues mostly clash with our current statistical knowledge and model selection procedures (whether consistent or not) are not capable of solving these. This is why, the idea of "falsification" is almost entirely abandoned in contemporary philosophy of science and statistics. Falsification is fraught with difficulties that current state of the arts cannot circumvent. Toward the end of the manuscript even Popper and Eddington observations (seems to be a recurrent pet-topic with physics in general) were dragged into the discussion, without a proper treatment of what they represent. Popper never accepted induction. Popper's falsification idea applies to hypotheses that can be decided based on a finite number of observations (which is not what model selection is about). Eddington observations are a type of Newton's experimentum crucis, where a single observation/study is sufficient to distinguish between two competing hypotheses, verifying the predictions of an extremely well-developed physics theory. Analyzing data statistically with the goal of model selection from a candidate set of models (that in the first place we have chosen) is a very different exercise. Statistical consistency --selecting the true model in the limit as sampling uncertainty goes to zero-- is a desirable property from statistical perspective, but equating this with falsification of theories is a naive interpretation of Popper's work at best, dismissing all the current development in philosophy of science in recent decades.

Regarding point 2 above. The proposed method is compared against some existing model selection criteria. To me it sounds that consistency of a model selection method is the most important criterion for the authors from repeated references to it in the text. I could not shake-off the feeling that all the negative aspects of the existing --classical-- criteria were exposed to make the proposed method look better (again, I think that the proposed method has merits but this is a poor way of going about showing them). In fact, over the decades there have been a number of improvements of classical model selection

criteria that surpass the classical usage of these methods. With respect to the current manuscript criticizing these, below are some examples.

i) One model selection criterion that authors make comparison with is AIC. They write that AIC is inconsistent (as n goes to infinity, it does not select the true model with probability 1), which is true. AIC was developed in 1978-1979. It was immediately apparent that this was not a consistent estimator of the true model. Schwarz has developed BIC around the same time, which is consistent (with an additional term that depends on the sample size). So, why not compare the performance of the proposed method with respect to BIC and beat AIC to death? Another state of the art model selection criteria is Minimum Description Length. This was not mentioned in the manuscript at all. It is worth noting that all these criteria for model selection are centered on the value of information brought in by the data and hence they are "information-theoretic" criteria. Even if they fail to satisfy the consistency criterion (which some of them do not) their purpose was not only to satisfy the consistency, but assessing how more information affects the model selection procedure. So, the comparisons are unfair. Along the same veins, we can ask how the proposed method of model selection in the manuscript evaluates information, for example measuring distances between distributions implied by models. I have nothing against using risk as a measure --this is the approach taken in the manuscript-- but taking the risk approach will be challenging to reconcile with the information-theory.

iii) Another model selection criterion that authors make comparison with is Bayes Factors. Bayes factors have a number of shortcomings and these are stated in the manuscript, but it was omitted that they converge to BIC and hence approximately consistent. Criticizing Bayes factors being affected by priors is also unfair. The whole Bayesian approach relies on prior information.

iv) Another method that was criticized is bootstrap. Bootstrap originally developed by Efron in 1978-1980, was not meant to be a model selection criterion. It is a method to estimate the sampling distributions of statistics. However, later --for example in the context of Bayesian bootstrap and other improvements-- it has been used as a model selection criterion. It was mentioned that bootstrap estimators are biased, which is true, but then, this again puts the consistency at the center as the sole criterion in model selection, neglecting the desirable property of quantifying and expressing uncertainty for finite sample sizes.

Regarding point 3 above. An important point is that statistical methods embrace uncertainty and aim to quantify it. Not to eliminate it, as seems to be the perspective taken in the tone of the manuscript. Practically, contemporary statistical view is that although consistency is a desirable property, we will never get infinite sample size in a realistic situation and that this is only a theoretical guide. Not the goal in developing statistical methods. I realized that authors have cited one of Andrew Gelman's papers. More of his work details about a holistic approach to statistical methods and analyses and not singlemindedly focusing on properties of a single method making it the focus of evaluating scientific theories. This is also echoed generally among philosophers and statisticians and is in stark contrast with the positioning of the method in the manuscript.

Other comments:

1. Abstract: "falsify" and "reject" are not used in the same sense in statistics. Rejection of a hypothesis does not mean falsifying a theory associated with it. Probably this was written in physics sense --and I can grudgingly accept that-- but at least I would appreciate if you can clarify it, perhaps with a footnote (since this is a multidisciplinary journal).

2. Line 9: "...non-identifiable:..." Hard-identifiability and soft identifiability have different meanings theoretically. Please clarify which one you are referring to.

3. Line 125-126: In a realistic model selection problem there will be more than two competing models. If you are comparing them pairwise, multiple use of the same data (using information twice) will be a problem that needs to be corrected statistically. This is a challenging problem, currently with no satisfactory solution and I do not require one, but the problem is worth acknowledging around lines 1215-126.

4. Lines 178-180, and 214-220: Do not sit well with me due to problems mentioned above.

5. Lines 257-258: Bayes risk and frequentist risk have very different interpretations. It is worth clarifying what equation (5) refers to. Looks like you are integrating over parameters to get to the unconditional expectation of the model which would imply Bayes risk, but this is unclear from the formula.

6. Line 388: Typo in "explicitely"

7. Line 435-436: epsilon is not "confidence". It seems to refer to type I error rate. Confidence is $100 \cdot (1 - \epsilon)$. I would not bring this into the discussion since this is a purely frequentist hypothesis testing criterion and has no business in model selection.

8. Line 478: Please describe "stationary distribution". It has a number of meanings. I assume you mean as time goes to infinity there is a distribution of interest that the process converges, but could not be sure.

9. Line 492: There are different definitions of Heaviside function, in particular for $H(0)$. Please clarify which one you are using.

10. Line 538: "light tails" should be "exponential tails" I think, since you are using Gaussian. Light tailed distributions are a separate class lighter than exponential tails.

11. Line 659: Equation (19a) has a comma on the right hand side. This does not makes sense to me, unless it is the bivariate nature of the expectation you are trying to capture with a single equation? Please add a brief explanation.

12. Line 896: "Is it 16 variations? or 18 variations? I think somewhere else in the paper it says 16?"

13. Line 939: Equation (34) is poorly written. The probabilities $P(A)$ and $P(B)$ are conditional on data, which makes all the difference to a Bayesian. Please use correct notation.

(Remarks on code availability)

Reviewer #3

(Remarks to the Author)

SUMMARY

In this manuscript, the authors propose a novel method called EMD (for Empirical modelling discrepancy) for model comparison. Their method construct a "risk distribution" for each model and then compute some distance measure between the two risk distributions as a proxy for the evidence for or against a given model compared to another one. The proposed method is contrasted with other model comparison methods.

The EMD method is original and the manuscript is overall well written. In my view, the biggest strengths of the paper are the novelty of the approach and the emphasis on the epistemic uncertainty (i.e. the uncertainty induced by modelling errors). However, the whole argumentation of the paper relies on few key points that that the authors fail to address convincingly.

MAJOR

1. What is the gold standard for model comparison? In the literature, there exists several model comparison methods. Even though it might be computationally intensive to evaluate, I believe the gold standard for model comparison remains the Bayesian Factor (if the prior assumptions are valid) and this for several reasons

a) first, it computes exactly what we are looking for, i.e. when the model priors are identical ($p(M_1) = p(M_2)$), then $BF = p(M_1|D)/p(M_2|D)$ which is the ratio of the posterior over the models given the data.

b) secondly, the integral needed to compute $p(D|M)$ naturally penalises for high complexity models

c) most importantly, it is consistent. See e.g. [1] and see point 2a below.

Curiously, the authors authors criticise the BF for with hardly convincing arguments:

i) on L 973, it is mentioned that $B^{\{Bayes\}}$ and $B^{\{elpd\}}$ provide extremely small values (on Table 2). It might be after all that the two models are effectively indistinguishable for the considered range of λ (this is actually the reason why Fig 8 does not display the range of wavelength considered in the testing of Fig 2 - precisely because we can't see the difference between the two models).

ii) on L 977, the BF is criticised for being dependent on L . I don't understand this critique. Any model comparison criterion has to depend on L .

iii) on 979, it is argued that a weakness of the BF is its strong dependence on the prior. Of course the BF depends on the prior - as it should. Now in the P and RJ models, the priors over λ and T are exactly the same for both models, so it is unclear what is the detrimental effect of the prior here.

Now even if the BF would be disregarded as the gold standard for model comparison, what should be the gold standard? If there is no reference, what is the point in proposing a new model comparison method since there is no way to argue how well it performs? Visual inspection should not be the criterion.

2. The proposed method is not principled. After having inappropriately criticised the BF for model comparison, the authors propose a new approach - which is ad-hoc for several reasons.

a) Firstly, and most importantly, no consistency proof is provided. The authors should formally show that if one of the candidate model is actually the true model, then the method should favor the true one. A theorem and a formal consistency proof is expected here.

b) Secondly, the EMD method depends on a free parameter c . Even though they propose some calibration method (see Eq

28 - which is actually even violated in the provided example, see Fig 7. See also minor comment 1 below), the presence of free parameters gives a sense of an ad-hoc method.

c) Thirdly, as recognised on line 786, the interpretation of $B^{\{EMD\}}$ is not very clear. Actually, the interpretability of $B^{\{EMD\}}$ only comes from another criterion (namely $B^{\{\text{epis}\}}$) with which it is correlated (provided that c has been chosen accordingly).

d) Finally, it is unclear if the proposed method performs well since there is no ground truth reference provided for the comparison (see point 1 above).

3. Overconfidence about overconfidence? The authors criticise existing model comparison methods for their presupposed overconfidence. According to them, overconfidence manifests itself in (a) the lack of saturation of the model comparison criterion when the sample size L increases and in (b) the non-robustness of the criterion (which could flip between strong evidence for and then against a model with small variations of a dataset). However, in order to make any statement about the overconfidence or underconfidence of a criterion, we need some ground truth reference for the confidence. In the absence of such ground truth, it is hard to interpret the authors' desiderata (a) and (b) as other than personal views.

4. Open questions in the Table 2. The model comparison is summarised in Table 2. However, there are still several open questions that need to be addressed.

a) The results displayed in Table 2 are hardly comparable. If all of them were log probability ratios, then the comparison would be fair, but this is not the case. The authors needed to "stretch" some definitions, but this is not very elegant and potentially misleading.

b) There are few $+\infty$ entries for the $B^{\{EMD\}}$ criterion. This is not a very good sign (isn't this a sign of overconfidence - or rather infinite confidence?). Furthermore, it is unclear why infinity should appear both in the low L and high L regimes (see $B^{\{EMD\}}$, last column, small λ).

c) Where do the zero entries come from for the $B^{\{elpd\}}$ criterion?

MINOR

1. Fig 7 all curves violate condition 28. Indeed, all the curves in Fig. 7 (and not only for the braun curve), have at least some part in the forbidden area (shaded area).

2. Eq 67 or Eq 36 want to stress the fact that some quantities are expressed in \log_{10} with the expression " $\log 10$ ". I find this writing style misleading as it could be wrongly interpreted as a division by $\log(10)$.

3. Eq 11. Shouldn't " $<$ " be replaced by " $>$ "?

REFERENCES

[1] Casella, George, and Elías Moreno. "Assessing Robustness of Intrinsic Tests of Independence in Two-Way Contingency Tables." *Journal of the American Statistical Association* 104, no. 487 (September 2009): 1261–71. <https://doi.org/10.1198/jasa.2009.tm08106>.

(Remarks on code availability)

Version 1:

Reviewer comments:

Reviewer #1

(Remarks to the Author)

I thank the authors for their revision and replies to my questions. I find the revision much easier to follow. In model comparison, typically we can compute the empirical distribution of point-wise loss functions. Take the mean, we get the ELPD. Take the empirical distribution, we can further calculate the empirical probability of one model being better than another model. This paper proposes a future step: to compare the discrepancy between the quantile function of the point-wise losses and the quantile under the assumed model.

All my previously raised questions are answered in the rebuttal, and I am satisfied with the answers. I only want to point out a recent reference "The Posterior Predictive Null" by Moran and Blei, which shares similar ideas of synthetic replications and is hence worth some discussion. Looking forward, it makes sense to compare more traditional tools such as the ELPD, Bayes factor, etc., with the R-distribution approach in terms of their convergence rate and robustness---but I understand such a comparison is likely beyond the scope of this current paper.

(Remarks on code availability)

Reviewer #3

(Remarks to the Author)

First, I would like to thank the authors for their detailed response on my comments. In particular, the focus on model misspecification is now very clear, right from the introduction, the new table 2 is more informative than the previous one and the comparison with other methods has been improved.

There are however few concerns that remain.

1. Well posed problem? I appreciate that the authors make it clear that that model misspecification is at the heart of their study. As I understand, this misspecification process is expressed through the epistemic distribution Ω (or the collection of such distributions). Now, I understand that the authors assume this epistemic distribution to be unknown (because in realistic settings, we precisely do not know the ground truth generative process), but this is precisely where the problem becomes ill-posed.

How can we tell if the proposed model comparison method is successful if the misspecification process is unknown? What is the success metric? As I understand, for any known distribution Ω (or collection of such distributions), the authors estimate the free parameter c that offers the best correlation between B^{EMD} and B^{epis} (see also point 2 below). So “success” is claimed when this correlation is high. However, because at the end of the day Ω will be unknown, how can we decide if the chosen c was good or not.

2. Calibration of the ad-hoc parameter c . Even though I understand that every method has its parameters (such as for example the parameters describing the prior for the Bayes factor), the parameter c is problematic for several reasons:

- As mentioned in my previous review, all calibration curves (of Fig 6) violate Eq 34 - not just around the specific (0,0) point, but on a large fraction of the curve.
- This point above suggests that the flexibility offered by the free parameter c is not sufficient to reflect the epistemic uncertainty. This is actually not a surprise because Ω is high-dimensional and c is only 1-dimensional.
- Because the parameter c is Ω -specific and because Ω is unknown, I do not see how we can be confident that we chose the right parameter c (see point 1 above).
- Finally, it is not straightforward to give an interpretation to the parameter c (let alone give an analytical expression of this parameter c as a function of the problem set up).

3. Relevant Risk distribution? My understanding is that the whole argument of the paper is that it is better to consider the $R(\text{isk})$ -distribution (instead of the empirical risk - which is a scalar) if we want to take into account the uncertainty due to the model misspecification. This R -distribution is obtained through some stochastic process on quantile functions. This brings two questions:

a. Dependence on the ad-hoc parameter c . Since the center of the R -distribution is precisely the (expected) risk (Eq 26), the ordering between model A and B will remain unchanged if we take into account the R -distribution perspective instead of the (expected) risk perspective. Indeed, if $E[R_A] < E[R_B]$ then $p(R_A < R_B) > 0.5$. So the only thing that changes is the level of confidence that one model wins over the one. This level confidence is directly influenced by the variance of the R -distribution which is scaled by the ad-hoc parameter c whose calibration is questionable (see comment 2). So it remains unclear to me to what extent this R -distribution perspective is really relevant.

b. Why not using the Q -distribution? Since $(x_i, y_i) \sim M_{\text{true}}$ are random variables, $Q(x_i, y_i, M_A)$ is also a random variable with a given distribution (let's call it the Q -distribution). The authors may want to argue why the R -distribution (which is rather complicated to compute and requires the ad-hoc parameter c) is better than the Q -distribution which is straightforward to use. One way to show this could be to perform the same analysis as in Fig. 6 but with B^Q instead of B^{EMD} . A priori it is unclear to me why B^{EMD} should be better correlated with B^{epis} than B^Q (especially if an additional free parameter c is given to the Q -distribution (in order to make the comparison fair), i.e. $Q' = Q + \epsilon$, where $\epsilon \sim N(0, c)$).

4. Consistency by construction?

At the beginning of the results, the authors write “A common consideration with model selection criteria is whether they are consistent, i.e. whether they eventually select the true model when the number of samples grows. When models are misspecified, this is usually expressed as whether a criterion selects the model with the lowest risk [27;28]”

It seems to me that redefining consistency as whether a criterion selects the model with the lowest risk is neither appropriate nor necessary. Indeed, this redefinition evades the real question (whether the true model is selected when the number of sample grows) and replaces it by a tautology. I quickly looked at those refs [27 and 28], but I didn't see this redefinition of consistency. Also, it should be noted that other authors (e.g. De Blasi, 2013) did consider the proper consistency definition in the context of misspecified models.

REFS

- De Blasi, Pierpaolo, and Stephen G. Walker. “Bayesian Asymptotics with Misspecified Models.” *Statistica Sinica*, 2013. <https://doi.org/10.5705/ss.2010.239>.

(Remarks on code availability)

Version 2:

Reviewer comments:

Reviewer #3

(Remarks to the Author)

I would like to thank the authors for having addressed my major comments. In particular, the authors

- added Fig 9 on the calibration method - which I find helpful,
- performed the proposed analysis with the B^Q and showed that it is less suitable than the B^{EMD} criterion - which I find convincing,
- better highlight the limitations of the proposed approach (in particular the paragraph from L 1536-1548)

Please find below few last minor comments:

1. The empirical risk equation is equation (5). However in several places in the manuscript, it is referred as equation (2.1). (e.g. lines 263, caption of Fig3, line 536, line 583,....).
2. Fig 6. The label of the y-axis (i.e. B^{ϵ}) should be displayed

(Remarks on code availability)

Reviewer #1 (Remarks to the Author):

I sympathize with the author for their ambitious attempt to nearly invent a new framework for model comparison, and it is nearly orthogonal to existing statistical tools. The proposed method is likely useful when comparing many physical models, especially when none is correct. However, after reading the paper, the question I find myself asking is whether the effort of constructing such a new framework and the overhead of introducing new concepts are fully justified by their additional practical or pedagogical merits over classical methods.

I find the writing difficult to follow. I am often overwhelmed by the introduction of numerous new notions and technical terminology. The key method is not introduced until roughly page 10. Even after reading the paper, some key information seems missing: for example, is each candidate model fitted by MLE or by Bayesian inference? The description of the replication process on pages 10-11 is especially confusing. I could make guesses to fill these gaps, but I would prefer that the author clarify the writing and make it more concise.

I think there are at least two unstated assumptions of the method: (1) The data points come in IID pairs, and (2) The sample size is assumed to be "big enough" such that we can ignore any finite sample/statistical uncertainty. Both assumptions are questionable, and arguably the finite sample behavior of model selection or model comparison is crucial to this field. For example, the proposed method may fail to account for "overfitting." When the candidate models are nested, it appears that the more complex model will always have the smaller in-sample risk, and thereby be picked by the proposed method. Perhaps I am misunderstanding the motivation of this work, but if the sample size is assumed to be infinite, the sample mean of empirical risk would already be good enough for model comparison.

The concept of aleatoric/epistemic uncertainty seems overused in the paper and does not fully clarify the proposed method. I must note that what the author refers to as "Aleatoric uncertainty model companions" feels somewhat like a strawman argument to me. Take the example of Fig 4 panel A: the more accepted model comparison approach is to compare the pointwise risk among four models. If we take model A as a benchmark and visualize the empirical distribution of pointwise excess risk of models B, C, and D, this would resemble the difference between a paired t-test and a t-test. This approach would eliminate a large portion of the sample variation. Ref [2] discusses more on uncertainty in model comparison.

In my opinion, the idea of “empirical model discrepancy” is not completely new. It seems closely related to posterior predictive checks (Ref [1]), where the test function is limited to the risk quantiles. This paper does make an extension, in that the classical posterior predictive check is only used for model checks, while here it is used for model comparison.

Overall, I can see the potential of this tool. But I am not yet convinced that existing tools cannot address the issue raised in this paper. A minimal illustration should include a comparison (in testing power or model selection accuracy) between this new tool and existing tools (mean of empirical risk, bootstrap, cross-validation, Bayesian posterior predictive check). Criticizing all existing tools as “aleatoric” does not make the newly proposed tool immune from comparison.

[1] POSTERIOR PREDICTIVE ASSESSMENT OF MODEL FITNESS VIA REALIZED DISCREPANCIES

[2] Uncertainty in Bayesian Leave-One-Out Cross-Validation Based Model Comparison

Authors’ reply to Reviewer #1

We thank the reviewer for their helpful comments, and especially for pointing us to Ref [1] (Posterior predictive assessment...) – this work certainly attacks similar challenges, although the overlap with our work is perhaps somewhat less than the reviewer suggests. We clarify this below.

I find the writing difficult to follow. I am often overwhelmed by the introduction of numerous new notions and technical terminology. The key method is not introduced until roughly page 10.

The new version in our view is a substantial improvement; in particular:

- Our work is now much better located within the statistics literature, which should help making the terminology less overwhelming.
- Our working assumptions and goals are stated much more explicitly.

We have also removed material which upon further consideration was unnecessary.

All told, these changes even out in terms of the number pages, so that the description of the stochastic process still starts at page 10. We also do a better job now of announcing it earlier in the paper, but are reluctant to further reduce the exposition: it is essential to establishing our framing, which as the reviewer notes, is somewhat orthogonal to the standard statistical approaches. We want to give

readers every chance to understand the problem we set off to solve before diving into our mathematical solution.

Even after reading the paper, some key information seems missing: for example, is each candidate model fitted by MLE or by Bayesian inference?

It has been made clearer now that the method is agnostic to how the models were obtained. In particular this sentence on p.2 is now in bold:

“we therefore assume that we have already obtained a set of candidate models”

Given the importance of this point however, we have also added this remark at the top of section 2.2, when we explain the neural circuit model:

“It is important to note that we treat the models M_A to M_D as simply given. We make no assumption as to how they were obtained, or whether they correspond to the maximum of a likelihood. We chose this example, where parameters are obtained with a multi-stage, semi-automated grid search, partly to illustrate that our method is agnostic to the fitting procedure.”

The description of the replication process on pages 10-11 is especially confusing. I could make guesses to fill these gaps, but I would prefer that the author clarify the writing and make it more concise.

In this new version, we have clarified the related text, especially the first few paragraphs of this section. We have added additional references to Fig 5 to help guide the reader, and also added visualisations of the beta distributions in Fig 5b. It is difficult however to make this more concise: Section 2.6 is the theoretical core of our paper, which already fits on only two pages. That said, we are more than willing to incorporate more specific suggestions which the reviewer feels would make the text more understandable to readers.

I think there are at least two unstated assumptions of the method: (1) The data points come in IID pairs, and (2) The sample size is assumed to be “big enough” such that we can ignore any finite sample/statistical uncertainty. Both assumptions are questionable, and arguably the finite sample behavior of model selection or model comparison is crucial to this field.

Regarding the mentioned assumptions:

1. It is true that our derivations assume samples to be IID, as does most work on model selection. This is mostly for convenience: most of the time, one can subsample the dataset to get a smaller one where the samples are closer to IID (we can then talk of an “effective” number of samples). This is especially true for natural phenomena, where the goal is to describe a type of universal behaviour. It is worth noting that, at least for time series data,

this is equivalent to obtaining IID samples from an MCMC chain – for which practice has now established that there is generally no statistical benefit to thinning the chain to the number of effective samples [1].

- o In fact, in our main neural model example, the data are time series and thus *not* IID. This poses no problem to the method.
 - o Treating time series as highly-correlated IID is also not uncommon in the statistics literature; see e.g. the work of Golden [2] on model selection tests for time series.
 - o The real assumption is *stationarity* within a realisation of the data distribution.
 - o This assumption is now made explicitly clear in §2.4.
 - o We thank the reviewer for this remark; it has helped sharpen our thinking, and thereby also our exposition.
2. There are many situations (often referred to these days with the expression “big data”) where large numbers of samples are in fact available.
- o In machine learning, splitting data into training and test subsets is standard procedure.
 - o Even small tabletop experiments often generate large amounts of data. For example, a single electrode might record at 1 kHz for multiple minutes.
 - o On the grounds that experimental measurements always have some level of uncertainty, the large sample limit is especially instructive to help identify where that uncertainty comes from.
 - o Moreover, the empirical CDFs will actually reflect uncertainty due to smaller sample sizes, in the form of jaggedness. This in turn will affect the discrepancy δ_{emd} , and thus increase the variance of our HB process over quantile functions. Therefore, although we don’t explicitly model the effect of finite samples, they do contribute to the spread of the R-distribution (and therefore the uncertainty). We have added a comment to this effect at the end of our discussion.
 - o As can be seen from the new Fig. 8, our method does in fact implicitly account for finite sample size; in the discussion we also now mention that doing so explicitly could be a direction for future research.

For example, the proposed method may fail to account for “overfitting.” When the candidate models are nested, it appears that the more complex model will always have the smaller in-sample risk, and thereby be picked by the proposed method.

This is a modelling choice, to select models based on their predictive performance on data which was not used for fitting. It is by far the most common choice in the

natural science and machine learning, where data tend to be less limited. Some applied statisticians also strongly favour comparing models based on their expected log predictive density (elpd), which is essentially the risk scaled by dataset size; this is the argument made e.g. by Vehtari and Ojanen (2012).

Of course, exclusively focussing on predictive performance means that in a nested model scenario, the more complex model is not penalized. This is true also of the elpd. This in our opinion is the lesser evil: nothing prevents a modeller from also considering model complexity in a second step, after having compared models using our method.

Note that in the case of nested models, if the true model is the simplest one, then the result of our criterion will be that the models are indistinguishable from the point of view of performance since their R-distributions overlap. Conversely, the more complex model is also not favoured unless it actually improves performance. This is more obvious now in our new Fig. 8. Moreover, overfitting which negatively impacts generalisation performance (this is the definition typically used in machine learning, where over-parametrised models are commonplace) will be identified, since models are evaluated on held-out test data.

The new section at the start of the results now clearly sets out this choice to select models based on risk, and explains the pros and cons. We also remind readers of this choice in the second paragraph of the discussion.

Perhaps I am misunderstanding the motivation of this work, but if the sample size is assumed to be infinite, the sample mean of empirical risk would already be good enough for model comparison.

We have rewritten parts of the introduction to clarify our motivation, and to make clear early on the importance that our criterion account for variability in experimental replications. This is important because in the natural sciences, the goal is usually induction beyond the specific conditions which generated the data. This consideration of a non-stationary data generation process is a key difference with other work studying uncertainty in the presence of misspecification, including of that Sivula et al (2023) which the reviewer suggested.

The concept of aleatoric/epistemic uncertainty seems overused in the paper and does not fully clarify the proposed method. I must note that what the author refers to as "Aleatoric uncertainty model companions" feels somewhat like a strawman argument to me. Take the example of Fig 4 panel A: the more accepted model comparison approach is to compare the pointwise risk among four models. If we take model A as a benchmark and visualise the empirical distribution of point-wise excess risk of models B, C, and D, this would resemble the difference between a

paired t-test and a t-test. This approach would eliminate a large portion of the sample variation. Ref [2] discusses more on uncertainty in model comparison.

This has been substantially improved, especially in the introduction and first and third sections of the results. Epistemic uncertainty has been further distinguished between uncertainty due to finite samples, and uncertainty due to replications.

The old Fig 4 has been moved to the supplementary, and is no longer used to argue in favour of our method. (With the revised text it became superfluous.) Now it is used only to discuss the effect of finite samples.

A t-test would indeed more closely approximate what we are doing in Eq. (11), but would not actually solve the problem we are considering: in the large data limit, it would always select the model with the lowest pointwise risk with perfect confidence. This is because the distribution of average pointwise risk (shown at the top of Fig. 3) degenerates to a Dirac distribution – falsely indicating no uncertainty on the risk, when in fact some uncertainty remains due to non-stationarity. This is precisely the type of “overconfidence” we seek to avoid.

(In the suggested reference 2, the standard error estimates scale as σ/\sqrt{n} (see e.g. Eq. (10) therein), so that if we divide by \sqrt{n} to get average point-wise std error, they scale as σ and therefore converge to a Dirac. In other words, if the models are misspecified, the suggested methods in Ref. 2 will also underestimate uncertainty.)

More generally, we hope our reply to the previous point, and more importantly the changes we made to the text, clarify that our method in particular accounts for epistemic uncertainty arising from non-stationarity (i.e. experimental variations).

In my opinion, the idea of “empirical model discrepancy” is not completely new. It seems closely related to posterior predictive checks (Ref [1]), where the test function is limited to the risk quantiles. This paper does make an extension, in that the classical posterior predictive check is only used for model checks, while here it is used for model comparison.

We thank again the reviewer for pointing us to this highly relevant paper by Gelman, Meng and Stern. The approach they use is indeed similar in two main points:

- Like us, they measure discrepancy between replications of the experiment.
- Like us, they emphasise that the statistic used to assess performance is not tied to the model itself, but rather can be tailored to the prediction task.

As the reviewer notes, Gelman et al. do not use their approach for model selection. This however highlights a more fundamental difference between our methodologies: their goal is to answer the question “is this model plausible given the data?”. They do this by using the Bayesian posterior to construct a reference distribution for the discrepancy, then locate the discrepancy of the real data within that distribution.

In contrast, in our case we consider the question “is this model plausible” to possibly be ill-posed. “All models are wrong” as the adage goes, and so with enough data, a strict test will eventually reject every model. Instead, we ask the question “Is model A better than model B”: this question remains well-posed, independently of whether one, both, or none of the candidate models are actually plausible given the data. Therefore, while our approach is neither Frequentist nor Bayesian, it is fundamentally *comparative*. The other key difference is in how we measure, and then use, the discrepancy between models:

- First, a key insight for us was to measure discrepancies not in the data space (as Gelman et al. do), but in the *loss* space. More precisely, we measure the difference between two cumulative distributions of the loss (mixed and synthetic PPFs).
- Second, rather than use the discrepancy directly as a measure of the fit quality, we use it to determine the variance of our stochastic HB process. This in turn determines the spread of R-distributions.

We now include this reference and the points above in our discussion.

Overall, I can see the potential of this tool. But I am not yet convinced that existing tools cannot address the issue raised in this paper. A minimal illustration should include a comparison (in testing power or model selection accuracy) between this new tool and existing tools (mean of empirical risk, bootstrap, cross-validation, Bayesian posterior predictive check). Criticizing all existing tools as “aleatoric” does not make the newly proposed tool immune from comparison.

We are glad that on balance, and despite some lack of clarity and possible misunderstandings, our work is perceived positively.

Based on feedback from all three reviewers, we rewrote the section comparing our method to other established criteria.

- The new Fig. 8 (which replaces the old Table 2) uses a collection of nested and non-nested models, with a dataset designed to be at the intersection of what each method is designed to do. This way, each is presented and compared fairly.

- Instead of attempting to show which method behaves better, we focus on how each works within a different paradigm, and therefore answers a slightly different question, with different assumptions. We think this is fairer than forcing them all to answer the same question (as a straight comparison would require), and more useful to readers who can then decide whether the method we propose is better adapted to their needs.

References

1. Luengo et al. (2020). A Survey of Monte Carlo Methods for Parameter Estimation. *EURASIP Journal on Advances in Signal Processing* 2020, 25(1). <https://doi.org/10.1186/s13634-020-00675-6>.
2. Golden, R. M. (2003). Discrepancy Risk Model Selection Test Theory for Comparing Possibly Misspecified or Nonnested Models. *Psychometrika*, 68(2), 229–249. <https://doi.org/10.1007/BF02294799>
3. Hüllermeier, E., & Waegeman, W. (2021). Aleatoric and epistemic uncertainty in machine learning: An introduction to concepts and methods. *Machine Learning*, 110(3), 457–506. <https://doi.org/10.1007/s10994-021-05946-3>
4. Vehtari, A., & Ojanen, J. (2012). A survey of Bayesian predictive methods for model assessment, selection and comparison. *Statistics Surveys*, 6(none). <https://doi.org/10.1214/12-SS102>

Reviewer #2 (Remarks to the Author):

This is a review of the manuscript entitled: "Epistemically robust selection of the fitted models".

As a note to authors, this review is written in good faith with the goal to improve the scientific work presented in the manuscript, and trying to indicate its merits and shortcomings. As a reference, my formal training is in statistics and I work in theoretical statistics, philosophy of science, and metascience. I mainly develop theories on reproducibility, replicability, and statistical methodologies to test how sciences progress.

The manuscript presents a novel method of model selection with the claim that it is epistemically robust. The key methodological improvement to me is the work that was done on separating and quantifying the model uncertainty and sampling uncertainty in statistical terms. The method has its commended merits due to this development. In separating and quantifying these two sources of uncertainty the method uses a clever modeling approach and fundamental quantities such as quantiles, which are all positives. Using risk as a desirable criterion will alineate somewhat information-theory grounded people, but it is a valid approach and one of the best available approaches that we know, so I am fine with this.

As a personal note, I really liked the methodological development, especially the modeling of stochastic process on quantile functions. This is all interesting solid work and with simple enough concepts to boot.

On the other hand, I have many issues with the motivation and presentation of the work. I am afraid my review might sound mostly critical in a negative way, but I want to clarify the reasons. I have nothing against the method developed in the manuscript. In fact, I enjoyed reading it, it piqued my curiosity, and it motivated my to thinking about challenging aspects of model selection. My criticisms mainly revolve around to the following.

1. The way the idea is motivated with a misleading/misinformed perspective taken with respect to our contemporary understanding of science/statistics,
2. Unfair comparisons with other existing statistical methods,
3. Overreaching and overgeneralization of the merits of the method.

These three factors weigh down what otherwise would be a good methodological contribution. The presentation dives into considerably loaded issues

unnecessarily. My suggestion is to present the method as what it is, a model selection procedure with potentially some good properties, compare/contrast it with state of the art model selection procedures and leave it at that. In particular, I suggest to stay away from the idea of “falsification” and overemphasizing “convergence” (what is known as consistency in statistics).

Regarding point 1 above. The manuscript motivates the method presented starting with a reference to Tarantola’s 95 work. This is mostly a philosophy of science work. There is a long way from the issues raised by Tarantola to the solution of these issues (if indeed, they exist). These issues mostly clash with our current statistical knowledge and model selection procedures (whether consistent or not) are not capable of solving these. This is why, the idea of “falsification” is almost entirely abandoned in contemporary philosophy of science and statistics. Falsification is fraught with difficulties that current state of the arts cannot circumvent. Toward the end of the manuscript even Popper and Eddington observations (seems to be a recurrent pet-topic with physics in general) were dragged into the discussion, without a proper treatment of what they represent. Popper never accepted induction. Popper’s falsification idea applies to hypotheses that can be decided based on a finite number of observations (which is not what model selection is about). Eddington observations are a type of Newton’s experimentum crucis, where a single observation/study is sufficient to distinguish between two competing hypotheses, verifying the predictions of an extremely well-developed physics theory. Analyzing data statistically with the goal of model selection from a candidate set of models (that in the first place we have chosen) is a very different exercise. Statistical consistency –selecting the true model in the limit as sampling uncertainty goes to zero– is a desirable property from statistical perspective, but equating this with falsification of theories is a naive interpretation of Popper’s work at best, dismissing all the current development in philosophy of science in recent decades.

Regarding point 2 above. The proposed method is compared against some existing model selection criteria. To me it sounds that consistency of a model selection method is the most important criterion for the authors from repeated references to it in the text. I could not shake-off the feeling that all the negative aspects of the existing –classical– criteria were exposed to make the proposed method look better (again, I think that the proposed method has merits but this is a poor way of going about showing them). In fact, over the decades there have been a number of improvements of classical model selection criteria that surpass the classical usage of these methods. With respect to the current manuscript criticizing these, below are some examples.

- i) One model selection criterion that authors make comparison with is AIC. They write that AIC is inconsistent (as n goes to infinity, it does not select the true model with probability 1), which is true. AIC was developed in 1978-1979. It was immediately apparent that this was not a consistent estimator of the true model. Schwarz has developed BIC around the same time, which is consistent (with an additional term that depends on the sample size). So, why not compare the performance of the proposed method with respect to BIC and beat AIC to death? Another state of the art model selection criteria is Minimum Description Length. This was not mentioned in the manuscript at all. It is worth noting that all these criteria for model selection are centered on the value of information brought in by the data and hence they are “information-theoretic” criteria. Even if they fail to satisfy the consistency criterion (which some of them do not) their purpose was not only to satisfy the consistency, but assessing how more information affects the model selection procedure. So, the comparisons are unfair. Along the same veins, we can ask how the proposed method of model selection in the manuscript evaluates information, for example measuring distances between distributions implied by models. I have nothing against using risk as a measure –this is the approach taken in the manuscript– but taking the risk approach will be challenging to reconcile with the information-theory.
- ii) Another model selection criterion that authors make comparison with is Bayes Factors. Bayes factors have a number of shortcomings and these are stated in the manuscript, but it was omitted that they converge to BIC and hence approximately consistent. Criticizing Bayes factors being affected by priors is also unfair. The whole Bayesian approach relies on prior information.
- iii) Another method that was criticized is bootstrap. Bootstrap originally developed by Efron in 1978-1980, was not meant to be a model selection criterion. It is a method to estimate the sampling distributions of statistics. However, later –for example in the context of Bayesian bootstrap and other improvements– it has been used as a model selection criterion. It was mentioned that bootstrap estimators are biased, which is true, but then, this again puts the consistency at the center as the sole criterion in model selection, neglecting the desirable property of quantifying and expressing uncertainty for finite sample sizes.

Regarding point 3 above. An important point is that statistical methods embrace uncertainty and aim to quantify it. Not to eliminate it, as seems to be the

perspective taken in the tone of the manuscript. Practically, contemporary statistical view is that although consistency is a desirable property, we will never get infinite sample size in a realistic situation and that this is only a theoretical guide. Not the goal in developing statistical methods. I realized that authors have cited one of Andrew Gelman's papers. More of his work details about a holistic approach to statistical methods and analyses and not singlemindedly focusing on properties of a single method making it the focus of evaluating scientific theories. This is also echoed generally among philosophers and statisticians and is in stark contrast with the positioning of the method in the manuscript.

Other comments:

1. Abstract: "falsify" and "reject" are not used in the same sense in statistics. Rejection of a hypothesis does not mean falsifying a theory associated with it. Probably this was written in physics sense –and I can grudgingly accept that– but at least I would appreciate if you can clarify it, perhaps with a footnote (since this is a multidisciplinary journal).
2. Line 9: "...non-identifiable:..." Hard-identifiability and soft identifiability have different meanings theoretically. Please clarify which one you are referring to.
3. Line 125-126: In a realistic model selection problem there will be more than two competing models. If you are comparing them pairwise, multiple use of the same data (using information twice) will be a problem that needs to be corrected statistically. This is a challenging problem, currently with no satisfactory solution and I do not require one, but the problem is worth acknowledging around lines 1215-126.
4. Lines 178-180, and 214-220: Do not sit well with me due to problems mentioned above.
5. Lines 257-258: Bayes risk and frequentist risk have very different interpretations. It is worth clarifying what equation (5) refers to. Looks like you are integrating over parameters to get to the unconditional expectation of the model which would imply Bayes risk, but this is unclear from the formula.
6. Line 388: Typo in "explicitely"
7. Line 435-436: epsilon is not "confidence". It seems to refer to type I error rate. Confidence is $100 \cdot (1 - \epsilon)$. I would not bring this into the discussion

since this is a purely frequentist hypothesis testing criterion and has no business in model selection.

8. Line 478: Please describe “stationary distribution”. It has a number of meanings. I assume you mean as time goes to infinity there is a distribution of interest that the process converges, but could not be sure.
9. Line 492: There are different definitions of Heaviside function, in particular for $H(0)$. Please clarify which one you are using.
10. Line 538: “light tails” should be “exponential tails” I think, since you are using Gaussian. Light tailed distributions are a separate class lighter than exponential tails.
11. Line 659: Equation (19a) has a comma on the right hand side. This does not makes sense to me, unless it is the bivariate nature of the expectation you are trying to capture with a single equation? Please add a brief explanation.
12. Line 896: “Is it 16 variations? or 18 variations? I think somewhere else in the paper it says 16?”
13. Line 939: Equation (34) is poorly written. The probabilities $P(A)$ and $P(B)$ are conditional on data, which makes all the difference to a Bayesian. Please use correct notation.

Authors’ reply to Reviewer #2

As a note to authors, this review is written in good faith with the goal to improve the scientific work presented in the manuscript, and trying to indicate its merits and shortcomings. As a reference, my formal training is in statistics and I work in theoretical statistics, philosophy of science, and metascience. I mainly develop theories on reproducibility, replicability, and statistical methodologies to test how sciences progress.

The manuscript presents a novel method of model selection with the claim that it is epistemically robust. The key methodological improvement to me is the work that was done on separating and quantifying the model uncertainty and sampling uncertainty in statistical terms. The method has its commended merits due to this development. In separating and quantifying these two sources of uncertainty the method uses a clever modeling approach and fundamental quantities such as quantiles, which are all positives. Using risk as a desirable criterion will alineate somewhat information-theory grounded people, but it is a valid approach and one of the best available approaches that we know, so I am fine with this.

As a personal note, I really liked the methodological development, especially the modeling of stochastic process on quantile functions. This is all interesting solid work and with simple enough concepts to boot.

On the other hand, I have many issues with the motivation and presentation of the work. I am afraid my review might sound mostly critical in a negative way, but I want to clarify the reasons. I have nothing against the method developed in the manuscript. In fact, I enjoyed reading it, it piqued my curiosity, and it motivated me to thinking about challenging aspects of model selection. My criticisms mainly revolve around to the following.

1. The way the idea is motivated with a misleading/misinformed perspective taken with respect to our contemporary understanding of science/statistics,
2. Unfair comparisons with other existing statistical methods,
3. Overreaching and overgeneralization of the merits of the method.

These three factors weigh down what otherwise would be a good methodological contribution. The presentation dives into considerably loaded issues unnecessarily. My suggestion is to present the method as what it is, a model selection procedure with potentially some good properties, compare/contrast it with state of the art model selection procedures and leave it at that. In particular, I suggest to stay away from the idea of “falsification” and overemphasizing “convergence” (what is known as consistency in statistics).

We thank the reviewer for their helpful comments. Their broader expertise is much appreciated, and we believe that our revised version closely follows their suggestions. We find that this has much improved the manuscript and hope that they agree.

We note in passing that, partly to address concerns of reviewer 3, we did end up adding a few comments in the manuscript concerning the method’s consistency. In short, we can make it consistent by construction, by a) choosing a proper scoring rule for the risk function, such as the log likelihood, and b) comparing the means of the R-distributions, in addition to their overlap. See “EMD rejection rule”.

Regarding point 1 above. The manuscript motivates the method presented starting with a reference to Tarantola’s 95 work. This is mostly a philosophy of science work. There is a long way from the issues raised by Tarantola to the solution of these issues (if indeed, they exist). These issues mostly clash with our current statistical knowledge and model selection procedures (whether consistent or not) are not capable of solving these. This is why, the idea of “falsification” is almost entirely abandoned in contemporary philosophy of science and statistics. Falsification is fraught with difficulties that current state of the arts cannot circumvent. Toward the end of the manuscript even Popper and Eddington observations (seems to be a recurrent pet-topic with physics in general) were dragged into the discussion, without a proper treatment of what they represent. Popper never accepted induction. Popper’s falsification idea applies to hypotheses that can be decided based on a finite number of observations (which is not what model selection is about). Eddington observations are a type of Newton’s experimentum crucis, where a single observation/study is sufficient to distinguish between two competing hypotheses, verifying the predictions of an extremely well-developed physics

theory. Analyzing data statistically with the goal of model selection from a candidate set of models (that in the first place we have chosen) is a very different exercise. Statistical consistency –selecting the true model in the limit as sampling uncertainty goes to zero– is a desirable property from statistical perspective, but equating this with falsification of theories is a naive interpretation of Popper's work at best, dismissing all the current development in philosophy of science in recent decades.

[Since this comment concerns philosophy of science, which is far from our expertise, we ask the reviewer's indulgence if our wording below is imprecise.]

We should start by clarifying that what we cite is Tarantola's 2006 commentary [1], rather than his 1995 work with Mosegaard [2] (which we presume is the one the reviewer is referring to). While it seems clear that the thoughts in [1] were influenced by the work in [2], the commentary is much less prescriptive, and does away with the (in our view difficult to justify) idea of assigning probabilities to models.

We found citing Tarantola's commentary helpful to streamline our introduction, since in it he lays down, in three short pages, three key arguments which frame the model selection problem in the way that we need: - Often our inverse problems (ie. inferring model parameters from observations) are ill-posed, and for science to progress we need to deal with that. - The outcome of a model selection need not be a single model: it could be a collection of models. - The set of non-rejected models is the solution to the inverse problem. The models are not assigned probabilities.

Especially the last point is an important distinction with other methods like GLUE, Bayes factors, MDL, and indeed Tarantola's own 1995 work. In order to arrive at probabilities, these methods all make assumptions which don't always hold. In our view, it is instead better to let the modeller interpret the set of non-rejected models, using their domain knowledge (and/or that of their collaborators). If weighing the models makes sense for their analysis, they can do so at this stage.

We would prefer to keep the reference to [1] for the reasons given above, however if the reviewer really objects to it we can remove it.

We acknowledge however that our original introduction failed to clearly communicate how our approach differs from [1] and other model comparison methods. The new version does a much better job in this regard:

- When we lay out the program Tarantola describes in [1], we use "rejection criterion" instead of Tarantola's "falsification criterion". (We agree that

Popperian falsification, as advocated in [1], is probably not practical.) In this vein, we have also replaced the word “falsification” with “rejection” throughout the manuscript.

- We considered clarifying that our philosophical position (or lack thereof) differs from Tarantola’s in this regard, but judged that for the vast majority of our readers, such a precision is likely to be more confusing than helpful.
- We clearly state that the ability of a model to generalise – the property we are actually after – is quite different from consistency.

For the record, our work was not motivated by Tarantola’s, or any question in philosophy of science. Rather we were pushed to develop this method in response to a real need within our research on data-driven modelling (which one could broadly situate in the field of AI for science): we needed a way to reliably select from fitted models, and none of the classical model selection methods were working. Through a series of numerical experiments, we identified that the problem was that they do not allow the data-generating process to change between replications. Thus, we designed tests which are especially sensitive to epistemic errors, and set out to design a method which would pass them. The result is the B^AEMD criterion we propose in this paper.

Incidentally, our concern with real use cases means that we are interested in a method which also works in the presence of sampling (aleatoric) uncertainty, even though the latter is not the focus of this work. Indeed, our examples do include both sampling and modelling uncertainty.

Regarding point 2 above. The proposed method is compared against some existing model selection criteria. To me it sounds that consistency of a model selection method is the most important criterion for the authors from repeated references to it in the text. I could not shake-off the feeling that all the negative aspects of the existing –classical– criteria were exposed to make the proposed method look better (again, I think that the proposed method has merits but this is a poor way of going about showing them).

This perceived equivalence to consistency is an important source of confusion, which we had not anticipated. To remedy this, we have made major changes to the Introduction, sections 2.1, 2.3 and 2.8 of the Results, and Discussion.

The short form: What we are after is what experimental scientists call “reproducibility” and machine learners call “generalisability”. In other words, we want the selected model(s) to continue to perform well on replication datasets, *even when the data-generating distribution (i.e. the experimental setup) changes*,

as it inevitably does. This is the crucial difference with consistency, which only considers replications from the same data-generating distributions.

As a side note: when models are misspecified, the most common approach to defining consistency seems to be to select the model with lowest risk as the “true” model [6,7]. As we now point out on at the start of our Results, a method such as ours which uses risk as the criterion is consistent by definition (as long as the risk functional is proper). The question of consistency is thus not particularly interesting. We do however use the word consistency in the two following contexts: in the context of statistical physics (“consistency equation”) and the context of stochastic processes (“self-consistency”).

Regarding now the reviewer’s point concerning the negative aspects we chose to highlight. These choices are entirely informed by our experience fitting models to real neuroscience data. We found that studying the behaviour of criteria in terms of dataset size was effective at accentuating the behaviour we wished to avoid (in particular underestimation of the uncertainty). We then used this as a guide to design a criterion which would not have this behaviour.

Thus it is not so much that we chose those aspects to make our method look good, but we did design it to perform well on those aspects. We agree however that we overemphasized them. The new version makes it clearer that while the behaviour of our criterion in cases of misspecification and large dataset sizes is useful for distinguishing it from other criteria – especially if it will applied to experimental data – these are far from the only aspects to consider.

In fact, over the decades there have been a number of improvements of classical model selection criteria that surpass the classical usage of these methods. With respect to the current manuscript criticizing these, below are some examples. i) One model selection criterion that authors make comparison with is AIC. They write that AIC is inconsistent (as n goes to infinity, it does not select the true model with probability 1), which is true. AIC was developed in 1978-1979. It was immediately apparent that this was not a consistent estimator of the true model. Schwarz has developed BIC around the same time, which is consistent (with an additional term that depends on the sample size). So, why not compare the performance of the proposed method with respect to BIC and beat AIC to death? Another state of the art model selection criteria is Minimum Description Length. This was not mentioned in the manuscript at all. It is worth noting that all these criteria for model selection are centered on the value of information brought in by the data and hence they are “information-theoretic” criteria. Even if they fail to satisfy the consistency criterion (which some of them do not) their purpose was not only to satisfy the consistency, but assessing how more information affects the model selection procedure. So, the comparisons are unfair. Along the same veins, we can ask how the proposed method of model selection in the manuscript evaluates information, for example measuring distances between distributions implied by models. I have nothing

against using risk as a measure –this is the approach taken in the manuscript– but taking the risk approach will be challenging to reconcile with the information-theory.

As noted above, we are not particularly interested in consistency. Nevertheless, the reviewer’s suggestions are still appropriate and we have taken this opportunity to extend our comparison to other methods. This section now:

- Uses models and a data-regime at the intersection of what all methods are designed to do, and thus allows for a fairer comparison.
- Uses figures rather than a table to better convey how criteria depend on the dataset size.
- Adds criteria to the comparison following the reviewer’s suggestion:
 - BIC
 - MDL
- Highlights that each method operates within a slightly different paradigm.
 - Which method is best will depend on the data and the modelling goals.
 - The section now focuses on locating our method within the constellation of existing ones, to help readers determine when it is appropriate for their needs.
- We focus on the behaviour of criteria as a function of dataset size .
 - Studying the effect of misspecification on other criteria is relegated to the supplementary, with a revised version of the original table.
 - This ensures a fairer comparison, since not all criteria are designed to account for misspecification.

The new section we think clearly illustrates how our proposal is different from existing criteria, without putting any of them down.

In this comparison, all classic criteria turn out to have comparable behaviour with respect to (with the exception of MDL). This is not too surprising considering that we chose a regime where the Laplace approximations underlying AIC and BIC are reasonable. We hope however that the reviewer will forgive us for not “beating AIC to death”. This would be difficult to motivate, since from the view of our stated metric (minimizing risk), it performs similarly to BIC. In fact, on the basis of predictive performance, Findley [6] showed that one should expect it to perform better than BIC when models are misspecified.

We also decided not to open a discussion about information-theoretic aspects of model comparison, with the exception of a comment in the Discussion. This is an interesting question, but we think readers will be best served by us maintaining a

single consistent analytical viewpoint throughout (that of comparing models based on risk). It seems clear however that if one defines the risk as the expected log likelihood, then there should be possibilities to use R-distributions to compute information-theoretic quantities which account for model variations. In particular, the R-distribution itself corresponds to the uncertainty on entropy.

An essential feature of our method is that it can compare specific fitted models (as evidenced by the use of “fitted” in our title). It so happens that our criterion is also well-defined in the usual setting where one compares two structurally different models – this is how we were able to perform the comparisons – but in general neither MDL nor Bayes factors are even defined for the kind of problem we want to solve. We now remind the readers of this in the section where we compare different criteria.

[When enumerating their comments, the reviewer skipped ii). We therefore continue below with point iii), matching their enumeration.]

iii) Another model selection criterion that authors make comparison with is Bayes Factors. Bayes factors have a number of shortcomings and these are stated in the manuscript, but it was omitted that they converge to BIC and hence approximately consistent. Criticizing Bayes factors being affected by priors is also unfair. The whole Bayesian approach relies on prior information.

Indeed, this was insufficiently clear. What we wished to point out is the tendency of Bayes factors to exhibit a strong dependence on priors, to the point that the data may weakly determine the comparison outcome. This does not always occur, but the class where it can (distinct models with continuous parameters) is quite broad; this is the point made by Gelman and Yao [4, §6], which the text simply paraphrases (and of course cites).

This point should also be less of an issue in the new version, since the text focuses now specifically on what distinguishes our own method from other methods, rather than the shortcomings of other methods. The new version does a better job of putting the various methods in relation, including BIC and Bayes factors. See especially the new Fig. 8.

iv) Another method that was criticized is bootstrap. Bootstrap originally developed by Efron in 1978-1980, was not meant to be a model selection criterion. It is a method to estimate the sampling distributions of statistics. However, later –for example in the context of Bayesian bootstrap and other improvements– it has been used as a model selection criterion. It was mentioned that bootstrap estimators are biased, which is true, but then, this again puts the consistency at the center as the sole criterion in model selection, neglecting the desirable property of quantifying and expressing uncertainty for finite sample sizes.

The comparison to bootstrap was added in response to discussions with scientist colleagues, who felt that it helped clarify the distinction between what we do, and what they might have otherwise done to assess uncertainty. The goal was to anticipate alternative methods a typical scientist might consider – not necessarily what we think is the best or even second-best approach.

With the reworked Introduction, we felt that this was now mostly superfluous. The discussion of bootstrap was moved to the Supplementary, as part of a discussion on different forms of uncertainty which are mostly orthogonal to our work.

Regarding point 3 above. An important point is that statistical methods embrace uncertainty and aim to quantify it. Not to eliminate it, as seems to be the perspective taken in the tone of the manuscript. Practically, contemporary statistical view is that although consistency is a desirable property, we will never get infinite sample size in a realistic situation and that this is only a theoretical guide. Not the goal in developing statistical methods. I realized that authors have cited one of Andrew Gelman's papers. More of his work details about a holistic approach to statistical methods and analyses and not singlemindedly focusing on properties of a single method making it the focus of evaluating scientific theories. This is also echoed generally among philosophers and statisticians and is in stark contrast with the positioning of the method in the manuscript.

There seems to be an essential misunderstanding here, which we attribute to a failure of explanation on our part.

The point of contention seems to be the limit of infinite sample size, or at least the study of a criterion's behaviour as we approach that limit. As noted in some of our replies above however, the reason we do this is that this limit is particularly effective at identifying the type of uncertainty we are worried about, namely uncertainty due to misspecified models and non-stationary replications.

In a controlled lab setting with devices recording at 500 Hz or more, it is not uncommon to deal with datasets that are effectively infinite from the point of view of some statistical tests. In such cases, it should be uncontroversial that a) one can return to the lab to collect more data, and b) at least theoretically, at some point one has more data than they need. As scientists, we therefore want a criterion which is compatible with that experimental reality. In other words, a) the outcome of the criterion should not be too sensitive to the number of data points (i.e. when we decide to stop the experiment), and b) infinite data does not breed infinite certainty.

Therefore the original impetus for this work was to design a criterion for which the uncertainty *doesn't* vanish in the large data limit. Considering the large data

limit is also an effective way to obtain a criterion which is mostly independent of the dataset size.

We also of course want the method to work in real-world scenarios, which is why the examples we use to illustrate it also have sampling and modelling errors. Furthermore, our past Fig. 4 (now in Supplementary) and new Fig. 8 allow comparisons as a function of sample size. One thing Fig. 8 makes clear now is that our criterion does in fact implicitly account for the effect of finite sample size.

The new version includes the following major changes which address this point:

- We more clearly relate experimental practice to desirable features of a criterion, in particular that reproducibility requires a certain robustness to non-stationary replications.
- We stress that sampling uncertainty is still important (and present in our examples); it is just not our focus. (Since there are already tools for that.)
- We mention in the discussion that explicitly accounting for finite sample sizes could be a direction for future research.
- Some of these comments are expanded upon in a new section of the Supplementary, “Other forms of uncertainty”.

Other comments:

1. Abstract: “falsify” and “reject” are not used in the same sense in statistics. Rejection of a hypothesis does not mean falsifying a theory associated with it. Probably this was written in physics sense –and I can grudgingly accept that– but at least I would appreciate if you can clarify it, perhaps with a footnote (since this is a multidisciplinary journal).

We have expunged “falsify” from the manuscript. We see no point in using a potentially misunderstood term.

2. Line 9: “...non-identifiable...” Hard-identifiability and soft identifiability have different meanings theoretically. Please clarify which one you are referring to.

In this case we are thinking of hard identifiability, although the distinction is not really important at this phase of the argument. Since we expect the expression “non-identifiable model” to be familiar to many more readers than the distinction between “hard” and “soft” identifiability, we would prefer to keep our definition (which follows the colon) unchanged to avoid sidetracking readers already in the first paragraph of the introduction. The meaning becomes clear in the next paragraph when we talk about models differing only in their specific parameters.

3. Line 125-126: In a realistic model selection problem there will be more than two competing models. If you are comparing them pairwise, multiple use of the same data (using information twice) will be a problem that needs to be corrected

statistically. This is a challenging problem, currently with no satisfactory solution and I do not require one, but the problem is worth acknowledging around lines 1215-126.

We have added a comment to this effect in the discussion.

4. Lines 178-180, and 214-220: Do not sit well with me due to problems mentioned above.

These points have now been addressed in our responses to the main points above.

5. Lines 257-258: Bayes risk and frequentist risk have very different interpretations. It is worth clarifying what equation (5) refers to. Looks like you are integrating over parameters to get to the unconditional expectation of the model which would imply Bayes risk, but this is unclear from the formula.

In this example we use frequentist risk since we have point estimates for the parameters, but in the broader scheme this does not actually matter: our formalism starts from a risk functional, and modellers can choose how they quantify predictive performance. This is now stated explicitly at the top of our Results, below Eq. (4).

In our view equation (5) is already sufficiently clear, and any additional comment in its vicinity is more likely to distract than help readers.

6. Line 388: Typo in "explicitely"

Fixed

7. Line 435-436: epsilon is not "confidence". It seems to refer to type I error rate. Confidence is $100 \times (1 - \text{epsilon})$. I would not bring this into the discussion since this is a purely frequentist hypothesis testing criterion and has no business in model selection.

Changed to "threshold"

8. Line 478: Please describe "stationary distribution". It has a number of meanings. I assume you mean as time goes to infinity there is a distribution of interest that the process converges, but could not be sure.

We now clarify that we assume a strictly stationary process.

9. Line 492: There are different definitions of Heaviside function, in particular for $H(0)$. Please clarify which one you are using.

Done

10. Line 538: "light tails" should be "exponential tails" I think, since you are using Gaussian. Light tailed distributions are a separate class lighter than exponential tails.

Changed to “exponential tails”

11. Line 659: Equation (19a) has a comma on the right hand side. This does not makes sense to me, unless it is the bivariate nature of the expectation you are trying to capture with a single equation? Please add a brief explanation.

The comma was just punctuating the equation. We removed it to avoid confusion.

12. Line 896: “Is it 16 variations? or 18 variations? I think somewhere else in the paper it says 16?”

It was indeed 18 variations (or 36 if we also include those with/without bias). This has now been moved to the Supplementary though, under “Comparison of selection criteria with inconclusive data”

References

1. Tarantola, Albert. ‘Popper, Bayes and the Inverse Problem’. *Nature Physics* 2, no. 8 (August 2006): 492–94. <https://doi.org/10.1038/nphys375>.
2. Mosegaard, Klaus, and Albert Tarantola. ‘Monte Carlo Sampling of Solutions to Inverse Problems’. *Journal of Geophysical Research: Solid Earth* 100, no. B7 (1995): 12431–47. <https://doi.org/10.1029/94JB03097>.
3. Grünwald, Peter, and Teemu Roos. ‘Minimum Description Length Revisited’. *International Journal of Mathematics for Industry*, 12 March 2020. <https://doi.org/10.1142/S2661335219300018>.
4. Gelman, Andrew, and Yuling Yao. ‘Holes in Bayesian Statistics’. *Journal of Physics G: Nuclear and Particle Physics* 48, no. 1 (1 January 2021): 014002. <https://doi.org/10.1088/1361-6471/abc3a5>.
5. Sivula, T., Magnusson, M., Matamoros, A. A., & Vehtari, A. (2023). Uncertainty in Bayesian Leave-One-Out Cross-Validation Based Model Comparison (No. arXiv:2008.10296). arXiv. <https://doi.org/10.48550/arXiv.2008.10296>
6. Findley, D. F. (1991). Counterexamples to parsimony and BIC. *Annals of the Institute of Statistical Mathematics*, 43(3), 505–514. <https://doi.org/10.1007/BF00053369>
7. Grünwald, P., & Ommen, T. van. (2017). Inconsistency of Bayesian Inference for Misspecified Linear Models, and a Proposal for Repairing It. *Bayesian Analysis*, 12(4), 1069–1103. <https://doi.org/10.1214/17-BA1085>
8. Hüllermeier, Eyke, and Willem Waegeman. ‘Aleatoric and Epistemic Uncertainty in Machine Learning: An Introduction to Concepts and Methods’. *Machine Learning* 110, no. 3 (1 March 2021): 457–506. <https://doi.org/10.1007/s10994-021-05946-3>.

Reviewer #3 (Remarks to the Author):

Summary

In this manuscript, the authors propose a novel method called EMD (for Empirical modelling discrepancy) for model comparison. Their method constructs a “risk distribution” for each model and then computes some distance measure between the two risk distributions as a proxy for the evidence for or against a given model compared to another one. The proposed method is contrasted with other model comparison methods.

The EMD method is original and the manuscript is overall well written. In my view, the biggest strengths of the paper are the novelty of the approach and the emphasis on the epistemic uncertainty (i.e. the uncertainty induced by modelling errors). However, the whole argumentation of the paper relies on few key points that the authors fail to address convincingly.

Major

1. What is the gold standard for model comparison? In the literature, there exists several model comparison methods. Even though it might be computationally intensive to evaluate, I believe the gold standard for model comparison remains the Bayesian Factor (if the prior assumptions are valid) and this for several reasons
 - a) first, it computes exactly what we are looking for, i.e. when the model priors are identical ($p(M_1) = p(M_2)$), then $BF = p(M_1|D)/p(M_2|D)$ which is the ratio of the posterior over the models given the data.
 - b) secondly, the integral needed to compute $p(D|M)$ naturally penalises for high complexity models
 - c) most importantly, it is consistent. See e.g. [1] and see point 2a below. Curiously, the authors criticise the BF for with hardly convincing arguments:
 - d) on L 973, it is mentioned that B^{Bayes} and B^{elpd} provide extremely small values (on Table 2). It might be after all that the two models are effectively indistinguishable for the considered range of λ (this is actually the reason why Fig 8 does not display the range of wavelength considered in the testing of Fig 2 - precisely because we can't see the difference between the two models).
- ii) on L 977, the BF is criticised for being dependent on L . I don't understand this critique. Any model comparison criterion has to depend on L .

- iii) on 979, it is argued that a weakness of the BF is its strong dependence on the prior. Of course the BF depends on the prior - as it should. Now in the P and RJ models, the priors over λ and T are exactly the same for both models, so it is unclear what is the detrimental effect of the prior here.

Now even if the BF would be disregarded as the gold standard for model comparison, what should be the gold standard? If there is no reference, what is the point in proposing a new model comparison method since there is no way to argue how well it performs? Visual inspection should not be the criterion.

2. The proposed method is not principled. After having inappropriately criticised the BF for model comparison, the authors propose a new approach - which is ad-hoc for several reasons.
 - a) Firstly, and most importantly, no consistency proof is provided. The authors should formally show that if one of the candidate model is actually the true model, then the method should favor the true one. A theorem and a formal consistency proof is expected here.
 - b) Secondly, the EMD method depends on a free parameter c . Even though they propose some calibration method (see Eq 28 - which is actually even violated in the provided example, see Fig 7. See also minor comment 1 below), the presence of free parameters gives a sense of an ad-hoc method.
 - c) Thirdly, as recognised on line 786, the interpretation of $B^{\{EMD\}}$ is not very clear. Actually, the interpretability of $B^{\{EMD\}}$ only comes from another criterion (namely $B^{\{epis\}}$) with which it is correlated (provided that c has been chosen accordingly).
 - d) Finally, it is unclear if the proposed method performs well since there is no ground truth reference provided for the comparison (see point 1 above).
3. Overconfidence about overconfidence? The authors criticise existing model comparison methods for their presupposed overconfidence. According to them, overconfidence manifests itself in (a) the lack of saturation of the model comparison criterion when the sample size L increases and in (b) the non-robustness of the criterion (which could flip between strong evidence for and then against a model with small variations of a dataset). However, in order to make any statement about the overconfidence or underconfidence of a criterion, we need some ground truth reference for the confidence. In the absence of such ground truth, it is hard to interpret the authors' desirata (a) and (b) as other than personal views.
4. Open questions in the Table 2. The model comparison is summarised in Table 2. However, there are still several open questions that need to be addressed.

- a) The results displayed in Table 2 are hardly comparable. If all of them were log probability ratios, then the comparison would be fair, but this is not the case. The authors needed to “stretch” some definitions, but this is not very elegant and potentially misleading.
- b) There are few + infinity entries for the $B^{\{EMD\}}$ criterion. This is not a very good sign (isn't this a sign of overconfidence - or rather infinite confidence?). Furthermore, it is unclear why infinity should appear both in the low L and high L regimes (see $B^{\{EMD\}}$, last column, small lambda).
- c) Where do the zero entries come from for the $B^{\{elpd\}}$ criterion?

Minor

1. Fig 7 all curves violate condition 28. Indeed, all the curves in Fig. 7 (and not only for the braun curve), have at least some part in the forbidden area (shaded area).
2. Eq 67 or Eq 36 want to stress the fact that some quantities are expressed in log₁₀ with the expression “/log 10”. I find this writing style misleading as it could be wrongly interpreted as a division by log(10).
3. Eq 11. Shouldn't “<” be replaced by “>”?

References

[1] Casella, George, and Elías Moreno. “Assessing Robustness of Intrinsic Tests of Independence in Two-Way Contingency Tables.” *Journal of the American Statistical Association* 104, no. 487 (September 2009): 1261–71.
<https://doi.org/10.1198/jasa.2009.tm08106>.

Author's reply to Reviewer #3

We thank the reviewer for their comments.

As a general point, we now make it clearer that the specific question we set out to address was how to do model comparisons when models are misspecified, and in particular when replication datasets are drawn from slightly different data-generating distributions (as is the case with real experimental data). Therefore everything we do, and in particular the comparisons to other methods, is with the goal of answering “will a model selected with this method continue to perform when the data-generating process changes slightly?” (e.g. due to experimental variations).

This question is now better framed in the revised version.

1. What is the gold standard for model comparison? In the literature, there exists several model comparison methods. Even though it might be

computationally intensive to evaluate, I believe the gold standard for model comparison remains the Bayesian Factor (if the prior assumptions are valid) and this for several reasons

We acknowledge that Bayes factors are commonly considered a gold standard for model comparison. We ourselves lean to the Bayesian side on the Frequentist ↔ Bayesian scale. Nevertheless, our method is agnostic to one's statistical leanings: it does not itself use priors, but priors can be used when fitting the model (rather than during our focus, i.e. when selecting amongst fitted models). Furthermore, certain practitioners discourage the use of Bayes factors due in particular to their excessive sensitivity to irrelevant aspects of the prior – see for example [1, §7.4], [2, §5], [3, §3.3.2], and the discussion of the “prior predictive” (aka the marginal likelihood) in [4]. There are certainly cases where they *are* appropriate, coding theory being a natural example.

The existing methods which most closely resembles ours, although still with important differences, are probably estimators for the elpd (such as WAIC or PSIS-LOO), which are based on predictive performance. Some would call these gold standards for model comparison [5], although of course this is not universally accepted. The new section 2.9 (and especially Fig. 8) makes it much easier to see how our proposed method relates to existing ones. An important point we stress in that section is that model comparison is not a uniquely defined problem: different situations will warrant different methods. The method we propose is suited to situations where we expect models to be misspecified, and where we want the learned model to generalise beyond the training data.

For the next three points a)-c), we provide our perspective, but elaborate on these issues only minimally in the text of the revised version.

- a) first, it computes exactly what we are looking for, i.e. when the model priors are identical ($p(M_1) = p(M_2)$), then $BF = p(M_1|D)/p(M_2|D)$ which is the ratio of the posterior over the models given the data.

We agree with the formulation of Bayes factors, but whether it is exactly what we are looking for depends on the model selection goal. One could also reasonably argue that the risk is exactly the quantity we care about, since it measures the expected cost of prediction errors.

As we show in the revised Section 2.9, different criteria work within different paradigms. Bayes factors in particular are not designed to compare structurally identical models, and so would not really work with our neural circuit example. The new section at the top of our Results now makes this clear.

It is also difficult to test generalisation performance with Bayes factors.

b) secondly, the integral needed to compute $p(D|M)$ naturally penalises for high complexity models

This is true of course.

An alternative approach, and the one we propose in our Discussion, is to assess a model's predictive performance and complexity separately (e.g. sequentially).

c) most importantly, it is consistent. See e.g. [1] and see point 2a below.

To our understanding, consistency of Bayes factors is only guaranteed when the true model is among the candidates. Indeed, Findley [6] and Grünwald and Ommen [7] give examples where Bayesian methods become inconsistent once the models are misspecified. Since misspecified models are an important part of our motivation, as we have now made clearer, this is something we would be worried about.

In these examples of [6,7], the consistency is instead defined as the criterion recovering the model with the lowest prediction error. If we keep with that definition therefore, a criterion which selects the model minimizing prediction error (i.e. minimizes risk) is consistent by definition, as we explain below. We also included a comment to this effect at the top of our results.

Curiously, the authors authors criticise the BF for with hardly convincing arguments:

The new Section 2.9 and Fig. 8 (which replaces the old Table 2) do a much better job of presenting the various methods in a fair manner. One of the major changes was, rather than comparing specific values in a table, we now graph each criterion's behaviour as a function of dataset size. From this the tendency of criteria to overstate the strength of evidence is more clearly manifested.

We also rewrote the associated text. The original version focused too specifically on those shortcomings we set out to address, and thus presented other criteria too negatively. The new version is more balanced, focusing on how ours is different (in good and bad) from other methods, so that readers can better judge in which circumstances our and other methods are most appropriate.

i) on L 973, it is mentioned that B^{Bayes} and B^{elpd} provide extremely small values (on Table 2). It might be after all that the two models are effectively indistinguishable for the considered range of λ (this is actually the reason why Fig 8 does not display the range of wavelength considered in the testing of Fig 2 - precisely because we can't see the difference between the two models).

The ranges in the previous table (which is now in the Supplementary) was chosen so that in some cases, we can distinguish the models by eye, and in other cases

we can't. So at least in some cases we would expect criteria to pick up a difference. That said, there were implementation issues with the code we used to compute Bayesian criteria, which we have now fixed. The values are still small, but not *as* small.

The new Fig. 8 (which replaces the old Table 2) uses a similar regime as Fig. 7 and the elpd/log evidence values are in line with what one would expect. (We presume the reviewer meant Fig. 7 rather than Fig. 2, since the latter describes a completely different model.)

ii) on L 977, the BF is criticised for being dependent on L. I don't understand this critique. Any model comparison criterion has to depend on L.

Criteria written as some form of $P(D)$ will depend on L since the probability depends on L . But criteria don't need to take this form. For example they can instead depend on the risk, which is the average pointwise loss and thus has a finite limit.

This is in fact one of the major reasons we chose the risk to compare models. As we now emphasize, we want to apply our criterion to experimental data, where it is often possible to return to the lab to collect more data. Therefore we want a criterion for which the precise number of data points does not matter.

Just as importantly, there comes a point where we hit the precision limits of our apparatus, and adding more data is no longer beneficial. Thus the uncertainty associated to our criterion should not vanish, even in the $L \rightarrow \infty$ limit.

Both of these points are now clearly articulated in Section 2.2, and repeated later.

iii) on 979, it is argued that a weakness of the BF is its strong dependence on the prior. Of course the BF depends on the prior - as it should. Now in the P and RJ models, the priors over λ and T are exactly the same for both models, so it is unclear what is the detrimental effect of the prior here.

The argument here is that this dependence can be excessively strong, to the point where the comparison outcome is determined by the choice of the prior rather than the data: Given two models and two similar, non-informative priors, the outcome of the comparison may depend more which one of the two (putatively similar) priors is chosen – than on the data itself. The effect is most sensitive to the width of the priors rather than their precise shape, so whether the models have identical or different priors is not the dominant factor here.

This paraphrases the argument of Gelman and Yao ([16] in the text, [2] below). We have reworded this to hopefully avoid misunderstandings (in particular, we added that this is an issue with non-informative priors), but we don't think it is worth

expanding on this too much, since ultimately this is not a paper about Bayes factors. Interested readers can look up the reference for more details.

While we don't illustrate this phenomenon in our figure, it remains that it is likely to occur whenever one has models with continuous parameters and non-informative priors. Thus this phenomenon should be considered when choosing the model selection criterion most appropriate to a given problem.

Now even if the BF would be disregarded as the gold standard for model comparison, what should be the gold standard? If there is no reference, what is the point in proposing a new model comparison method since there is no way to argue how well it performs? Visual inspection should not be the criterion.

We disagree with the reviewer on this point: if a problem has no accepted solution, this does not mean that we should not try to solve the problem. To address their question more specifically:

- The calibration experiments we propose can be a stringent test of the method's validity: a practitioner can set up as many experiments as they want, and the chosen α value must produce a satisfactory calibration curve for every one of them. More importantly, these tests are operationally driven: they allow the experimentalist to tell the modeller what kinds of confounds to test against. This kind of empirical, domain-informed test is, we believe, ultimately more relevant to modeling scientific data than any generic test we could propose.
- For the same reasons as in [9], we think that graphical analysis like the one we use in Fig.6 for the calibration tests is actually more informative than a single number. After all, it's a lot harder to fit an entire line by accident than a single point.
- The point is to push the field to develop solutions which address current unmet needs of scientists. One thing that seems clear is that developing model comparison methods which are epistemically robust will require many new ideas – more than can be done in a single paper, possibly more even than can be done by a single group. We think there are enough new ideas in this work to substantively push this research question forward.

2. The proposed method is not principled. After having inappropriately criticised the BF for model comparison, the authors propose a new approach - which is ad-hoc for several reasons.

a) Firstly, and most importantly, no consistency proof is provided. The authors should formally show that if one of the candidate model is actually the true

model, then the method should favor the true one. A theorem and a formal consistency proof is expected here.

One of our basic assumptions, which has now been emphasized, is that our models are misspecified, and therefore the original definition of consistency is not particularly relevant. As mentioned above, when models are misspecified, a common way to define consistency is to consider the true one to be the one with lowest prediction error [6,7]. Thus a criterion which selects the model which minimizes risk is consistent by definition.

For a ternary criterion like ours, which can also elect not to reject either model, we could further generalize the notion of consistency to require that, with enough data, the true model is never rejected. In other words: model B is only rejected if $B_{\text{emd}}\{AB\}$ is above our threshold, AND the empirical risk of A is less than that of B.

We have emended the definition of the criterion to include this second condition (EMD rejection rule, below Eq. (16)), and included a short discussion on the issue of consistency in the first section of the Results. We also note this below Eq. (5), along with the remark that – as long as we use a proper scoring rule [10] as our loss – the method will also be consistent in the original sense where one of the candidates is the true model. (The theory of empirical risk minimization assures us that this will still work with finitely many samples.)

In the 2nd paragraph of our discussion, we also remind readers of key differences between our context (which is more typical in science and machine learning) and the usual assumptions of statistical model selection – in particular how those differences impact the question of consistency.

b) Secondly, the EMD method depends on a free parameter c . Even though they propose some calibration method (see Eq 28 - which is actually even violated in the provided example, see Fig 7. See also minor comment 1 below), the presence of free parameters gives a sense of an ad-hoc method.

While true, we don't think that the choice of c is any less ad hoc than choosing a prior for Bayes factors, or the c correction factor in BIC. (And certainly less than the multiple ad hoc choices involved in an MDL analysis.)

It is good here to remember that our method is fundamentally non-parametric – that one parameter c saves us a lot of ad hoc modelling assumptions.

c) Thirdly, as recognised on line 786, the interpretation of B^{EMD} is not very clear. Actually, the interpretability of B^{EMD} only comes from another criterion (namely B^{epis}) with which it is correlated (provided that c has been chosen accordingly).

We think nevertheless that the $B^{\{EMD\}}$ is already robust enough to be used as-is, and hope that it might be improved in the future.

The revised version also better motivates the $B^{\{EMD\}}$, which should help make it more interpretable.

- d) Finally, it is unclear if the proposed method performs well since there is no ground truth reference provided for the comparison (see point 1 above).

As stated above, we a) disagree that the absence of a gold standard method for comparing misspecified models should be disqualifying, and b) believe that our proposed calibration experiments would serve this purpose.

- 3. Overconfidence about overconfidence? The authors criticise existing model comparison methods for their presupposed overconfidence. According to them, overconfidence manifests itself in (a) the lack of saturation of the model comparison criterion when the sample size L increases and in (b) the non-robustness of the criterion (which could flip between strong evidence for and then against a model with small variations of a dataset). However, in order to make any statement about the overconfidence or underconfidence of a criterion, we need some ground truth reference for the confidence. In the absence of such ground truth, it is hard to interpret the authors' desiderata (a) and (b) as other than personal views.

It should be uncontroversial to most natural scientists that, in many situations i) one can return to the lab to collect more data, and ii) at least theoretically, at some point one has more data than needed. As scientists, we therefore want a criterion which is compatible with that experimental reality. In other words, i) it should not matter exactly how many data points we have, and ii) infinite data does not breed infinite certainty.

It should be fairly clear that ii) implies desiderata (a), and i) implies desiderata (b). Since they follow so directly from standard experimental practice, qualifying this as a personal view we think is a bit unfair.

In the new version we explain these desiderata earlier. The new Fig. 8 also does a much better job at showing how, with classic criteria, the difference in criteria between two models is unbounded with L . As can be seen from that figure, our method does in fact implicitly account for finite L ; in the discussion we now also mention that an explicit treatment of the sample size could be a direction for future research.

- 4. Open questions in the Table 2. The model comparison is summarised in Table 2. However, there are still several open questions that need to be addressed.
 - a) The results displayed in Table 2 are hardly comparable. If all of them were log probability ratios, then the comparison would be fair, but this is not the case.

The authors needed to “stretch” some definitions, but this is not very elegant and potentially misleading.

The new Fig. 8 reports log evidence instead of Bayes factors. (Bayes factors are easily recovered as the difference between two log evidence curves.) This allows each criterion to be presented in its usual form, no stretching required.

- b) There are few $+\infty$ entries for the $B^{\{EMD\}}$ criterion. This is not a very good sign (isn't this a sign of overconfidence - or rather infinite confidence?). Furthermore, it is unclear why infinity should appear both in the low L and high L regimes (see $B^{\{EMD\}}$, last column, small λ).

Infinite values just mean the R-distributions don't overlap, so yes, in theory this would mean complete confidence. In this case however the values ∞ appearing in the low L and high L regimes were due to an oversight on our part: we forgot to increase the number of R-distribution samples for this experiment. We were redrawing new datasets for each cell, and for the middle row, it happened that by chance, a few samples did overlap. (The default settings are for computing the $B^{\{EMD\}}$ around the rejection threshold, which requires much fewer samples than far into the tails.)

(As an aside, this is what we meant by the mapping being unfavourable to our own method. In the actual form of the $B^{\{EMD\}}$, which is given as a tail probability, we would get 1 or 0 when there is no overlap. It is only in this log prob ratio form – which we did to match the Bayes factors – that we get these confusing values of ∞ .)

Generally our recommendation would always be to look at the risk-distributions, since they contain strictly more information than the tail probabilities. This is what our new Fig. 8 does.

We redid the calculations for this table (which is now in the Supplementary) with much larger sample sizes, and now the values we get for $B^{\{EMD\}}$ are much closer to what one would expect.

- c) Where do the zero entries come from for the $B^{\{elpd\}}$ criterion?

These were due to implementation errors, which have now been fixed.

Minor

1. Fig 7 all curves violate condition 28. Indeed, all the curves in Fig. 7 (and not only for the braun curve), have at least some part in the forbidden area (shaded area).

To completely avoid the shaded area at all, a curve would need to pass through *exactly* (0.5, 0.5). This seems arbitrary and unreasonable.

A similar issue occurs in [9]: there the test is that the produced histogram should approach a uniform one, but in practice some deviation from this must be tolerated. This is why we say “The visual representation makes it easier to judge the extent to which small violations of equation (34) can be tolerated”.

2. Eq 67 or Eq 36 want to stress the fact that some quantities are expressed in \log_{10} with the expression “/log 10”. I find this writing style misleading as it could be wrongly interpreted as a division by $\log(10)$.

We were in fact dividing by $\log(10)$ here. We used natural logarithms throughout, so if we interpret R as a log probability (which isn't too far off, since our loss is the neg log likelihood), then we need to change its basis from e to 10 . We do this by dividing by $\log 10$, which is the standard change of basis formula for logs.

In any case this has been moved to the Supplementary, along with the table, and we have added parentheses to make the meaning clearer.

3. Eq 11. Shouldn't “<” be replaced by “>”?

Indeed, the transitivity result we originally reported was not quite correct. Thank you for identifying this. We have now corrected it.

References

1. Gelman, A., Carlin, J. B., Stern, H. S., Dunson, D. B., Vehtari, A., & Rubin, D. B. (2021). Bayesian data analysis (online) [Textbook]. <http://www.stat.columbia.edu/~gelman/book/>
2. Gelman, A., & Yao, Y. (2021). Holes in Bayesian Statistics. *Journal of Physics G: Nuclear and Particle Physics*, 48(1), 014002. <https://doi.org/10.1088/1361-6471/abc3a5>
3. Betancourt, M. (2015). A Unified Treatment of Predictive Model Comparison (No. arXiv:1506.02273). arXiv. <https://doi.org/10.48550/arXiv.1506.02273>
4. Gelman, A., Meng, X.-L., & Stern, H. (1996). Posterior Predictive Assessment of Model Fitness Via Realized Discrepancies. *Statistica Sinica*, 6(4), 733.
5. Vehtari, A., Gelman, A., & Gabry, J. (2017). Practical Bayesian model evaluation using leave-one-out cross-validation and WAIC. *Statistics and Computing*, 27(5), 1413–1432. <https://doi.org/10.1007/s11222-016-9696-4>
6. Findley, D. F. (1991). Counterexamples to parsimony and BIC. *Annals of the Institute of Statistical Mathematics*, 43(3), 505–514. <https://doi.org/10.1007/BF00053369>
7. Grünwald, P., & Ommen, T. van. (2017). Inconsistency of Bayesian Inference for Misspecified Linear Models, and a Proposal for Repairing It. *Bayesian Analysis*, 12(4), 1069–1103. <https://doi.org/10.1214/17-BA1085>
8. Vapnik, V. N. (2000). *The Nature of Statistical Learning Theory*. Springer New York: Imprint: Springer.

9. Talts, S., Betancourt, M., Simpson, D., Vehtari, A., & Gelman, A. (2018). Validating Bayesian Inference Algorithms with Simulation-Based Calibration. arXiv:1804.06788 [Stat]. <https://doi.org/10.48550/arXiv.1804.06788>
10. Gneiting, T., & Raftery, A. E. (2007). Strictly Proper Scoring Rules, Prediction, and Estimation. *Journal of the American Statistical Association*, 102(477), 359.

We thank both reviewers for taking the time again to assess our manuscript.

This new version focuses on clarifying our calibration procedure, to address the issues raised by Reviewer #3. We have to this end revised text of the relevant section in the Results, expanded the relevant section of the Methods (adding therein a new Fig. 9 illustrating the effect of the c parameter), and added material discussing limitation to both the Discussion and Supplementary Discussion.

To aid the reviewers, we have included in the resubmission a redlined copy of the manuscript with line numbers, to which we refer to in the answers below.

We hope they find this new version satisfactory.

Reviewer #1 (Remarks to the Author):

I thank the authors for their revision and replies to my questions. I find the revision much easier to follow. In model comparison, typically we can compute the empirical distribution of point-wise loss functions. Take the mean, we get the ELPD. Take the empirical distribution, we can further calculate the empirical probability of one model being better than another model. This paper proposes a future step: to compare the discretionary between the quantile function of the point-wise losses and the quantile under the assumed model.

All my previously raised questions are answered in the rebuttal, and I am satisfied with the answers. I only want to point out a recent reference “The Posterior Predictive Null” by Moran and Blei, which shares similar ideas of synthetic replications and is hence worth some discussion. Looking forward, it makes sense to compare more traditional tools such as the ELPD, Bayes factor, etc., with the R-distribution approach in terms of their convergence rate and robustness—but I understand such a comparison is likely beyond the scope of this current paper.

We are glad the reviewer is satisfied with our revisions.

We also thank them for suggesting the PPN paper by Moran and Blei, which is certainly relevant. We have added the following to the discussion (**lines 1387–1393**):

Moran et al. [61] propose the posterior predictive null check (PPN), which tests whether the predictions of two models are statistically indistinguishable. In contrast to the B^{EMD} , the PPN is purely a significance test: it can detect if two models are distinguishable, but not which one is better, or by how much.

Reviewer #3 (Remarks to the Author):

First, I would like to thank the authors for their detailed response on my comments. In particular, the focus on model misspecification is now very clear, right from the introduction, the new table 2 is more informative than the previous one and the comparison with other methods has been improved.

We are glad the reviewer agrees with most of the changes of the previous version, and thank them for their continued efforts with our manuscript.

In this new version, we have expanded on the calibration experiments, including new experiments using the alternative “Q-distribution” suggested by the reviewer.

We also added a calibration experiment (**Figure 9**) for the other example used in the paper (comparing black body radiation models), which accomplishes two things:

- First, it provides readers an illustration of what an ideal calibration result might look like, with a clean sigmoid remaining within the underconfident regions over the entire interval; we still believe that the original example (comparing neuron models) is more representative of what people may find in practice, but this ideal case provides a pedagogically useful complement.
- Second, it illustrates especially well the relative insensitivity of these curves c , with almost perfect overlap of the curves for c between 2^{-3} and 2^0 . (In the figure we intentionally take c over a very large range, to show how curves behave outside this range.) This is evidence that for the B^{EMD} , the selection of the c parameter is a secondary concern: instead the correlation between B^{EMD} and B^{epis} comes firstly from the special structure of the EMD distribution. Calibration serves to avoid extreme values of the c parameter, rather than fine-tuning it to a precise value.

Figure 9: Calibration plots for the $B_{\text{P,RJ}}^{\text{EMD}}$ criterion comparing Planck and Rayleigh-Jeans models. The value of c increases from left to right and dark to bright. Each curve consists of 4096 experiments aggregated into 128 bins. Wide gaps between points indicate that few experiments produced similar values of B^{EMD} ; this is especially

salient in the left panel, where almost all experiments have either $B^{\text{EMD}} = 0$ or $B^{\text{EMD}} = 1$.

As part of these revisions, we have also further clarified, for ourselves and the readers, how to interpret our induced distributions. The key points are as follows (see answers below for references to text changes):

- We model epistemic uncertainty as *linearly proportional to misspecification*, as measured by the discrepancy between two quantile functions of the loss (c.f. Eq. (27)). It is therefore not all possible epistemic distributions that we model, but a kind of local approximation around the (model+misspecification) combination.
- The parameter c is the conversion factor between misspecification and epistemic uncertainty.

We also reran the size calibration experiments for the neuron models shown in the main text with more simulated experiments (2048 instead of 512). This reduced the level of noise and made trends easier to see.

There are however few concerns that remain.

1. Well posed problem? I appreciate that the authors make it clear that that model misspecification is at the heart of their study. As I understand, this misspecification process is expressed through the epistemic distribution Ω (or the collection of such distributions). Now, I understand that the authors assume this epistemic distribution to be unknown (because in realistic settings, we precisely do not know the ground truth generative process), but this is precisely where the problem becomes ill-posed.

How can we tell if the proposed model comparison method is successful if the misspecification process is unknown? What is the success metric? As I understand, for any known distribution Ω (or collection of such distributions), the authors estimate the free parameter c that offers the best correlation between B^{EMD} and B^{epis} (see also point 2 below). So “success” is claimed when this correlation is high. However, because at the end of the day Ω will be unknown, how can we decide if the chosen c was good or not.

This is an issue with any model comparison method: if the misspecifications are unknown and vary across replications, it is hard to imagine how one could prove deductively that *any* method would succeed in all cases. However, as scientists, unknown epistemic distributions are a fact of life and we need to deal with them somehow.

Thankfully our ignorance is not complete: in fact, we typically know—or can control—quite a lot about the experimental conditions, so that most of the experimental variations can be

modelled with reasonably high accuracy. Focussing on concrete problems should mitigate this ill-posedness issue in at least two ways:

- With more accurate epistemic distributions, we are more confident in the inferred value of c , and thus in the success of the method.
- With more tightly-controlled experiments, we don't need as varied a collection of epistemic distributions Ω . Thus it is also more likely that the same c value will work for all of them (and by extension, for recorded data where the Ω is unknown).

The ultimate measure of success, of course, will be the same as for any scientific theory: whether the selected model continues to predict well on new experiments.

These points are now made in the discussion; the first on **lines 1490–1500**:

[The] property that the same value of c can be used to compare different models in different contexts is the key reason why we expect the B^{EMD} to generalise from simulated epistemic distributions to real-world data. It is clear however that this cannot be true for arbitrary epistemic distributions: we give some generic approaches for improving the outcome of calibrations in the Supplementary Discussion, but the B^{EMD} criterion will always work best in cases where one has a solid experimental control (to minimize variability between replications) and deep domain knowledge (to accurately model both the system and the replications).

The second point is addressed at the end of the discussion, on **lines 1556–1560**:

These choices, along with the assumptions underpinning the B^{EMD} , should also become testable: if they lead to selecting models which continue to predict well on new data, that in itself will provide for them a form of empirical validation.

We have also added a section to the Supplementary Discussion expanding on possible strategies to mitigate this issue and improve calibration experiments.

2. Calibration of the ad-hoc parameter c . Even though I understand that every method has its parameters (such as for example the parameters describing the prior for the Bayes factor), the parameter c is problematic for several reasons:

- a) As mentioned in my previous review, all calibration curves (of Fig 6) violate Eq 34 - not just around the specific $(0,0)$ point, but on a large fraction of the curve.
- b) This point a above suggests that the flexibility offered by the free parameter c is not sufficient to reflect the epistemic uncertainty. This is actually not a surprise because Ω is high-dimensional and c is only 1-dimensional.
- c) Because the parameter c is Ω -specific and because Ω is unknown, I do not see how we can be confident that we chose the right parameter c (see point 1 above).
- d) Finally, it is not straightforward to give an interpretation to the parameter c (let alone give an analytical expression of this parameter c as a function of the problem set up).

(a) As we noted above, and as the new version now emphasizes, we don't actually try to model all epistemic uncertainty—only that which is linearly proportional to misspecification, in the sense of Eq. (27) (which we now emphasize on **lines 518, 662–664, 685–687 and 1536–1540**). This is already an improvement over current practice of ignoring misspecification altogether (what amounts to setting $c=0$ in our formalism), but it means that we should still expect some violations in the calibration plots. Moreover, since the proportionality assumptions amount to a variational approximation around the quantile function q^* , we can expect further violations when the models being compared are very different from the data-generating process (i.e. when the misspecification is high). We added the following to the main text on **lines 982–991**, to explain that one should not expect Eqs. (34,35) to be satisfied perfectly:

There are limits however to the range of validity. Too small values of c will remove any overlap between R -distributions and produce an overconfident B^{EMD} . Too large values of will exaggerate overlaps, underestimating statistical power. Large values can also lead to distortions in the Q process, due to the monotonicity constraint placing an upper bound on the achievable metric variance; we see this as the curves reversing in Fig. 2.5 for $c \geq 2^2$. In the Supplementary Discussion we list some possible approaches to enlarge the range of validity, or otherwise improve the calibration of the B^{EMD} .

The approaches proposed in the Supplementary Discussion (**lines 2682–2715**) include increasing the rejection threshold, using multiple comparison steps (in cases where the optimal c is not the same for all epistemic distributions Ω) and tightening the experimental control (to reduce the variability of Ω).

In Fig. 6, the cases with substantial violations across the range are the ones where the models are most different (M_A vs M_D) and where the inferior model can be rejected just by looking at traces. We are not too worried about such cases, since they correspond to clearly separated R -distributions: if the B^{emd} predicts a tail probability of 0.01%, it doesn't matter if it is off by a factor of 50 – one model is still clearly rejected. It is also the case that the pointwise loss we used to evaluate the neuron activity traces is somewhat naive. We did this to avoid side-tracking the presentation with irrelevant details of neural modelling, but as noted **lines 940–942**, practitioners in this field often prefer system-specific loss functions. As we had already noted at the end of §4.5 in the Methods (**lines 1854–1863**), the loss we chose is biased towards models which produce fewer spikes; we have now promoted this comment to the main Results (**lines 936–948**):

As one might expect, we see some violations of Eq. 2.31 in Fig. 2.5, which can partly be explained by our choice of a pointwise loss: although pedagogical, it tends to prefer models which produce fewer spikes, as we explain in the Methods. This is partly why, for these types of models, practitioners often prefer loss functions based on domain-specific features like the number of spikes or the presence of bursts [23]. We are also less concerned with the calibration of M_A vs M_D , since those models are very different (c.f. Fig. 2.1). Not only is the B^{EMD} not needed to reject M_D in favour of M_A , but the variational approximation expressed by Eq. 2.24 works best when the discrepancy between the model and the observed data (i.e. δ^{EMD}) is not too large.

We would also argue that even when a calibration curve does not satisfy Eqs (34,35), it can still provide useful information. A correlation between B^{EMD} and B^{epis} , even in the presence of violations, suggests that the basic EMD assumptions are sound. One may even be able to proceed with additional cautions, like increasing the rejection threshold or adding a post-hoc correction. This is now included as part of the suggestions in the Supplementary Discussion (**lines 2682–2686** and **2706–2715**). In contrast, if that correlation disappears or goes negative, then a different analysis method is likely needed.

(b) As we have now clarified on **lines 766–768**, the distributions we obtain are a kind of variational approximation around the “mixed PPF” q^* . Most of the heavy lifting is done in the definition of the Q process over quantile functions, which is nonparametrically moulded

to the modelling discrepancy by fitting the functions $\alpha(\Phi)$ and $\beta(\Phi)$ to the misspecification, measured by the function δ^{EMD} (see §4.6.3, starting on line 1990). Through α and β there is therefore quite a lot of flexibility for fitting a large variety of epistemic distributions, as long as they can be expressed as proportional to δ^{EMD} .

However, as the reviewer notes, not all epistemic distributions can be expressed this way. We now make this limitation clearer in the new version (see lines 1493–1500, also quoted above, starting with “It is clear however that this cannot be true for arbitrary epistemic distributions: [...]”). There may be better ways of relating epistemic uncertainty to δ^{EMD} , as we now suggest at the end of the Discussion (lines 1540–1548):

Going forward, it will be important therefore to consider the B^{EMD} criterion an imperfect solution, and to continue looking for ways to better account for epistemic uncertainty. These could include relating misspecification directly to the distribution of losses (going further than the B^Q considered our Supplementary Discussion), or more sophisticated self-consistency metrics than our proposed B^{EMD} . A key strategy here may be to look for domain-specific solutions.

(c) Here, however, two things work in our favour:

- The tight control of epistemic distributions, which can be improved with better domain knowledge and/or experiments;
- The relative insensitivity of B^{EMD} results to the precise value of c , as shown in Supplementary Figure 1 and the new Figure 9. In other words, there is not one right-parameter c : the interpretability of B^{EMD} holds for all c within a range.

The situation would be different if we used something like the Q -distributions suggested by the referee in point 3b below, where tuning c directly rescales the calibration curves (see the new Supp Fig. 4 below).

(d) We have clarified this in the new version by a series of clearly stated assumptions, which progressively build up our basic hypothesis that epistemic uncertainty is linearly proportional to misspecification, as measured by the discrepancy function δ^{EMD} (Eq. 27):

EMD assumption (version 1): Candidate models represent that part of the experiment which we understand and control across replications. (lines 481–483)

EMD assumption (version 2): The variability of R_A across replications is predicted by the model discrepancy δ_A^{EMD} . (lines 508–510)

EMD principle: For a given model M_A , differences between its PPFs on two replicate datasets should be proportional to δ_A^{EMD} . **(lines 665–667)**

The parameter c is precisely the proportionality or conversion constant between those two quantities, as given by Eq. (27). Since the process Q is already fitted nonparametrically to q^* and δ^{EMD} using our HB approach (see Fig. 5d), we are not particularly concerned by the lack of analytical expression for c .

3. Relevant Risk distribution? My understanding is that the whole argument of the paper is that it is better to consider the R(isk)-distribution (instead of the empirical risk - which is a scalar) if we want to take into account the uncertainty due to the model misspecification. This R-distribution is obtained through some stochastic process on quantile functions.

Indeed, accounting for epistemic uncertainty, which we relate to misspecification, is the point of this paper.

This brings two questions:

- a) Dependence on the ad-hoc parameter c . Since the center of the R-distribution is precisely the (expected) risk (Eq 26), the ordering between model A and B will remain unchanged if we take into account the R-distribution perspective instead of the (expected) risk perspective. Indeed, if $E[R_A] < E[R_B]$ then $p(R_A < R_B) > 0.5$. So the only thing that changes is the level of confidence that one model wins over the one. This level confidence is directly influenced by the variance of the R-distribution which is scaled by the ad-hoc parameter c whose calibration is questionable (see comment 2). So it remains unclear to me to what extent this R-distribution perspective is really relevant.
- b) Why not using the Q-distribution? Since $(x_i, y_i) \sim M_{\text{true}}$ are random variables, $Q(x_i, y_i, M_A)$ is also a random variable with a given distribution (let's call it the Q-distribution). The authors may want to argue why the R-distribution (which is rather complicated to compute and requires the ad-hoc parameter c) is better than the Q-distribution which is straightforward to use. One way to show this could be to perform the same analysis as in Fig. 6 but with B^Q instead of B^{EMD} . A priori it is unclear to me why B^{EMD} should be better correlated with B^{epis} than B^Q (especially if an additional free parameter c is given to the Q-distribution (in order to make the comparison fair), i.e. $Q' = Q + \text{epsilon}$, where $\text{epsilon} \sim N(0, c)$).

It is perhaps worth emphasizing that the moments in our Eqs. (26) and (27) are with respect to the Aitchison measure, meaning that distributions satisfying those equations are indeed symmetric about q^* , but only after applying a logistic transformation. One consequence of this is that changing the value of c can – to a limited extent – reshape R-distributions and even displace their centers. We have added an example of this with **Supp. Fig. 2**.

(As a secondary point, it is actually not the case that $\mathbb{E}[R_A] < \mathbb{E}[R_B]$ implies $p(R_A < R_B) > 0.5$. Consider for example two binary random variables: $R_A \sim \text{Bern}(0.5) - 0.5$ and $R_B \sim 2 \text{Bern}(0.9)$. Then $\mathbb{E}[R_A] = 0 < 1 = \mathbb{E}[R_B]$, and yet $P(R_A < R_B) = 0.45 < 0.5$. This apparent paradox is related to the question of transitivity which we discuss in the supplementary, and is the reason why in the previous revision we added the second condition to the EMD rejection rule (c.f. **line 533**.)

The reshaping of R -distributions is due to the fact that there are structural constraints on quantile functions, stated in our Desiderata (Sec. 2.6.1): they must be monotone and generated by a non-accumulating process. By defining the process Q directly on quantile functions, we are able to include those constraints in the definition of Q , thus ensuring that the samples used to compute R -distributions are more faithful representatives of experimental variations. In contrast, these structural constraints are ignored when we simply add Gaussian noise to the Q -distributions.

Supplementary Figure 2: R -distributions for the four candidate neuron models for different values of c .

The enforcement of these constraints is, to a large extent, the reason why our process Q is more complicated. However, it is also what makes the induced R -distributions more than just a Gaussian spreading of the empirical risk. By encoding the key structural constraints into the definition of Q , we make the entire procedure much less sensitive to the choice of c . We see this in the new **Supplementary Figure 4**: for c values within a certain range, the calibration curves for B^{EMD} overlap, whereas those for the equivalent B^Q criterion proposed by the reviewer in **3(b)** do not overlap.

Supplementary Figure 4: (a) Calibration experiments for the neuron model, using B^Q (equation (S26)) instead of B^{EMD} (equation 16) as a comparison criterion. To better show the distribution of experiments, each histogram bin (see Calibration experiments in the Methods) is represented as a point. (b) Same curves as in Fig. 6, this time with bins presented as points to ease comparison with (a). Compared to (a), points are more uniformly distributed along both the horizontal and vertical axes. All panels use the same set of sensitivity values (either c_Q or c), with colours as indicated in the central legend.

The observed insensitivity to c is possible because c does not scale the variance of the R -distributions directly, but rather determines the conversion from model discrepancy to epistemic uncertainty. In contrast, the suggested B^Q calibration curves behave exactly as one would expect: increasing c proportionally squeezes the curves horizontally towards the center at 0.5.

Note however that this does not mean that we get the same B^{EMD} for different values of c ; we explain this as follows on **lines 973–981**:

Within its range of validity, the effect of c is somewhat analogous to a confidence level: just like different confidence levels will lead to different confidence intervals, different values of c can lead to different (but equally valid) values of $B_{AB;c}^{\text{EMD}}$. See the Supplementary Discussion for more details. Note also that due to the constraints of the Q process, increasing c does not simply increase the spread of R -distributions, but can also affect their shape; this is illustrated in **Supp. Fig. 2**.

In short, regarding **3(a)**, an R -distribution takes its relevance from the fact that a) the underlying Q process is fitted to the discrepancy function δ^{EMD} , under b) structural constraints particular to quantile functions. The relation from discrepancy to epistemic uncertainty is then a hypothesis—expressed via the proportionality parameter c —which was already stated in the previous version (top of §2.4) but which we have now further

emphasized. In our experiments, the B^{EMD} probabilities were relatively insensitive to the value of c over certain ranges, supporting the idea that the shape of the discrepancy function δ^{EMD} (i.e. the measure of misspecification) is more important than the value of c . This in turn means that the value of c does not need to be refitted to each epistemic distribution: similar epistemic distributions can use the same value.

As for **3(b)**, the Q distribution in a sense trades a simpler distribution for a more complicated set of assumptions. Indeed, for a B^Q criterion to work, it requires that the differences in losses (Q_A vs Q_B) be predictive of the epistemic uncertainty—not utterly inconceivable, but certainly less direct than simply scaling up the misspecification measured by δ^{EMD} . This is borne out in the new experiments of **Supp. Fig. 4**: the B^Q curves show very sharp transitions from $B^{\text{epis}}=0$ to $B^{\text{epis}}=1$. In other words, while we can use the $B^Q=0.5$ threshold to select a model, the value of B^Q itself does not tell us with which certainty we can make that selection. Moreover, in some cases B^Q becomes *anti-correlated* with B^{epis} at the edges (c.f. **Supp. Fig. 4**). These results suggests that the loss differences by themselves are insufficient to get a good predictor of epistemic uncertainty.

We agree however that Q -distributions should contain at least some information on misspecification; as we say on **lines 2730–2738**:

A possible argument in favour of using B^Q as a criterion is that increasing the amount of misspecification – for example by increasing the unmodelled bias B_0 in equation (38) – will affect the distribution of losses in some way. So although the shape of the PPFs is mostly determined by aleatoric uncertainty (as evidenced by the similarity of the distributions in Supp. Fig. 3a), one can expect them to also contain some information about the amount of misspecification – and thereby also the amount of epistemic uncertainty.

For that reason we agree that it would be interesting to investigate what Q -distributions might say about epistemic uncertainty, and now mention this in the main text’s Discussion on **lines 1540–1548**. (The new text, which begins “Going forward, it will be important [...]”, was already quoted above.)

4. Consistency by construction? At the beginning of the results, the authors write “A common consideration with model selection criteria is whether they are consistent, i.e. whether they eventually select the true model when the number of samples grows. When models are misspecified, this is usually expressed as whether a criterion selects the model with the lowest risk [27;28]”

It seems to me that redefining consistency as whether a criterion selects the model with the lowest risk is neither appropriate nor necessary. Indeed, this redefinition evades the real question (whether the true model is selected when the number of sample grows) and replaces it by a tautology. I quickly looked at those refs [27 and 28], but I didn't see this redefinition of consistency. Also, it should be noted that other authors (e.g. De Blasi, 2013) did consider the proper consistency definition in the context of misspecified models.

REFS - De Blasi, Pierpaolo, and Stephen G. Walker. “Bayesian Asymptotics with Misspecified Models.” *Statistica Sinica*, 2013.
<https://doi.org/10.5705/ss.2010.239>.

The unavoidable problem with discussing consistency in the case of misspecified models is that by definition, the true model can never be recovered. So one necessarily needs to generalize the notion of consistency somehow. Both de Blasi (2013) and Grünwald and van Ommen [28] do this by defining “pseudo-true” models: those models which minimize the Kullback-Leibler divergence with respect to the true model. (The authors then prove or disprove convergence to the pseudo-true model under their assumptions.) While there are good conceptual arguments for choosing the KL-divergence as the notion of distance, it is still a choice, and depending on the application may not be the most useful one.

Note that since the true model is fixed, minimizing the KL-divergence is equivalent to minimizing the expected log likelihood. So when we define the loss Q as the negative log likelihood—as we do in our paper—these pseudo-true models really are exactly those that minimize the risk. This equivalence between consistency and minimization of the risk is formalized by “proper” objective functions: those objectives which recover the true model when they are optimized (and the true model is in the model space). As we state in §2.1, the log likelihood is well-known to be proper.

There is some justification therefore for defining consistency in terms of convergence to the minimal risk solution; indeed, this is exactly the definition of a consistent learning process in statistical learning theory [Vapnik §2.1,2.2], and also the one used by Findley [27, C^* on p. 507]. Note that there is still something to prove here, since the “key theorem of learning

theory” [Vapnik §2.2] is precisely about giving necessary and sufficient conditions for (non-trivial) consistency of a learning machine.

However, we agree with the reviewer that the text addressing this in §2.1 was too expedient: while the consistency of a selection rule and that of a learning machine are clearly related, the two concepts are not entirely interchangeable (the learning machine frame subsumes the model selection one). We have now updated §2.1 as follows:

- Clearly distinguished these two definitions of consistency.
 - Added de Blasi 2013 as reference [29].
 - Recalled the sufficient conditions for consistency in statistical learning theory.
- Stated clearly that if one seeks consistency in the sense of minimal KL-divergence, then using the negative log likelihood as the loss function is likely the best choice.

The updated text reads as follows (**lines 204–232**):

When models are misspecified, none of the candidates are actually the true model, so the definition of consistency is often generalised to mean convergence to the “pseudo-true” model: the one with the smallest Kullback-Leibler divergence D_{KL} from the true model [28,29]. If consistency in this sense is desired, then using the log likelihood for Q is recommended, since for a fixed true model, minimizing the D_{KL} is equivalent to minimizing the log likelihood. The log likelihood is also known to be *proper* (see e.g. references 30, 31 or 32 (Chap. 7.1)); a selection rule which minimizes the log likelihood is thus also consistent in the original sense of asymptotically selecting the true model when it is among the candidates.

The statistical learning literature instead defines a “learning machine” as *consistent* if it asymptotically minimizes the true risk R when trained with the empirical risk \hat{R}_A [24,26]. As long as we have a finite number of candidate models, our procedure will satisfy this notion of consistency.

Response to Reviewers

We thank all reviewers for their time and contributions throughout the review process.

Following the positive assessment of the previous round, this final revision contains only small fixes addressing the issues identified by Reviewer #3.

Reviewer #1 (Remarks to the Author):

I would like to thank the authors for having addressed my major comments. In particular, the authors

- added Fig 9 on the calibration method - which I find helpful,
- performed the proposed analysis with the \hat{B}^Q and showed that it is less suitable than the \hat{B}^{EMD} criterion - which I find convincing,
- better highlight the limitations of the proposed approach (in particular the paragraph from L 1536-1548)

We are happy to learn that our additions have assuaged the reviewer's remaining doubts, and thus strengthened the manuscript.

Please find below few last minor comments:

1. The empirical risk equation is equation (5). However in several places in the manuscript, it is referred as equation (2.1). (e.g. lines 263, caption of Fig3, line 536, line 583,...).

Fixed.

2. Fig 6. The label of the y-axis (i.e. $\hat{B}^{\{\text{epis}\}}$) should be displayed

Indeed, this was an omission on our part. It is fixed in the final version.